# Extrapolation and Spectral Bias of Neural Nets with Hadamard Product: a Polynomial Net Study

**Yongtao Wu,    Zhenyu Zhu,    Fanghui Liu,    Grigorios G Chrysos,    Volkan Cevher**

EPFL, Switzerland

{[first name].[surname]}@epfl.ch

## Abstract

Neural tangent kernel (NTK) is a powerful tool to analyze training dynamics of neural networks and their generalization bounds. The study on NTK has been devoted to typical neural network architectures, but it is incomplete for neural networks with Hadamard products (NNs-Hp), e.g., StyleGAN and polynomial neural networks (PNNs). In this work, we derive the finite-width NTK formulation for a special class of NNs-Hp, i.e., polynomial neural networks. We prove their equivalence to the kernel regression predictor with the associated NTK, which expands the application scope of NTK. Based on our results, we elucidate the separation of PNNs over standard neural networks with respect to extrapolation and spectral bias. Our two key insights are that when compared to standard neural networks, PNNs can fit more complicated functions in the extrapolation regime and admit a slower eigenvalue decay of the respective NTK, leading to a faster learning towards high-frequency functions. Besides, our theoretical results can be extended to other types of NNs-Hp, which expand the scope of our work. Our empirical results validate the separations in broader classes of NNs-Hp, which provide a good justification for a deeper understanding of neural architectures.

## 1 Introduction

In deep learning theory, neural tangent kernel (NTK) [Jacot et al., 2018] is a powerful analysis tool that links the training dynamics of neural networks (NNs) trained by gradient descent to kernel regression [Jacot et al., 2018, Arora et al., 2019]. NTK provides a tractable analysis for several phenomena in deep learning, e.g., the global convergence of gradient descent [Chizat et al., 2019, Du et al., 2019a,c], the inductive bias behind NNs [Bietti and Mairal, 2019], the spectral bias toward different frequency components [Cao et al., 2019, Choraria et al., 2022], the extrapolation behavior [Xu et al., 2021], and the generalization ability [Huang et al., 2020]. The study on the NTK has been devoted to typical NNs architectures, e.g., fully-connected NNs [Jacot et al., 2018], residual NNs [Tirer et al., 2020, Huang et al., 2020], convolutional NNs [Arora et al., 2019], graph NNs [Du et al., 2019b] and recurrent NNs [Alemohammad et al., 2021].

Recently, NNs with Hadamard products (NNs-Hp), e.g., StyleGAN [Karras et al., 2019], polynomial neural networks [Chrysos et al., 2020], non-local multiplicative networks [Babiloni et al., 2021], have received increasing attention due to their expressivity and efficiency over traditional NNs [Chrysos et al., 2021a, Campbell and Broun, 2000, Su et al., 2020]. There have been several works attempting to demystify the success of NNs-Hp. For instance, Fan et al. [2021] prove that second-degree multiplicative interactions allow NNs-Hp to enlarge the set of functions that can be represented exactly with zero error. Choraria et al. [2022] reveal that NNs-Hp with second-degree multiplicative interactions yield a faster learning of high-frequency function during training in the NTK regime. Yet, the theoretical analysis of NNs-Hp with high-degree multiplicative interactions is still unclear. More importantly, when using NTK for analysis, only deriving the NTK matrix is not enough. The

36th Conference on Neural Information Processing Systems (NeurIPS 2022).

complete and rigorous proof is achieved by including the stability of empirical NTK during training and the equivalence to kernel regression. This is crucial to allow for NTK-based analysis of typical NNs [Arora et al., 2019, Tirer et al., 2020] but is still missing for NNs-Hp.

Polynomial neural networks (PNNs) [Chrysos et al., 2021a], a special class of NNs-Hp [Jayakumar et al., 2020], have showcased remarkable performance on a broad range of applications. As a step for analyzing NNs-Hp, in this work, we take PNNs as an example, derive the NTK for PNNs with high-degree multiplicative interactions and present a rigorous proof for the equivalence to the kernel regression predictor. This analysis enables us to further examine properties of PNNs in a theoretical perspective, e.g., the extrapolation [Haley and Soloway, 1992, Barnard and Wessels, 1992, Xu et al., 2021]. Neural networks have demonstrated a stellar in-distribution performance but admit some weaknesses in extrapolating simple arithmetic problems [Saxton et al., 2019] or learning simple functions [Haley and Soloway, 1992, Sahoo et al., 2018]. Recently, Xu et al. [2021] theoretically and empirically point out that two-layer fully-connected NNs with ReLU can only extrapolate to linear functions. The contrast on the in-/out of-distribution performance of standard NNs motivates us to scrutinize the extrapolation performance of PNNs. Additionally, studying the NTK of PNNs also allows us to investigate its spectral bias.

Overall, our main contributions and findings can be summarized as follows:

- We derive the NTK formulation for PNNs with high-degree multiplicative interactions, and give a concrete bound of the widths requirement for convergence to the NTK at initialization, and stability during training, which allows us to bridge the gap among PNNs trained via gradient descent and kernel regression predictor.

- We provably demonstrate the extrapolation behavior of PNNs as well as other NNs-Hp, including multiplicative filter networks and non-local multiplicative networks. Our findings highlight that PNNs can extrapolate to unseen data in a non-linear way. Besides, the spectral analysis of NTK of PNNs is also given for better understanding. PNNs admit a slower eigenvalue decay when compared to standard NNs, which leads to a faster learning towards high-frequency functions.

- We empirically show the advantage of NNs-Hp over standard NNs in learning commonly used functions, performing arithmetic extrapolation in real-world dataset, and conducting visual analogy extrapolation task. We scrutinize the role of multiplicative interactions in the task of learning spherical harmonics.

## 2   Background

In this section, we establish the notation, provide an overview of the NTK, and summarize the most closely related work in NNs-Hp as well as extrapolation.

### 2.1   Notation

The core operators and symbols are summarized in Table 2 at Appendix A. Vectors (matrices) are symbolized by lowercase (uppercase) boldface letters, e.g., $\boldsymbol{a}$, $\boldsymbol{A}$. We use the shorthand $[n] := \{1, 2, \ldots, n\}$ for a positive integer $n$. We use $\{\boldsymbol{x}_i\}_{i=1}^{|\mathcal{X}|}$ and $\{y_i\}_{i=1}^{|\mathcal{X}|}$ to present the input features and their labels of the training set $(\mathcal{X}, \mathcal{Y})$ in a compact space, where $|\mathcal{X}|$ denotes the cardinality. We symbolize by $K(\boldsymbol{x}, \boldsymbol{x}')$ the neural tangent kernel with respect to input $\boldsymbol{x}$ and $\boldsymbol{x}'$, the kernel matrix $\boldsymbol{K} \in \mathbb{R}^{|\mathcal{X}| \times |\mathcal{X}|}$ with $K^{(ij)} = K(\boldsymbol{x}_i, \boldsymbol{x}_j)$. Next, we denote by $\boldsymbol{\theta}_t$ the parameter vector, $\ell_2(\boldsymbol{\theta}_t)$ the empirical training loss, and $\hat{\boldsymbol{K}}_t$ the empirical NTK Gram matrix at time step $t$. The following notation is used:

$$\ell_2(\boldsymbol{\theta_t}) = \frac{1}{2} \sum_{(\boldsymbol{x}_i, y_i) \in (\mathcal{X}, \mathcal{Y})} (f(\boldsymbol{x}_i; \boldsymbol{\theta}_t) - y_i)^2, \quad f(\boldsymbol{\theta}_t) = \mathrm{vec}(\{f(\boldsymbol{x}_i; \boldsymbol{\theta}_t)\}_{\boldsymbol{x}_i \in \mathcal{X}}) \in \mathbb{R}^{|\mathcal{X}|},$$

$$J(\boldsymbol{\theta}_t) = \frac{\partial f(\boldsymbol{\theta}_t)}{\partial \boldsymbol{\theta}} \in \mathbb{R}^{|\mathcal{X}| \times |\boldsymbol{\theta}|}, \qquad \hat{\boldsymbol{K}}_t = J(\boldsymbol{\theta}_t)J(\boldsymbol{\theta}_t)^\top \in \mathbb{R}^{|\mathcal{X}| \times |\mathcal{X}|}.$$

## 2.2 Neural tangent kernel

Neural networks (NNs) are relevant to the kernel method, under proper initialization [Daniely et al., 2016, de G. Matthews et al., 2018]. Jacot et al. [2018] provably demonstrate the equivalence between the training dynamics by gradient descent and kernel regression induced by NTK when employing the $\ell_2$ loss. Below, we recall the exact formula regarding the NTK of $N$-layer ($N > 2$) fully-connected NNs with ReLU activation functions $\sigma$. The corresponding NTK $K(\boldsymbol{x}, \boldsymbol{x}') = K_N(\boldsymbol{x}, \boldsymbol{x}')$ could be computed recursively by:

$$K_0(\boldsymbol{x}, \boldsymbol{x}') = \Sigma_0(\boldsymbol{x}, \boldsymbol{x}') = \boldsymbol{x}^\top \boldsymbol{x}', \quad K_n(\boldsymbol{x}, \boldsymbol{x}') = \Sigma_n(\boldsymbol{x}, \boldsymbol{x}') + 2K_{n-1}(\boldsymbol{x}, \boldsymbol{x}') \cdot \dot{\Sigma}_n(\boldsymbol{x}, \boldsymbol{x}'),$$

$\forall n \in [N]$, where the covariance $\Sigma_n$ and its derivative $\dot{\Sigma}_n$ are defined as:

$$\Sigma_n(\boldsymbol{x}, \boldsymbol{x}') = 2\mathbb{E}_{(u,v)\sim\mathcal{N}(0,\boldsymbol{\Lambda}_i)}[\sigma(u)\sigma(v)], \quad \dot{\Sigma}_n(\boldsymbol{x}, \boldsymbol{x}') = \mathbb{E}_{(u,v)\sim\mathcal{N}(0,\boldsymbol{\Lambda}_n)}[\sigma'(u)\sigma'(v)]$$

$$\boldsymbol{\Lambda}_n = \begin{pmatrix} \Sigma_{n-1}(\boldsymbol{x}, \boldsymbol{x}) & \Sigma_{n-1}(\boldsymbol{x}, \boldsymbol{x}') \\ \Sigma_{n-1}(\boldsymbol{x}, \boldsymbol{x}') & \Sigma_{n-1}(\boldsymbol{x}', \boldsymbol{x}') \end{pmatrix}, \forall n \in [N].$$

Furthermore, the aforementioned NTK is extended to residual NNs [Tirer et al., 2020, Huang et al., 2020], convolutional NNs [Arora et al., 2019], graph NNs [Du et al., 2019b], and recurrent NNs [Alemohammad et al., 2021]. One of the roles of such kernel is to analyze the training behavior of the neural network in the over-parameterization regime [Allen-Zhu et al., 2019, Chizat et al., 2019, Du et al., 2019a,c, Zou et al., 2020]. For instance, Lee et al. [2019] showcase that NNs under the NTK parameterization trained via gradient descent of any depth evolve to linear models. Meanwhile, the inductive bias of convolutional networks, e.g., deformation stability of the images, has been studied in the NTK regime [Bietti and Mairal, 2019].

## 2.3 Neural networks with Hadamard product

The ideas of augmenting NNs with Hadamard products to allow multiplicative interactions can be traced back to at least [Ivakhnenko, 1971] that investigate the learnable polynomial relationships. Most of the early work e.g., Group Method of Data Handling [Ivakhnenko, 1971], pi-sigma network [Shin and Ghosh, 1991] do not scale well for high-dimensional signals. Chrysos et al. [2021b] factorize the weight of NNs-Hp based on tensor decompositions to reduce the number of parameters. They exhibit how to convert popular networks, such as residual networks, and convolutional NNs to the form of NNs-Hp. StyleGAN can be also considered as a special type of NNs-Hp [Chrysos et al., 2019]. New efforts have recently emerged to improve the architecture of the network with Hadamard products [Chrysos et al., 2022, Babiloni et al., 2021, Chrysos et al., 2021a]. In this work, we adopt the complementary approach and focus on the extrapolation as well as the spectral bias from a theoretical perspective.

## 2.4 Extrapolation

The study of extrapolation properties of NNs dates at least back to the 90's [Barnard and Wessels, 1992, Kramer and Leonard, 1990]. Experimental results show poor performance of NNs in case of learning simple functions [Barnard and Wessels, 1992]. Browne [2002] also suggest that fully-connected NNs cannot extrapolate well and then illustrate how the representation of the input impact the extrapolation. Xu et al. [2021] provably present the extrapolation behavior of fully-connected NNs and Graph neural networks. Specifically, they show that two-layer fully-connected NNs with ReLU activation function extrapolate to linear function in extrapolation region. Our work exhibits that NNs-Hp can learn high degree nonlinear function. Apart from fully-connected NNs, Martius and Lampert [2016], Sahoo et al. [2018] showcase a novel family of functions with linear mapping and a non-linear transformation, which allows to use sine and cosine as nonlinearities, enabling such networks to learn well in analytical expressions. Note that there exist multiplication units in EQL, which is similar to the multiplicative interactions in NNs-Hp. Lastly, extrapolation is often considered in the context of out-of-distribution (OOD). There are other types of OOD problems with specific setting among machine learning community [Shen et al., 2021]. Domain adaption assumes the source and the target domains lie in the same feature space but with different distributions [Kouw and Loog, 2019], which differs from extrapolation. Another category of methodologies to solve the OOD generalization problem, called invariant learning, aims to discover high-level invariance feature from low-level observations through latent causal mechanisms [Arjovsky et al., 2019, Rosenfeld et al., 2021]. We believe our analysis can also encourage the usage of NNs-Hp in these OOD problems.

# 3 Analysis of polynomial neural networks

Our analysis admits the following structure: we firstly study the NTK of PNNs in Section 3.1, which allows us to conduct analysis towards extrapolation in Section 3.2, and spectral bias in Section 3.3. In Appendix F and G, of the supplementary, we consider extensions beyond PNNs to other families of NNs-Hp, e.g. multiplicative filter networks and non-local networks with Hadamard product.

## 3.1 Neural tangent kernel

We now derive the NTK for PNNs, then we bridge the gap between the PNNs trained by gradient descent with respect to squared loss and the kernel regression predictor involving the NTK. The goal of such networks is to learn an $N$-degree ($N \geq 2$) polynomial expansion that outputs $f(\boldsymbol{x}) \in \mathbb{R}$ with respect to the input $\boldsymbol{x} \in \mathbb{R}^d$. For simplifying the proof, we consider the following formulation, which is a reparameterization version of PNNs [Zhu et al., 2022]. The output is given by:

$$\boldsymbol{y}_1 = \sqrt{\frac{2}{m}}\sigma(\boldsymbol{W}_1\boldsymbol{x}), \ \ f(\boldsymbol{x}) = \sqrt{\frac{2}{m}}(\boldsymbol{W}_{N+1}\boldsymbol{y}_N), \ \ \boldsymbol{y}_n = \sqrt{\frac{2}{m}}\sigma\left(\boldsymbol{W}_n\boldsymbol{x}\right) * \boldsymbol{y}_{n-1}, \ n = 2,\ldots,N \,, \tag{1}$$

where $\sigma$ is the ReLU activation function, each element in $\boldsymbol{W}_{N+1} \in \mathbb{R}^{1 \times m}$ and $\boldsymbol{W}_n \in \mathbb{R}^{m \times d}$, $\forall n \in [N]$ is independently sampled from $\mathcal{N}(0,1)$. Three remarks are in place: a) We multiply by the scaling factor $\sqrt{\frac{2}{m}}$ after each degree to ensure that the norm of the network output is preserved at initialization with infinite-width setting. b) ReLU is usually required to increase the performance of NNs-Hp in experiments [Chrysos et al., 2021b]. c) The original formulation before reparameterization that is used in practice can be founded in Appendix A.1.

**Theorem 1.** *The NTK of $N$-degree PNNs, denoted by $K(\boldsymbol{x}, \boldsymbol{x}')$, can be derived as:*

$$K(\boldsymbol{x}, \boldsymbol{x}') = 2N \cdot \langle \boldsymbol{x}, \boldsymbol{x}'\rangle \kappa_1(\boldsymbol{x}, \boldsymbol{x}')(\kappa_2(\boldsymbol{x}, \boldsymbol{x}'))^{N-1} + 2(\kappa_2(\boldsymbol{x}, \boldsymbol{x}'))^N \,, \tag{2}$$

*where $\kappa_1$ and $\kappa_2$ are defined by taking the random Gaussian vector $\boldsymbol{w} \in \mathbb{R}^d$*

$$\kappa_1 = \mathbb{E}_{\boldsymbol{w} \sim \mathcal{N}(\boldsymbol{0}, \sqrt{\frac{2}{m}} \cdot \boldsymbol{I})}\left(\dot{\sigma}(\boldsymbol{w}^\top \boldsymbol{x}) \cdot \dot{\sigma}(\boldsymbol{w}^\top \boldsymbol{x}')\right), \kappa_2 = \mathbb{E}_{\boldsymbol{w} \sim \mathcal{N}(\boldsymbol{0}, \sqrt{\frac{2}{m}} \cdot \boldsymbol{I})}\left(\sigma(\boldsymbol{w}^\top \boldsymbol{x}) \cdot \sigma(\boldsymbol{w}^\top \boldsymbol{x}')\right). \tag{3}$$

The proof, which is provided in Appendix B.1, is based on the standard NTK calculations. Differently from the NTK of fully-connected NNs, the existence of multiplicative interaction in PNNs induces the product form of multiple kernels.

Next, we provide the following theorem that gives a concrete requirement for the width of the networks that is sufficient for nonasymptotic convergence to the NTK at initialization,

**Theorem 2.** *(Convergence to the NTK). Consider $N$-degree PNNs, and assume that the width $m \geq 2^{4N-2}\log^{2N-1}(2N/\delta)$ for any $\delta \in (0,1)$, then given two inputs $\boldsymbol{x}, \boldsymbol{x}'$ on the unit sphere, with probability at least $1 - \delta$ over the randomness of initialization, we have that*

$$|\langle \nabla f(\boldsymbol{x}), \nabla f(\boldsymbol{x}')\rangle - K(\boldsymbol{x}, \boldsymbol{x}')| \leq 4N\rho e\sqrt{\frac{\log(2N/\delta)}{m}},$$

*where $\rho = \sqrt{2}^{2N-1}\sqrt{8}e^3(2\pi)^{1/4}e^{1/24}\left(e^{2/e}(2N-1)/2\right)^{(2N-1)/2}$.*

**Remark:** This result exhibits that the inner product of the Jacobian converges to the NTK at initialization, which has not been studied before for PNNs. This theorem allows us to further analyze the extrapolation of networks from the perspective of the NTK. It should be noted that the term $\rho$ and the width are exponential with respect to the degree $N$, but the degree $N$ is not large in practice, e.g., at most 15 in Chrysos et al. [2021a]. Hence the bound is fair and reasonable.

The technical key issue of the proof is to provide probability estimates for the multiplication of several sub-exponential random variables. To this end, we rely on the concentration of sub-Weibull random variables [Zhang and Chen, 2020] to complete the proof, which is deferred to Appendix B.2.

Below, we show that under certain conditions, the limiting NTK of PNNs stays constant when training with gradient descent using the squared loss.

**Theorem 3** (Stability of the NTK during training). *Given PNNs in Eq. (1), assume $\lambda_{\min}(\boldsymbol{K}) > 0$ and the training data $(\mathcal{X}, \mathcal{Y})$ in a compact space admitting $\boldsymbol{x} \neq \tilde{\boldsymbol{x}}$ for all $\boldsymbol{x}, \tilde{\boldsymbol{x}} \in \mathcal{X}$, then there exist some constants $R_0 > 0$, $M > 1$, and $Q > 1$ such that for every $m > M$, when minimizing the squared loss with gradient descent and sufficient small learning rate $\eta_0 < 2(\lambda_{\min}(\boldsymbol{K}) + \lambda_{\max}(\boldsymbol{K}))^{-1}$, the following inequality holds with high probability over the random initialization of model parameters:*

$$\sup_t \|\hat{\boldsymbol{K}}_t - \hat{\boldsymbol{K}}_0\|_F \leq \frac{6Q^3 R_0}{\lambda_{\min}(\boldsymbol{K})\sqrt{m}} \, . \tag{4}$$

**Remark:** Eq. (4) shows that $\hat{\boldsymbol{K}}_t \xrightarrow{m \to \infty} \hat{\boldsymbol{K}}_0$. Combining this with Theorem 2 that states $\hat{\boldsymbol{K}}_0 \xrightarrow{m \to \infty} \boldsymbol{K}$, we have $\hat{\boldsymbol{K}}_t \xrightarrow{m \to \infty} \boldsymbol{K}$. Thus, the equivalence to the kernel regression is established. Note that Theorem 3 is an extension from the corresponding theorem of NNs with residual connection [Tirer et al., 2020] to PNNs. This property allows us to characterize the training process as kernel regression.

Regarding the proof of Theorem 3, we firstly introduce the norm control of the Gaussian weight matrices and then derive the local boundness and local Lipschitzness. The last step is to apply the induction rules over different time steps. Details are presented in the Appendix B.3.

## 3.2 Extrapolation behavior

Firstly, we provide the definition of extrapolation from Xu et al. [2021] as follows.

**Definition 1.** Extrapolation occurs when the domain of test samples is larger than the support of the training distribution.

**Remark:** The definition presented above is different from the one in Balestriero et al. [2021] that claims extrapolation occurs when the test samples fall outside of the convex hull of the training set. Even though these definitions are not completely compatible, both definitions are suitable for our subsequent analysis.

The derived kernel in the previous section enables us to study how PNNs with ReLU activation trained by gradient descent extrapolates. Note that our theorem can also be extended to the raw PNNs without activation function.

**Theorem 4** ($\gamma$-degree extrapolation of $N$-degree PNNs). *Suppose we train $N$-degree ($N \geq 2$) PNNs $f : \mathbb{R}^d \to \mathbb{R}$ with infinite-width on $\{(\boldsymbol{x}_i, y_i)\}_{i=1}^{|\mathcal{X}|}$, and the network is optimized with the squared loss in the NTK regime. For any direction $\boldsymbol{v} \in \mathbb{R}^d$ that satisfies $\|\boldsymbol{v}\|_2 = \max\{\|\boldsymbol{x}_i\|^2\}$, let $\boldsymbol{x}_0 = t\boldsymbol{v}$ and $\boldsymbol{x} = \boldsymbol{x}_0 + h\boldsymbol{v}$ with $t > 1$ and $h > 0$ be the extrapolation data points, the output $f(\boldsymbol{x}_0 + h\boldsymbol{v})$ follows a $\gamma$-degree ($\gamma \leq N$) function with respect to $h$.*

Apart from PNNs, we also consummate Xu et al. [2021] that consider the extrapolation of fully-connected NNs with only two-layer. We provide the following generalized theorem for $N$-layer ($N > 2$) fully-connected NNs.

**Theorem 5** (Linear extrapolation of $N$-layer fully-connected NNs). *Suppose we train $N$-layer ($N \geq 2$) fully-connected NNs $f : \mathbb{R}^d \to \mathbb{R}$ on $\{(\boldsymbol{x}_i, y_i)\}_{i=1}^{|\mathcal{X}|}$. For any direction $\boldsymbol{v} \in \mathbb{R}^d$ that satisfies $\|\boldsymbol{v}\|_2 = \max\{\|\boldsymbol{x}_i\|^2\}$, $\boldsymbol{x}_0 = t\boldsymbol{v}$ and $\boldsymbol{x} = \boldsymbol{x}_0 + h\boldsymbol{v}$ with $t > 1$ and $h > 0$ are extrapolation data points, the output $f(\boldsymbol{x}_0 + h\boldsymbol{v})$ follows a linear function with respect to $h$.*

We have already shown that PNNs extrapolate to a function with specific degree and are more flexible than fully-connected NNs. However, only knowing the information of the degree of the extrapolation function is not enough. Naturally, we might ask under which condition PNNs can achieve successful extrapolation. Below, we build our analysis in the NTK regime and show how the geometry of the training set affects the behavior of PNNs.

**Theorem 6** (Condition for exact extrapolation of PNNs). *Let $f_\rho(\boldsymbol{x}) = \boldsymbol{x}^\top \boldsymbol{\beta} \boldsymbol{x}$ be the target function with $\boldsymbol{x} \in \mathbb{R}^d$ and $\boldsymbol{\beta} \in \mathbb{R}^{d \times d}$. Suppose that $\{\boldsymbol{x}_i\}_{i=1}^{|\mathcal{X}|}$ contains the orthogonal basis $\{\boldsymbol{e}_i\}_{i=1}^d$ and $\{-\boldsymbol{e}_i\}_{i=1}^d$. Then if we train two-degree PNNs $f$ on $\{(\boldsymbol{x}_i, f_\rho(\boldsymbol{x}_i))\}_{i=1}^{|\mathcal{X}|}$ with the squared loss in the NTK regime, we have $f(\boldsymbol{x}) = \boldsymbol{x}^\top \boldsymbol{\beta} \boldsymbol{x}$ for all $\boldsymbol{x} \in \mathbb{R}^d$.*

**Remark:** This result only considers quadratic functions as our proof heavily relies on the construction of the feature map of the NTK, which is harder for the high-degree case.

Due to constrained space, the proof of aforementioned theorems can be found in Appendix C.1 to C.3.

## 3.3  Spectral analysis

In this section, we characterize the approximation properties of $N$-degree PNNs in the in-distribution regime. By studying the spectral analysis in the form of a Mercer decomposition, we explicitly show the eigenvalues and eigenfunctions of NTK. We firstly introduce some notation. Denote by $\{Y_{k,j}\}_{j=1}^{N(d,k)}$ the spherical harmonics of degree $k$ in $d+1$ variables. $G_k^{(\gamma)}$ represents the Gegenbauer polynomials with respect to the weight function $x \mapsto (1-x^2)^{\gamma-\frac{1}{2}}$ and degree $k$. Finally, denote by $F(d,k) := \frac{2k+d-1}{k}\binom{k+d-2}{d-1}$.

The following lemma enables us to connect spherical harmonics to Gegenbauer polynomials.

**Lemma 1.** *[Frye and Efthimiou, 2012, Theorem 4.11] For any $\boldsymbol{x}, \boldsymbol{x}' \in \mathbb{S}^d$, the $k$-degree spherical harmonics in $d+1$ variables satisfies:*

$$\sum_{j=1}^{F(d,k)} Y_{k,j}(\boldsymbol{x})Y_{k,j}(\boldsymbol{x}') = F(d,k)G_k^{(\frac{d-1}{2})}(\langle \boldsymbol{x}, \boldsymbol{x}' \rangle).$$

For any dot product Mercer kernel $K'$, denote by $(\mu_k)_{k=0}^{\infty}$ the eigenvalues associated to the kernel, we can apply the following Mercer's decomposition in the form of spherical harmonics, and using Lemma 1 we obtain:

$$K'(\boldsymbol{x}, \boldsymbol{x}') = \sum_{k=0}^{\infty} \mu_k \sum_{j=1}^{F(d,k)} Y_{k,j}(\boldsymbol{x})Y_{k,j}(\boldsymbol{x}') = \sum_{k=0}^{\infty} \mu_k F(d,k)G_k^{(\frac{d-1}{2})}(\langle \boldsymbol{x}, \boldsymbol{x}' \rangle), \tag{5}$$

In order to study the decay rate of the eigenvalues, we can express the NTK as the product of multiple kernels and present the decay rate of the eigenvalues of PNNs.

**Theorem 7.** *Consider PNNs with $N$-degree ($N \geq 2$) multiplicative interactions and denote by $(\mu_k)_{k=0}^{\infty}$ the eigenvalues associated to the NTK. Then for $k \gg d$, we have $\mu_k = \Omega((N^2k)^{-d/2})$.*

The proof can be found in Appendix D. As a comparison, the decay rate for both deep fully-connected NNs and residual NNs is $\Omega((k)^{-d})$ [Belfer et al., 2021]. Thus, we can see a slower decay rate when inserting Hadamard product into standard NNs, which leads to a faster learning towards high-frequency functions.

## 4  Experiments

Our experiments are organized as follows: We firstly showcase the extrapolation of NNs-Hp in learning some common functions in Section 4.1. Next, we assess the extrapolation performance on non-synthetic dataset in Section 4.2 and conduct the experiment in learning spherical harmonics in Section 4.3. Due to the constrained space, the extrapolation in a visual analogy task and the spectral bias in image classification task are deferred to Appendix E.5 and Appendix E.6, respectively.

### 4.1  Extrapolation in learning analytically-known functions

These experiments aim to examine the extrapolation behavior of NNs-Hp in regression tasks. Our first experiment includes training the networks via the squared loss to fit several well-known and analytically-known underlying functions. During prediction, we sample data points beyond the training regime and observe the extrapolation performance. More details on implementation can be found in Appendix E.1. We set the target function as $f_\rho(x) = x^3 + x^2 - 10x + 5$ and use four-layer fully-connected NN. As presented in Figure 1(a) and Figure 1(b), fully-connected NN extrapolates linearly while NN-Hp approximates better the extrapolation part of the underlying non-linear function, which are consistent with Theorem 4 and Theorem 5.

Learning $f_\rho(x) = \cos(2x)$. We choose eleven-layer fully-connected NN. The training set and testing set are the same as in the previous experiment. Observing Figure 1(c) and Figure 1(d), we find that NN-Hp is more flexible to learn the non-linear function outside the training region while fully-connected NN still extrapolates linearly.

Learning $f_\rho(\boldsymbol{x}) = (x^{(1)})^2 + (x^{(2)})^2$, where $x^{(1)}$ and $x^{(2)}$ is the first and second dimension of $\boldsymbol{x} \in R^2$. In this task, we choose three-layer NNs. Each model is trained with different data distribution, i.e.,

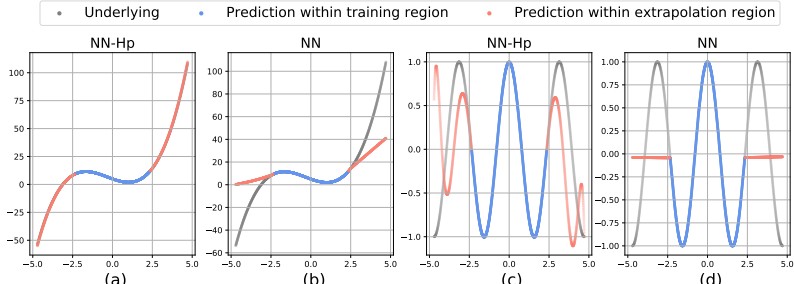

Figure 1: **Extrapolation function.** The blue curve indicates the training regime while the pink color symbolizes the extrapolation regime. (a) and (b) show the fitting results towards $f_\rho(x) = x^3 + x^2 - 10x + 5$. We can see that NN extrapolates linearly without the Hadamard product (Hp) while NN-Hp is able to extrapolate to the underlying non-linear function nearly. (c) and (d) present the fitting results towards $f_\rho(x) = \cos(2x)$. Notably, NN-Hp is more flexible to learn the non-linear function outsides the training region.

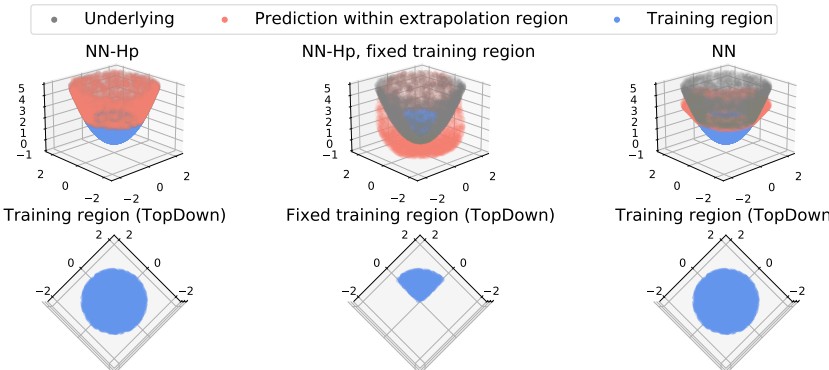

Figure 2: Fitting results for the underlying function $f_\rho(\boldsymbol{x}) = \left(x^{(1)}\right)^2 + \left(x^{(2)}\right)^2$, where $x^{(1)}$ and $x^{(2)}$ are the first and second dimension of $\boldsymbol{x} \in R^2$. Blue points indicate the training regime, pink points symbolize the extrapolation regime, gray points indicate the underlying function. **Left**: We train NNs-Hp with the training set containing support in all directions, the network is able to extrapolate successfully. **Middle**: We train NN-Hp with the training set wherein two dimensions of the data are fixed to be positive , NN-Hp fails to extrapolate. **Right**. We remove the Hadamard product of the network, which leads to linear extrapolation.

the training set contains support in all directions in the first case while two dimension of the training set are fixed to be positive in the second case. The result is visually depicted in Figure 2, which shows that NNs-Hp can achieve exact extrapolation if the training set contains support in all directions and thereby validates Theorem 6. On the other hand, NNs without the Hadamard product fail to extrapolate to the underlying function due to its linear extrapolation.

## 4.2 Extrapolation in real-world dataset

In this section, we assess the extrapolation performance beyond synthetic datasets.

**Variation of brightness.** This experiment is conducted on two well-known grayscale image datasets: MNIST dataset [LeCun et al., 1998] and Fashion-MNIST dataset Xiao et al. [2017]. For these two datasets, the original range of the pixel of each image is $[0, 1]$, we divided it by 10 for the raw training set to construct the new one where the pixels range from 0 to 0.1. During extrapolation, we limit the range of the original testing set to $[0, r_{\max}]$ through division, where $r_{\max} \in \{0.1, 0.2, 0.3, ..., 1.0\}$, as illustrated in the top two panels in Figure 3. Then we feed these images into the trained network and evaluate the accuracy. More details on the implementation can be found in Appendix E.2. The accuracy is summarized in the two bottom plots of Figure 3. We find that both networks achieve

similar accuracy in the case $r_{\max} = 0.1$ while inserting Hadamard product (Hp) into NN improves the performance during extrapolation.

**Arithmetic extrapolation.** Now we turn to a more challenging task. As human we can usually extrapolate to arbitrarily large numbers in arithmetic. How do the neural networks perform during extrapolation? Following the setup of Bloice et al. [2020], we use MNIST dataset, where there are 100 different two-image combinations of the digits $0 \sim 9$. We randomly pick up 90 combinations as the training set and the remaining 10 combinations as the extrapolation set. This problem is treated as regression instead of classification for higher error tolerance following Bloice et al. [2020]. In addition, if we design the network as a classifier, the number of the class will vary as the change of the splitting for the training set and testing set. The network only outputs one single discrete value. However, we still measure the accuracy by rounding the network output. five-layer fully-connected NNs and convolution NNs are chosen as the baselines. For comparison, we implement NN-Hp with dense layers and NN-Hp with convolution layers, respectively. More details on the implementation can be found in the Appendix E.3. The results obtained by a three-fold cross validation are summarized in Table 1, where we can see NN-Hp has a better extrapolation behavior in such more difficult task.

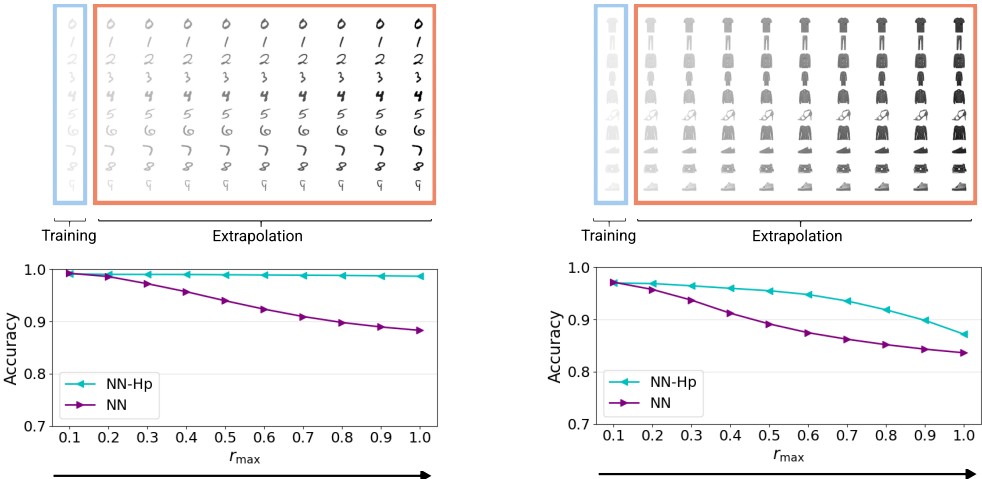

(a) Examples and results on MNIST dataset.     (b) Examples and results on Fashion-MNIST dataset.

Figure 3: The top two panels show the examples of extrapolation in MNIST dataset and Fashion-MNIST dataset. $r_{\max}$ varies from 0.1 to 1.0. from left to right, indicating the variation of the darkness of the image. The bottom two panels show the accuracy as $r_{\max}$ increasing. Both networks achieve similar accuracy in the case $r_{\max} = 0.1$ while inserting Hadamard product (Hp) into NN improves the performance during extrapolation.

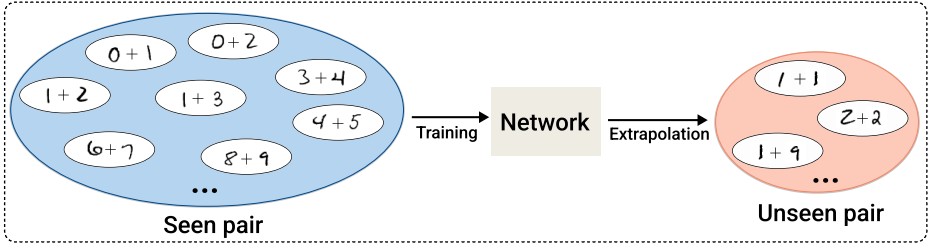

Figure 4: A schematic illustration for the task of arithmetic extrapolation.

## 4.3 Spectral bias in learning spherical harmonics

Here, we aim to learn the linear combinations of spherical Harmonics where inputs are sampled from the uniform distribution on the unit sphere. Our experiment follows the setup in Choraria et al. [2022], which only considers NNs-Hp with one Hadamard product. The target function is

Table 1: Results in the task of arithmetic extrapolation, which aims to predict the target label with regression. 'Interpolation' indicates the accuracy in the seen pairs during training while 'Extrapolation' indicates the accuracy tested in those unseen pairs. Three ways are used for the network output: (a) Rounding, the output is rounded to the nearest integer. (b) Floor/ceiling, A floor and ceiling function is applied for the output and if one of those equals to the ground truth label, we treat it as a correct prediction. (c) ±1. An error of ±1 is allowed. We can observe that NN-Hp has a better extrapolation behavior compared with the baselines.

| Method | | Rounding | Floor/ceiling | ±1 |
|---|---|---|---|---|
| NN(Dense) | Interpolation | $0.980 \pm 0.002$ | $0.999 \pm 0.000$ | $0.999 \pm 0.000$ |
| | Extrapolation | $0.436 \pm 0.065$ | $\mathbf{0.805 \pm 0.042}$ | $0.887 \pm 0.011$ |
| NN-Hp (Dense) | Interpolation | $0.926 \pm 0.031$ | $0.996 \pm 0.001$ | $0.999 \pm 0.000$ |
| | Extrapolation | $\mathbf{0.554 \pm 0.011}$ | $0.802 \pm 0.010$ | $\mathbf{0.889 \pm 0.008}$ |
| | | | | |
| NN(Conv) | Interpolation | $0.945 \pm 0.983$ | $0.983 \pm 0.021$ | $0.994 \pm 0.007$ |
| | Extrapolation | $0.617 \pm 0.103$ | $0.918 \pm 0.016$ | $0.953 \pm 0.006$ |
| NN-Hp (Conv) | Interpolation | $0.991 \pm 0.002$ | $0.998 \pm 0.000$ | $0.999 \pm 0.000$ |
| | Extrapolation | $\mathbf{0.825 \pm 0.109}$ | $\mathbf{0.948 \pm 0.006}$ | $\mathbf{0.963 \pm 0.007}$ |

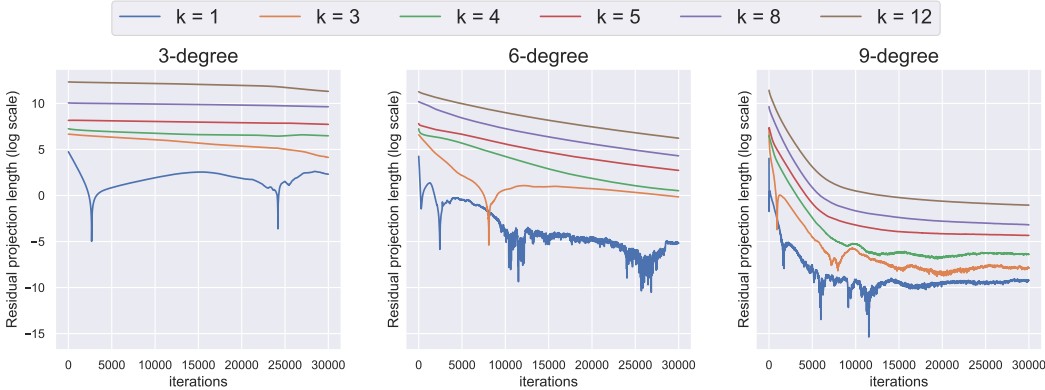

Figure 5: Comparison of convergence curve of error projection lengths for NNs-Hp with $N$-degree multiplicative interactions, where $N \in \{3, 6, 9\}$ for different order harmonics with $K \in \{1, 3, 4, 5, 8, 12\}$ We can see the improvement for high-degree interactions in the rate of convergence of error.

defined by: $f^*(\boldsymbol{x}) = \frac{1}{N(\mathcal{K})} \sum_{k \in K} A_k P_k(\langle \boldsymbol{x}, \zeta_k \rangle)$, $\mathcal{K} = \{1, 3, 4, 5, 8, 12\}$, where $P_k(t)$ denotes the $k$-degree Gegenbauer polynomial, $\zeta_k$ are fixed vectors that are independently sampled from uniform distribution on unit sphere, and $N(\mathcal{K})$ is the normalizing constant. The error residuals with different $\mathcal{K}$ are compared during the training process. Implementation details are deferred to Appendix E.4. In this experiment, we show that increasing the number of multiplicative interactions can improve the rate of convergence of error, as presented in Figure 5.

## 5 Conclusion

This paper examines neural network with Hadamard product with a particular focus on polynomial neural networks from a theoretical perspective. The analysis of the NTK paves the way for knowing interesting properties of the networks, such as the extrapolation behavior and the spectral bias. Experimental results in learning analytically-known functions validate our hypothesis. We further conduct several experiments in real-world datasets and demonstrate the advantage of inserting Hadamard products into standard neural networks. We believe not only our framework provides a good justification for a deeper understanding of neural architecture, but it also lays the foundations to investigate other more complicated OOD problems such as domain adaption and invariant learning in future work.

## Acknowledgements

We are thankful to the reviewers for providing constructive feedback. This project has received support from the European Research Council (ERC) under the European Union's Horizon 2020 research and innovation programme (grant agreement number 725594 - timedata). This work was sponsored by the Army Research Office and was accomplished under Grant Number W911NF-19-1-0404. This work was supported by Zeiss. This work has received funding from SNF project – Deep Optimisation of the Swiss National Science Foundation (SNSF) under grant number 200021_205011. This work was supported by Hasler Foundation Program: Hasler Responsible AI (project number 21043).

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
