## Contents of the Appendix

The Appendix is organized as follows:

## A    Theoretical background

Our analysis in the main paper is built on a specific family of neural network with Hadamard product, called Π-Nets. In Appendix A.1, we will overview the theoretical background of Π-Nets. Next, in Appendix A.2, we briefly introduce the background on neural tangent kernel (NTK), which is used for analyzing the networks. We start with introducing some notation. The *mode-$m$* vector product of a $M^{\text{th}}$ order tensor $\boldsymbol{\mathcal{A}} \in \mathbb{R}^{J_1 \times J_2 \times \cdots \times J_{m-1} \times J_m \times J_{m+1} \times \cdots \times J_M}$ and a vector $\boldsymbol{x} \in \mathbb{R}^{J_m}$ is denoted by $\boldsymbol{\mathcal{A}} \times_m \boldsymbol{x} \in \mathbb{R}^{J_1 \times J_2 \times \cdots \times J_{m-1} \times J_{m+1} \times \cdots \times J_M}$, resulting in $(M-1)^{\text{th}}$ order tensor:

$$(\boldsymbol{\mathcal{A}} \times_m \boldsymbol{x})_{j_1,\ldots,j_{m-1},j_{m+1},\ldots,j_M} = \sum_{j_m=1}^{J_m} a_{j_1,j_2,\ldots,j_M} z_{j_m} \,.$$

The *mode-$m$* vector product of a tensor and multiple vectors is denoted as:

$$\boldsymbol{\mathcal{A}} \times_1 \boldsymbol{x}^{(1)} \times_2 \boldsymbol{x}^{(2)} \times_3 \cdots \times_M \boldsymbol{x}^{(M)} = \boldsymbol{\mathcal{A}} \prod_{m=1}^{M} \times_m \boldsymbol{x}^{(m)} \,.$$

In *CANDECOMP/PARAFAC (CP) decomposition* [Kolda and Bader, 2009], the tensor is decomposed into a sum of component rank-one tensors. The rank-$R$ CP decomposition of an $M^{\text{th}}$ order tensor $\boldsymbol{\mathcal{A}}$ is symbolized by

$$\boldsymbol{\mathcal{A}} = \sum_{r=1}^{R} \boldsymbol{x}_r^{(1)} \circ \boldsymbol{x}_r^{(2)} \circ \cdots \circ \boldsymbol{x}_r^{(M)} \,, \tag{6}$$

where $\circ$ is the outer product of vectors.

Table 2: Core symbols

| Symbol | Dimension(s) | Definition |
|---|---|---|
| $\sigma(\cdot), \dot{\sigma}(\cdot)$ | - | ReLU function and its derivative |
| $\odot, *$ | - | Khatri-Rao product, Hadamard product |
| $\boldsymbol{e}_j$ | $\mathbb{R}^m$ | $j^{th}$ canonical basis vector of $\mathbb{R}^m$ |
| $n, N$ | $\mathbb{N}$ | Polynomial term degree and total degree |
| $\boldsymbol{x}$ | $\mathbb{R}^d$ | Input to the network |
| $f(\boldsymbol{x})$ | $\mathbb{R}$ | Output of the network |
| $\boldsymbol{\mathcal{A}}^{[n]}$ | $\mathbb{R}^{1 \times \prod_{i=1}^{n} \times_i d}$ | Parameter tensor of the polynomial |
| $\boldsymbol{b}, \boldsymbol{W}_n, \boldsymbol{W}_{N+1}$ | $\mathbb{R}, \mathbb{R}^{m \times d}, \mathbb{R}^{1 \times m}$ | Learnable parameters |
| $\ell_2(\boldsymbol{\theta}) = \frac{1}{2} \sum_{(\boldsymbol{x}_i, y_i) \in (\mathcal{X}, \mathcal{Y})} (f(\boldsymbol{x}_i; \boldsymbol{\theta}) - y_i)^2$ | $\mathbb{R}$ | Empirical training loss |
| $\{\boldsymbol{x}_i\}_{i=1}^{|\mathcal{X}|}, \{y_i\}_{i=1}^{|\mathcal{X}|}$ | | Features and labels of training set $(\mathcal{X}, \mathcal{Y})$ |

### A.1 A Primer on polynomial nets

The goal of $\Pi$-Nets is to learn an $N$-degree polynomial expansion that outputs $f(\boldsymbol{x}) \in \mathbb{R}^d$ with respect to the input $\boldsymbol{x} \in \mathbb{R}^d$:

$$f(\boldsymbol{x}) = \sum_{n=1}^{N} \left( \boldsymbol{\mathcal{A}}^{[n]} \prod_{j=2}^{n+1} \times_j \boldsymbol{x} \right) + \boldsymbol{b}, \tag{7}$$

where $\left\{ \boldsymbol{\mathcal{A}}^{[n]} \in \mathbb{R}^{1 \times \prod_{i=1}^{n} \times_i d} \right\}_{n=1}^{N}$ and $\boldsymbol{b} \in \mathbb{R}$ are learnable parameters. Nevertheless, as the degree of the polynomial increases, the number of parameters in Eq. (7) grows exponentially. In order to improve the scalability, a coupled CP decomposition (CCP) with factor sharing is used to reduce parameters [Kolda and Bader, 2009, Chrysos et al., 2021b]. With CCP, all the weight tensors $\{\boldsymbol{\mathcal{A}}^{[n]}\}_{n=1}^{N}$ are jointly factorized by a coupled CP decomposition where the factors between different degrees are shared. For instance, the parameters of the third degree expansion follows:

- First degree parameters: $\boldsymbol{A}^{[1]} = \boldsymbol{W}_4 \boldsymbol{W}_1$.
- Second degree parameters: $\boldsymbol{A}^{[2]}_{(1)} = \boldsymbol{W}_4(\boldsymbol{W}_3 \odot \boldsymbol{W}_1) + \boldsymbol{W}_4(\boldsymbol{W}_2 \odot \boldsymbol{W}_1)$.
- Third degree parameters: $\boldsymbol{A}^{[3]}_{(1)} = \boldsymbol{W}_4(\boldsymbol{W}_3 \odot \boldsymbol{W}_2 \odot \boldsymbol{W}_1)$.

Combining the aforementioned factorizations, the third degree expansion of Eq. (7) can be expressed as:

$$f(\boldsymbol{x}) = \boldsymbol{b} + \boldsymbol{W}_4 \boldsymbol{W}_1 \boldsymbol{x} + \boldsymbol{W}_4 \Big( \boldsymbol{W}_3 \odot \boldsymbol{W}_1 \Big)(\boldsymbol{x} \odot \boldsymbol{x}) + \boldsymbol{W}_4 \Big( \boldsymbol{W}_2 \odot \boldsymbol{W}_1 \Big)(\boldsymbol{x} \odot \boldsymbol{x}) + \\ \boldsymbol{W}_4 \Big( \boldsymbol{W}_3 \odot \boldsymbol{W}_2 \odot \boldsymbol{W}_1 \Big)(\boldsymbol{x} \odot \boldsymbol{x} \odot \boldsymbol{x}). \tag{8}$$

Next, we introduce the following lemma used to convert the Khatri-Rao products into a Hadamard product.

**Lemma 2.** *[Chrysos et al., 2019] Given two sets of real-valued matrices $\{\boldsymbol{A}_\nu \in \mathbb{R}^{I_\nu \times K}\}_{\nu=1}^{N}$ and $\{\boldsymbol{B}_\nu \in \mathbb{R}^{I_\nu \times L}\}_{\nu=1}^{N}$, the following equality holds:*

$$\left( \bigodot_{\nu=1}^{N} \boldsymbol{A}_\nu \right)^{\top} \cdot \left( \bigodot_{\nu=1}^{N} \boldsymbol{B}_\nu \right) = (\boldsymbol{A}_1^{\top} \cdot \boldsymbol{B}_1) * \ldots * (\boldsymbol{A}_N^{\top} \cdot \boldsymbol{B}_N). \tag{9}$$

Applying the above lemma on Eq. (8), we obtain:

$$f(\boldsymbol{x}) = \boldsymbol{b} + \boldsymbol{W}_4 \{ (\boldsymbol{W}_3 \boldsymbol{x}) * [(\boldsymbol{W}_2 \boldsymbol{x}) * (\boldsymbol{W}_1 \boldsymbol{x}) + \\ \boldsymbol{W}_1 \boldsymbol{x}] + (\boldsymbol{W}_2 \boldsymbol{x}) * (\boldsymbol{W}_1 \boldsymbol{x}) + \boldsymbol{W}_1 \boldsymbol{x} \}, \tag{10}$$

which can be further converted to the following recursive relationship and generalized to arbitrary degree:

$$\boldsymbol{y}_n = (\boldsymbol{W}_n \boldsymbol{x}) * \boldsymbol{y}_{n-1} + \boldsymbol{y}_{n-1}, \quad n = 2, \ldots, N \\ \boldsymbol{y}_1 = \boldsymbol{W}_1 \boldsymbol{x}, \qquad f(\boldsymbol{x}) = \boldsymbol{W}_{N+1} \boldsymbol{y}_N + \boldsymbol{b}. \tag{11}$$

The parameters $\boldsymbol{b} \in \mathbb{R}$, $\boldsymbol{W}_{N+1} \in \mathbb{R}^{1 \times m}$, $\boldsymbol{W}_n \in \mathbb{R}^{d \times m}$ for $n = 1, \ldots, N$, are learnable. To simplify the proof, we follow Zhu et al. [2022] to reparameterize Eq. (11) and obtain Eq. (1) (in the main body). Apart from CCP, we can also factorize the polynomial networks by other decompositions. The recursive formula of NCP is:

$$\boldsymbol{y}_n = (\boldsymbol{W}_n \boldsymbol{x}) * (\boldsymbol{F}_n^T \boldsymbol{y}_{n-1} + \boldsymbol{B}_n^T \boldsymbol{b}_n), \quad n = 2, \ldots, N \\ \boldsymbol{y}_1 = (\boldsymbol{W}_1 \boldsymbol{x}) * (\boldsymbol{B}_1^T \boldsymbol{b}_1), \quad f(\boldsymbol{x}) = \boldsymbol{W}_{N+1} \boldsymbol{y}_N + \boldsymbol{b}, \tag{12}$$

where the parameters $\boldsymbol{b}, \boldsymbol{W}_{N+1}, \boldsymbol{W}_n, \boldsymbol{B}_n, \boldsymbol{b}_n, \boldsymbol{F}_n$ are learnable. The recursive formula of NCP-skip is:

$$\boldsymbol{y}_n = (\boldsymbol{W}_n \boldsymbol{x}) * (\boldsymbol{F}_n^T \boldsymbol{y}_{n-1} + \boldsymbol{B}_n^T \boldsymbol{b}_n) + \boldsymbol{D}_n \boldsymbol{y}_{n-1}, \quad n = 2, \ldots, N \\ \boldsymbol{y}_1 = (\boldsymbol{W}_1 \boldsymbol{x}) * (\boldsymbol{B}_1^T \boldsymbol{b}_1), \quad f(\boldsymbol{x}) = \boldsymbol{W}_{N+1} \boldsymbol{y}_N + \boldsymbol{b}, \tag{13}$$

where the parameters $\boldsymbol{b}, \boldsymbol{W}_{N+1}, \boldsymbol{W}_n, \boldsymbol{B}_n, \boldsymbol{b}_n, \boldsymbol{F}_n, \boldsymbol{D}_n$ are learnable.

## A.2 A Primer on NTK

In this section, we summarize how training a neural network by minimizing squared loss, i.e., $\ell_2(\boldsymbol{\theta}_t) = \frac{1}{2}\sum_{(\boldsymbol{x}_i,y_i)\in(\mathcal{X},\mathcal{Y})}(f(\boldsymbol{x}_i;\boldsymbol{\theta}_t)-y_i)^2$, via gradient descent can be characterized by the kernel regression predictor with NTK.

By choosing an infinitesimally small learning rate, we can obtain the following gradient flow:

$$\frac{d\boldsymbol{\theta}_t}{dt} = -\nabla\ell_2(\boldsymbol{\theta}_t).$$

By substituting the loss into the above equation and using the chain rule, we can find that the network outputs $f(\boldsymbol{\theta}_t) = \mathrm{vec}(\{f(\boldsymbol{x}_i;\boldsymbol{\theta}_t)\}_{\boldsymbol{x}_i\in\mathcal{X}}) \in \mathbb{R}^{|\mathcal{X}|}$ admit the following dynamics:

$$\frac{df(\boldsymbol{\theta}_t)}{dt} = -\hat{\boldsymbol{K}}_t(f(\boldsymbol{\theta}_t)-\boldsymbol{y}), \tag{14}$$

where $\boldsymbol{y} = \mathrm{vec}(\{y_i\}_{y_i\in\mathcal{Y}}) \in \mathbb{R}^{|\mathcal{X}|}$, $\hat{\boldsymbol{K}}_t = J(\boldsymbol{\theta}_t)J(\boldsymbol{\theta}_t)^\top = \left(\frac{\partial f(\boldsymbol{\theta}_t)}{\partial\boldsymbol{\theta}}\right)\left(\frac{\partial f(\boldsymbol{\theta}_t)}{\partial\boldsymbol{\theta}}\right)^\top \in \mathbb{R}^{|\mathcal{X}|\times|\mathcal{X}|}$. Jacot et al. [2018], Arora et al. [2019] have shown that for fully-connected neural networks, under the infinite-width setting and proper initialization, $\hat{\boldsymbol{K}}_t$ will keep constant during training and $\hat{\boldsymbol{K}}_0$ will converge to a fixed matrix $\boldsymbol{K} \in \mathbb{R}^{|\mathcal{X}|\times|\mathcal{X}|}$, where $K_{ij} = K(\boldsymbol{x}_i,\boldsymbol{x}_j)$ is the NTK value for the inputs $\boldsymbol{x}_i$ and $\boldsymbol{x}_j$. Then, based on $\hat{\boldsymbol{K}}_t = \hat{\boldsymbol{K}}_0 = \boldsymbol{K}$, we can rewrite Eq. (14) as:

$$\frac{df(\boldsymbol{\theta}_t)}{dt} = -\boldsymbol{K}(f(\boldsymbol{\theta}_t)-\boldsymbol{y}). \tag{15}$$

This implies the network output for any $\boldsymbol{x} \in \mathbb{R}^d$ can be calculated by the kernel regression predictor with the associated NTK:

$$f(\boldsymbol{x}) = \big(K(\boldsymbol{x},\boldsymbol{x}_1),\cdots,K(\boldsymbol{x},\boldsymbol{x}_{|\mathcal{X}|})\big)\cdot\boldsymbol{K}^{-1}\boldsymbol{y},$$

where $K(\boldsymbol{x},\boldsymbol{x}_i)$ is the kernel value between test data $\boldsymbol{x}$ and training data $\boldsymbol{x}_i$.

# B Proofs of NTK

We derive the NTK of NNs-Hp with multiple multiplicative interactions in Appendix B.1. Next, we prove the width requirement for the empirical kernel to converge to NTK at initialization in Appendix B.2. Lastly, we analyze the training dynamics of NNs-Hp under gradient descent in Appendix B.3.

## B.1 Proof of Theorem 1

Recall that the NTK is defined as the limit of the following inner product:

$$\boldsymbol{K}(\boldsymbol{x},\boldsymbol{x}') = \lim_{m\to\infty}\langle\nabla_{\boldsymbol{\theta}}f_{\boldsymbol{\theta}}(\boldsymbol{x}),\nabla_{\boldsymbol{\theta}}f_{\boldsymbol{\theta}}(\boldsymbol{x}')\rangle,$$

where $\boldsymbol{\theta}$ represents all the parameters. Observing Eq. (1), we can compute the gradient with respect to each weight and then sum up the inner products to obtain the NTK.

Below, we denote by $\tilde{\boldsymbol{\alpha}}_n = \boldsymbol{W}_n\boldsymbol{x}$, $n \in [N]$ the pre-activation vectors and $\boldsymbol{\alpha}_n$ the vectors after applying the element-wise ReLU activation to $\tilde{\boldsymbol{\alpha}}_n$.

Firstly, we compute the contribution to the NTK w.r.t $\boldsymbol{W}_1$, its corresponding derivative is as follows:

$$\partial_{\boldsymbol{W}_1}f(\boldsymbol{x}) = \sqrt{\frac{2}{m}}\sum_{j=1}^m W_{N+1}^{(j)}\left(\prod_{n=2}^N\sigma\left(\sqrt{\frac{2}{m}}\tilde{\alpha}_n^j(\boldsymbol{x})\right)\right)\dot{\sigma}\left(\sqrt{\frac{2}{m}}\tilde{\alpha}_1^{(j)}(\boldsymbol{x})\right)\partial_{\boldsymbol{W}_1}\tilde{\alpha}_1^{(j)}(\boldsymbol{x})$$

$$= \sqrt{\frac{2}{m}}\sum_{j=1}^m W_{N+1}^{(j)}\left(\prod_{n=2}^N\sigma\left(\sqrt{\frac{2}{m}}\tilde{\alpha}_n^{(j)}(\boldsymbol{x})\right)\right)\dot{\sigma}\left(\sqrt{\frac{2}{m}}\tilde{\alpha}_1^{(j)}(\boldsymbol{x})\right)\partial_{\boldsymbol{W}_1}\left(\boldsymbol{e}_j^\top\boldsymbol{W}_N\boldsymbol{x}\right)$$

$$= \sqrt{\frac{2}{m}}\sum_{j=1}^m W_{N+1}^{(j)}\left(\prod_{n=2}^N\sigma\left(\sqrt{\frac{2}{m}}\tilde{\alpha}_n^{(j)}(\boldsymbol{x})\right)\right)\dot{\sigma}\left(\sqrt{\frac{2}{m}}\tilde{\alpha}_1^{(j)}(\boldsymbol{x})\right)\left(\boldsymbol{e}_j\boldsymbol{x}^\top\right).$$

The inner product of the derivative is:

$$\langle \partial_{\boldsymbol{W}_1} f(\boldsymbol{x}), \partial_{\boldsymbol{W}_1} f(\boldsymbol{x}') \rangle$$

$$= \frac{2}{m} \sum_{j,k=1}^{m} W_{N+1}^{(j)} W_{N+1}^{(k)} \left( \prod_{n=2}^{N} \left( \frac{2}{m} \sigma\left(\widetilde{\alpha}_n^{(j)}(\boldsymbol{x})\right) \sigma\left(\widetilde{\alpha}_n^{(j)}(\boldsymbol{x}')\right) \right) \right) \left( \frac{2}{m} \dot{\sigma}\left(\widetilde{\alpha}_1^{(j)}(\boldsymbol{x})\right) \dot{\sigma}\left(\widetilde{\alpha}_1^{(k)}(\boldsymbol{x}')\right) \right) \langle \boldsymbol{e}_j \boldsymbol{x}^\top, \boldsymbol{e}_k \boldsymbol{x}'^T \rangle$$

$$= \frac{2}{m} \sum_{j,k=1}^{m} W_{N+1}^{(j)} W_{N+1}^{(k)} \left( \prod_{n=2}^{N} \left( \frac{2}{m} \sigma\left(\widetilde{\alpha}_n^{(j)}(\boldsymbol{x})\right) \sigma\left(\widetilde{\alpha}_n^{(k)}(\boldsymbol{x}')\right) \right) \right) \left( \frac{2}{m} \dot{\sigma}\left(\widetilde{\alpha}_1^{(j)}(\boldsymbol{x})\right) \dot{\sigma}\left(\widetilde{\alpha}_1^{(k)}(\boldsymbol{x}')\right) \right) \boldsymbol{x}^\top \boldsymbol{x}' \delta_{jk}$$

$$= \frac{2}{m} \sum_{j=1}^{m} W_{N+1}^{(j)} W_{N+1}^{(j)} \left( \prod_{n=2}^{N} \left( \frac{2}{m} \sigma\left(\widetilde{\alpha}_n^{(j)}(\boldsymbol{x})\right) \sigma\left(\widetilde{\alpha}_n^{(j)}(\boldsymbol{x}')\right) \right) \right) \left( \frac{2}{m} \dot{\sigma}\left(\widetilde{\alpha}_1^{(j)}(\boldsymbol{x})\right) \dot{\sigma}\left(\widetilde{\alpha}_1^{(j)}(\boldsymbol{x}')\right) \right) \boldsymbol{x}^\top \boldsymbol{x}'.$$

$$(16)$$

By the law of large numbers and the fact that $\mathbb{E}_{w \sim \mathcal{N}(0,1)} w^2 = 1$, we obtain:

$$\lim_{m \to \infty} \langle \partial_{\boldsymbol{W}_1} f(\boldsymbol{x}), \partial_{\boldsymbol{W}_1} f(\boldsymbol{x}') \rangle = 2\langle \boldsymbol{x}, \boldsymbol{x}' \rangle \kappa_1(\boldsymbol{x}, \boldsymbol{x}')(\kappa_2(\boldsymbol{x}, \boldsymbol{x}'))^{N-1}, \tag{17}$$

where $\kappa_1$ and $\kappa_2$ are defined in Eq. (3). Given that Eq. (1) is symmetric w.r.t $\{\boldsymbol{W}_n\}_{n=1}^{N}$, the contributions of $\{\boldsymbol{W}_n\}_{n=1}^{N}$ to the NTK are the same, we can trivially multiply Eq. (17) by $N$.

Next, we compute the contribution to the NTK w.r.t $\boldsymbol{W}_{N+1}$, its corresponding derivative is as follows:

$$\partial_{\boldsymbol{W}_{N+1}} f(\boldsymbol{x}) = \sqrt{\frac{2}{m}} \left( \sqrt{\frac{2}{m}} \sigma(\widetilde{\boldsymbol{\alpha}}_N) * \ldots * \sqrt{\frac{2}{m}} \sigma(\widetilde{\boldsymbol{\alpha}}_1) \right).$$

The inner product of the derivative is:

$$\langle \partial_{\boldsymbol{W}_{N+1}} f(\boldsymbol{x}), \partial_{\boldsymbol{W}_{N+1}} f(\boldsymbol{x}') \rangle = \frac{2}{m} \sum_{j=1}^{m} \left( \prod_{n=1}^{N} \left( \frac{2}{m} \sigma\left(\widetilde{\alpha}_n^{(j)}(\boldsymbol{x})\right) \sigma\left(\widetilde{\alpha}_n^{(j)}(\boldsymbol{x}')\right) \right) \right).$$

By the law of large numbers:

$$\lim_{m \to \infty} \langle \partial_{\boldsymbol{W}_{N+1}} f(\boldsymbol{x}), \partial_{\boldsymbol{W}_{N+1}} f(\boldsymbol{x}') \rangle$$
$$= 2 \left( \mathbb{E}_{\boldsymbol{w} \sim \mathcal{N}(\boldsymbol{0}, \sqrt{\frac{2}{m}} \cdot \boldsymbol{I})} \left( \sigma(\boldsymbol{w}^\top \boldsymbol{x}) \cdot \sigma(\boldsymbol{w}^\top \boldsymbol{x}') \right) \right)^N$$
$$= 2 \cdot (\kappa_2(\boldsymbol{x}, \boldsymbol{x}'))^N. \tag{18}$$

The proof is completed by multiplying Eq. (17) by $N$ and adding by Eq. (18).

## B.2 Proof of Theorem 2

Before we prove Theorem 2, we need to tackle a technical key issue: how to provide probability estimates for the multiplication of several sub-exponential random variables. To this end, we introduce sub-Weibull random variables in the following definition, which allows for our case and still admits exponential decay tails.

**Definition 2** (Sub-Weibull distributions Zhang and Chen [2020]). Given positive constants $a, b$, a random variable $X$ is sub-Weibull if it satisfies $P(|X| \geq x) \leq ae^{-bx^\theta}$, where $\theta > 0$ is the order.

**Remark:** The classical sub-exponential and sub-Gaussian random variables are sub-Weibull by taking $\theta = 1$ and $\theta = 2$, respectively.

Based on the definition above, we have the following concentration inequality on sub-Weilbull random variables.

**Lemma 3** (Sub-Weibull Concentration Zhang and Chen [2020]). *Given some $\theta > 0$, if $\{X_k\}_{k=1}^{K}$ are independent mean zero random variable such that the sub-Weibull norm $\|X_k\|_{\psi_\theta} < \infty$ for all $1 \leq k \leq K$, then for any weight vector $\boldsymbol{w} = (w_1, \ldots, w_K) \in \mathbb{R}^n$, for any $\zeta \in (0,1)$ one has*

$$P(|\sum_{k=1}^{K} w_k X_k| \geq 2e\rho(\theta)\|\boldsymbol{b}\|_2 \{\sqrt{\zeta} + D(\theta)\zeta^{1/\theta}\}) \leq 2e^{-\zeta},$$

where $\boldsymbol{b} = (w_1 \|X_1\|_{\psi_\theta}, \ldots, w_K \|X_K\|_{\psi_\theta})^\top$, $D(\theta) := \frac{4^{1/\theta}}{\sqrt{2}\|\boldsymbol{b}\|_2} \times \begin{cases} \|\boldsymbol{b}\|_\infty, & \text{if } \theta < 1, \\ 4e\|\boldsymbol{b}\|_{\frac{\theta}{1-\theta}}/C(\theta), & \text{if } \theta \geq 1, \end{cases}$

$$\rho(\theta) := \max\{\sqrt{2}, 2^{1/\theta}\} \times \begin{cases} \sqrt{8}e^3(2\pi)^{1/4}e^{1/24}(e^{2/e}/\theta)^{1/\theta}, & \text{if } \theta < 1, \\ 4e + 2(\log 2)^{1/\theta}, & \text{if } \theta \geq 1. \end{cases}$$

*Proof of Theorem 2.* Recall from Eq. (16):

$$\langle \partial_{\boldsymbol{W}_1} f(\boldsymbol{x}), \partial_{\boldsymbol{W}_1} f(\boldsymbol{x}') \rangle$$
$$= \frac{1}{m} \sum_{k=1}^m \underbrace{2W_{N+1}^{(k)} W_{N+1}^{(k)} \left( \prod_{n=2}^N \left( \frac{2}{m} \sigma\left(\widetilde{\alpha}_n^{(k)}(\boldsymbol{x})\right) \sigma\left(\widetilde{\alpha}_n^{(k)}(\boldsymbol{x}')\right) \right) \right) \left( \frac{2}{m} \dot{\sigma}\left(\widetilde{\alpha}_1^{(k)}(\boldsymbol{x})\right) \dot{\sigma}\left(\widetilde{\alpha}_1^{(k)}(\boldsymbol{x}')\right) \right) \boldsymbol{x}^\top \boldsymbol{x}'}_{:= X_k},$$

where $\tilde{\boldsymbol{\alpha}}_n = \boldsymbol{W}_n \boldsymbol{x}$, $n = 1, \ldots, N$ represent the pre-activation. Firstly, we centralize $X_k$ and denote by $\widehat{X}_k$ as follows:

$$\widehat{X}_k = X_k - 2\kappa_1(\boldsymbol{x}, \boldsymbol{x}')(\kappa_2(\boldsymbol{x}, \boldsymbol{x}'))^{N-1} \boldsymbol{x}^\top \boldsymbol{x}'$$

Since all the weight matrices $\boldsymbol{W}_n$, $n = 1, \ldots, N+1$ are Gaussian, $\sigma\left(\widetilde{\alpha}_n^{(k)}(\boldsymbol{x})\right)$, $\sigma\left(\widetilde{\alpha}_n^{(k)}(\boldsymbol{x}')\right)$, and $\dot{\sigma}\left(\widetilde{\alpha}_1^{(k)}(\boldsymbol{x})\right) \dot{\sigma}\left(\widetilde{\alpha}_1^{(k)}(\boldsymbol{x}')\right)$ are sub-Gaussian random variables over the randomness of $\boldsymbol{W}_n$, Thus, $\{\widehat{X}_k\}_{k=1}^m$ is zero mean sub-Weibull random variable with order $\theta = 2/(2N+1)$. Plugging $w_1, \ldots, w_k = 1/m$ into Lemma 3, we get

$$\rho\left(\frac{2}{2N+1}\right) = \sqrt{2}^{2N-1} \sqrt{8}e^3(2\pi)^{1/4}e^{1/24}\left(e^{2/e}(2N-1)/2\right)^{(2N-1)/2}$$

$$\|\boldsymbol{b}\|_2 = \frac{1}{m}\|(\|\widehat{X}_1\|_{\psi_\theta}, \ldots, \|\widehat{X}_m\|_{\psi_\theta})\|_2, \qquad D\left(\frac{2}{2N+1}\right) = \frac{4^{(2N+1)/2}\|\boldsymbol{b}\|_\infty}{\sqrt{2}\|\boldsymbol{b}\|_2}.$$

Suppose that the width satisfies $m \geq 2^{4N-2} \log^{2N-1}(2N/\delta)$, then for any $\delta \in (0,1)$, with probability at least $1 - (\delta/N)$ over the randomness of initialization, we have

$$\left|\langle \partial_{\boldsymbol{W}_1} f(\boldsymbol{x}), \partial_{\boldsymbol{W}_1} f(\boldsymbol{x}') \rangle - 2\langle \boldsymbol{x}, \boldsymbol{x}' \rangle \kappa_1(\boldsymbol{x}, \boldsymbol{x}')(\kappa_2(\boldsymbol{x}, \boldsymbol{x}'))^{N-1}\right| \leq 4N\rho\left(\frac{2}{2N+1}\right) e\sqrt{\frac{\log(2N/\delta)}{m}}.$$

Note that we only consider one weight matrix above, by applying the union bound, with probability at least $1 - \delta$ over the randomness of initialization, we have:

$$\left|\langle \nabla f(\boldsymbol{x}), \nabla f(\boldsymbol{x}') \rangle - K(\boldsymbol{x}, \boldsymbol{x}')\right| \leq 4N\rho\left(\frac{2}{2N+1}\right) e\sqrt{\frac{\log(2N/\delta)}{m}}.$$

$\square$

### B.3 Proof of Theorem 3

Before starting the proof, we introduce the following lemmas that are used to analyze the random initialization of the weight matrices $\boldsymbol{W}_n \in \mathbb{R}^{m \times d}, \forall n \in [N]$ and $\boldsymbol{W}_{N+1} \in \mathbb{R}^{1 \times m}$.

**Lemma 4.** *[Vershynin, 2010, Corollary 5.35] For a weight matrix $\boldsymbol{W} \in \mathbb{R}^{m \times d}$ where each element is sampled independently from $\mathcal{N}(0,1)$, for every $\zeta \geq 0$, with probability at least $1 - 2\exp(-\zeta^2/2)$ one has:*

$$\sqrt{m} - \sqrt{d} - \zeta \leq \lambda_{min}(\boldsymbol{W}) \leq \lambda_{\max}(\boldsymbol{W}) \leq \sqrt{m} + \sqrt{d} + \zeta,$$

*where $\lambda_{\max}(\boldsymbol{W})$ and $\lambda_{\min}(\boldsymbol{W})$ represents the largest and smallest singular value of $\boldsymbol{W}$, respectively.*

Next, we will show the local boundedness and the local Lipschitzness of the Jacobian. We use $\|\cdot\|_F$ and $\|\cdot\|$ to represent the Frobenius norm and spectral norm of a matrix, respectively. The Euclidean norm of a vector is symbolized by $\|\cdot\|_2$.

**Lemma 5.** *Consider the $N$-degree NNs-Hp in Eq. (1), assume the input $\boldsymbol{x} \in \mathbb{R}^d$ is bounded $\|\boldsymbol{x}\|_2 \leq 1$, then there exists $\gamma_1 > 0$, $\gamma_2 > 0$ (both are independent of the width $m$) such that for every $r > 0$, $\delta \in (0,1)$, $m \geq \left( r + \sqrt{d} + 2\log\left((2N+2)/\delta\right)\right)^2$, with probability at least $1 - \delta$ over the random initialization, the following holds for all $\boldsymbol{\theta}, \widetilde{\boldsymbol{\theta}} \in D(\boldsymbol{\theta}, r) := \{\boldsymbol{\theta} : \|\boldsymbol{\theta} - \boldsymbol{\theta}_0\|_2 \leq r\}$*

$$\|J(\boldsymbol{\theta})\|_F \leq \gamma_1, \tag{19}$$

$$\|J(\boldsymbol{\theta}) - J(\widetilde{\boldsymbol{\theta}})\|_F \leq \gamma_2 \|\boldsymbol{\theta} - \widetilde{\boldsymbol{\theta}}\|_2. \tag{20}$$

*Proof of Lemma 5.* Based on Lemma 4 and union bound, when $m \geq \left(r + \sqrt{d} + 2\log\left((2N+2)/\delta\right)\right)^2$, with probability at least $1 - \delta$ for any $\delta \in (0,1)$, the following inequalities hold for all $n = 1, \ldots, N$ simultaneously:

$$\|\boldsymbol{W}_n\| \leq \sqrt{m} + \sqrt{d} + \delta \leq 2\sqrt{m},$$
$$\|\boldsymbol{W}_{N+1}\| \leq \sqrt{m} + 1 + \delta \leq 2\sqrt{m},$$
$$\left\|\widetilde{\boldsymbol{W}_n}\right\| = \|\boldsymbol{W}_n + \Delta\boldsymbol{W}_n\| \leq \|\boldsymbol{W}_n\| + \|\Delta\boldsymbol{W}_n\| \leq \sqrt{m} + \sqrt{d} + \delta + \|\Delta\boldsymbol{W}_n\|_F$$
$$\leq \sqrt{m} + \sqrt{d} + \delta + \|\Delta\boldsymbol{\theta}\|_2 \leq \sqrt{m} + \sqrt{d} + \delta + r \leq 2\sqrt{m}.$$

Below, we abbreviate the description of probability and the width requirement since the following events rely on the same random initialization of the weight matrices. The following shorthand notations are made:

$$\widetilde{\boldsymbol{T}}_n = \left( \sqrt{\frac{2}{m}} \sigma(\widetilde{\boldsymbol{W}}_n \boldsymbol{x}) * \ldots * \sqrt{\frac{2}{m}} \sigma(\widetilde{\boldsymbol{W}}_1 \boldsymbol{x}) \right),$$

$$\boldsymbol{T}_n = \left( \sqrt{\frac{2}{m}} \sigma(\boldsymbol{W}_n \boldsymbol{x}) * \ldots * \sqrt{\frac{2}{m}} \sigma(\boldsymbol{W}_1 \boldsymbol{x}) \right).$$

Firstly, we prove the local boundness (Eq. (19)). Given that Eq. (1) is symmetric w.r.t $\{\boldsymbol{W}_i\}_{i=1}^N$, we start with calculating the bound with respect of one of the parameter matrix $\boldsymbol{W}_N$. The derivate is:

$$\partial_{\boldsymbol{W}_N} f(\boldsymbol{x}) = \left( \frac{2}{\sqrt{m}} \boldsymbol{W}_{N+1} * \frac{2}{\sqrt{m}} \dot{\sigma}(\boldsymbol{W}_N x) * \frac{2}{\sqrt{m}} \sigma(\boldsymbol{W}_{N-1} x) * \ldots * \frac{2}{\sqrt{m}} \sigma(\boldsymbol{W}_1 x) \right) \boldsymbol{x}^\top$$

$$= \left( \frac{2}{\sqrt{m}} \boldsymbol{W}_{N+1} * \frac{2}{\sqrt{m}} \dot{\sigma}(\boldsymbol{W}_N x) * \boldsymbol{T}_{N-1} \right) \boldsymbol{x}^\top.$$

Its Frobenius norm satisfies with probability:

$$\|\partial_{\boldsymbol{W}_N} f(\boldsymbol{x})\|_F = \left\| \left( \frac{2}{\sqrt{m}} \boldsymbol{W}_{N+1} * \frac{2}{\sqrt{m}} \dot{\sigma}\left(\boldsymbol{W}_N x\right) * \boldsymbol{T}_{N-1} \right) \boldsymbol{x}^\top \right\|_F$$

$$\leq \left\| \frac{2}{\sqrt{m}} \boldsymbol{W}_{N+1} * \frac{2}{\sqrt{m}} \dot{\sigma}\left(\boldsymbol{W}_N x\right) * \boldsymbol{T}_{N-1} \right\|_2$$

$$\leq \left\| \frac{2}{\sqrt{m}} \boldsymbol{W}_{N+1} \right\|_2 \left\| \frac{2}{\sqrt{m}} \dot{\sigma}\left(\boldsymbol{W}_N \boldsymbol{x}\right) \right\|_2 \|\boldsymbol{T}_{N-1}\|_2$$

$$\leq 4 \left\| \frac{2}{\sqrt{m}} \dot{\sigma}\left(\boldsymbol{W}_N \boldsymbol{x}\right) \right\|_2 \|\boldsymbol{T}_{N-1}\|_2$$

$$\leq 8 \|\boldsymbol{T}_{N-1}\|_2 \leq 2^{2N+1},$$

where the second and the third inequality use the Cauchy–Schwarz inequality, the last inequality is based on the upper bound of $\|\boldsymbol{T}_{N-1}\|_2$, which holds with probability:

$$\|\boldsymbol{T}_{N-1}\|_2 = \left\| \frac{2}{\sqrt{m}} \sigma\left(\boldsymbol{W}_{N-1} x\right) * \ldots * \frac{2}{\sqrt{m}} \sigma\left(\boldsymbol{W}_1 x\right) \right\|_2 \leq \left\| \frac{2}{\sqrt{m}} \sigma\left(\widehat{\boldsymbol{W}} x\right) \right\|_2^{N-1} \tag{21}$$

$$\leq \left( \frac{2}{\sqrt{m}} \|\widehat{\boldsymbol{W}}\| \right)^{N-1} \leq \left| \frac{2}{\sqrt{m}} 2\sqrt{m} \right|^{N-1} = 4^{N-1}, \tag{22}$$

where $\widehat{\boldsymbol{W}} \in \mathbb{R}^{m \times d}$ is sampled from $\mathcal{N}(\mathbf{0}, \mathbf{1})$. Now we consider all the weight matrices except $\boldsymbol{W}_{N+1}$ that is not trained. We have the following bound with probability:

$$\|J(\boldsymbol{\theta})\|_{\mathrm{F}} = \sqrt{\sum_{\boldsymbol{x} \in \mathcal{X}} \left( \sum_{n=1}^{N} \left\| \frac{\partial f(\boldsymbol{x}; \boldsymbol{\theta})}{\partial \boldsymbol{W}_n} \right\|_{\mathrm{F}}^2 \right)} \leq 2^{2N+1} \sqrt{N|\mathcal{X}|} = \gamma_1,$$

where $\gamma_1$ does not depend on the width $m$. This completes the first part of the proof.

Next, we prove the local Lipschitzness (Eq. (19)). Similarly, since Eq. (1) is symmetric w.r.t $\{\boldsymbol{W}_i\}_{i=1}^{N}$, firstly, we calculate the perturbation with respect of one of the parameter matrix $\boldsymbol{W}_N$. The following inequality holds with probability:

$$\left\| \partial_{\widetilde{\boldsymbol{W}}_N} f(\boldsymbol{x}) - \partial_{\boldsymbol{W}_N} f(\boldsymbol{x}) \right\|_{\mathrm{F}} \tag{23}$$

$$= \left\| \left( \frac{2}{\sqrt{m}} \boldsymbol{W}_{N+1} * \frac{2}{\sqrt{m}} \dot{\sigma} \left( \widetilde{\boldsymbol{W}}_N \boldsymbol{x} \right) * \widetilde{\boldsymbol{T}}_{N-1} \right) \boldsymbol{x}^\top - \left( \frac{2}{\sqrt{m}} \boldsymbol{W}_{N+1} * \frac{2}{\sqrt{m}} \dot{\sigma} \left( \boldsymbol{W}_N x \right) * \boldsymbol{T}_{N-1} \right) \boldsymbol{x}^\top \right\|_{\mathrm{F}} \tag{24}$$

$$\leq \left\| \frac{2}{\sqrt{m}} \boldsymbol{W}_{N+1} * \frac{2}{\sqrt{m}} \dot{\sigma} \left( \widetilde{\boldsymbol{W}}_N \boldsymbol{x} \right) * \widetilde{\boldsymbol{T}}_{N-1} - \frac{2}{\sqrt{m}} \boldsymbol{W}_{N+1} * \frac{2}{\sqrt{m}} \dot{\sigma} \left( \boldsymbol{W}_N x \right) * \boldsymbol{T}_{N-1} \right\|_2 \|\boldsymbol{x}\|_2 \tag{25}$$

$$\leq \left\| \frac{2}{\sqrt{m}} \boldsymbol{W}_{N+1} \right\|_2 \left\| \frac{2}{\sqrt{m}} \dot{\sigma} \left( \widetilde{\boldsymbol{W}}_N \boldsymbol{x} \right) * \widetilde{\boldsymbol{T}}_{N-1} - \frac{2}{\sqrt{m}} \dot{\sigma} \left( \boldsymbol{W}_N \boldsymbol{x} \right) * \boldsymbol{T}_{N-1} \right\|_2 \tag{26}$$

$$\leq 4 \left\| \frac{2}{\sqrt{m}} \dot{\sigma} \left( \widetilde{\boldsymbol{W}}_N \boldsymbol{x} \right) * \widetilde{\boldsymbol{T}}_{N-1} - \frac{2}{\sqrt{m}} \dot{\sigma} \left( \boldsymbol{W}_N \boldsymbol{x} \right) * \boldsymbol{T}_{N-1} \right\|_2 \tag{27}$$

$$\leq 4 \left\| \left( \frac{2}{\sqrt{m}} \dot{\sigma} \left( \boldsymbol{W}_N \boldsymbol{x} \right) - \frac{2}{\sqrt{m}} \dot{\sigma} \left( \widetilde{\boldsymbol{W}}_N \boldsymbol{x} \right) \right) * \boldsymbol{T}_{N-1} \right\|_2 + 4 \left\| \frac{2}{\sqrt{m}} \dot{\sigma} \left( \widetilde{\boldsymbol{W}}_N \boldsymbol{x} \right) * \left( \boldsymbol{T}_{N-1} - \widetilde{\boldsymbol{T}}_{N-1} \right) \right\|_2 \tag{28}$$

$$\leq 4 \left\| \left( \frac{2}{\sqrt{m}} \dot{\sigma} \left( \widetilde{\boldsymbol{W}}_N \boldsymbol{x} \right) - \frac{2}{\sqrt{m}} \dot{\sigma} \left( \boldsymbol{W}_N \boldsymbol{x} \right) \right) \right\|_2 \|\boldsymbol{T}_{N-1}\|_2 + 4 \left\| \frac{2}{\sqrt{m}} \dot{\sigma} \left( \widetilde{\boldsymbol{W}}_N \boldsymbol{x} \right) \right\|_2 \left\| \boldsymbol{T}_{N-1} - \widetilde{\boldsymbol{T}}_{N-1} \right\|_2 \tag{29}$$

$$\leq \frac{8}{\sqrt{m}} \left\| \widetilde{\boldsymbol{W}}_N \boldsymbol{x} - \boldsymbol{W}_N \boldsymbol{x} \right\|_2 \|\boldsymbol{T}_{N-1}\|_2 + 8 \left\| \boldsymbol{T}_{N-1} - \widetilde{\boldsymbol{T}}_{N-1} \right\|_2 \tag{30}$$

$$\leq \frac{8}{\sqrt{m}} \left\| \widetilde{\boldsymbol{W}}_N - \boldsymbol{W}_N \right\| \|\boldsymbol{T}_{N-1}\|_2 + 8 \left\| \boldsymbol{T}_{N-1} - \widetilde{\boldsymbol{T}}_{N-1} \right\|_2, \tag{31}$$

where Eq. (25) is due to $\|\boldsymbol{a}\boldsymbol{b}^\top\|_{\mathrm{F}} \leq \|\boldsymbol{a}\|_2 \|\boldsymbol{b}\|_2$ for two arbitrary vectors $\boldsymbol{a}$ and $\boldsymbol{b}$, Eq. (28) comes from triangle inequality, Eq. (30) is based on Lipschitz continuous gradient of the ReLU activation function. Using the result in Eq. (22) and the following inequality:

$$\left\| \widetilde{\boldsymbol{W}}_N - \boldsymbol{W}_N \right\| \leq \left\| \widetilde{\boldsymbol{W}}_N - \boldsymbol{W}_N \right\|_{\mathrm{F}} \leq \left\| \boldsymbol{\theta} - \widetilde{\boldsymbol{\theta}} \right\|_2.$$

The first term in Eq. (31) can be bounded with probability by:

$$\frac{8}{\sqrt{m}} \left\| \widetilde{\boldsymbol{W}}_N - \boldsymbol{W}_N \right\| \|\boldsymbol{T}_{N-1}\|_2 \leq \frac{2^{2N+1}}{\sqrt{m}} \left\| \boldsymbol{\theta} - \widetilde{\boldsymbol{\theta}} \right\|_2.$$

For the second term in Eq. (31), we will bound $\left\| \widetilde{\boldsymbol{T}}_{N-1} - \boldsymbol{T}_{N-1} \right\|_2$ by induction. Base case satisfies with probability:

$$\left\| \widetilde{\boldsymbol{T}}_1 - \boldsymbol{T}_1 \right\|_2 = \left\| \frac{2}{\sqrt{m}} \sigma \left( \widetilde{\boldsymbol{W}}_1 \boldsymbol{x} \right) - \frac{2}{\sqrt{m}} \sigma \left( \boldsymbol{W}_1 \boldsymbol{x} \right) \right\|_2 \leq \left\| \frac{2}{\sqrt{m}} \left( \widetilde{\boldsymbol{W}}_1 \boldsymbol{x} - \boldsymbol{W}_1 \boldsymbol{x} \right) \right\|_2$$

$$\leq \frac{2}{\sqrt{m}} \left\| \widetilde{\boldsymbol{W}}_1 - \boldsymbol{W}_1 \right\| \leq \frac{2}{\sqrt{m}} \left\| \boldsymbol{\theta} - \widetilde{\boldsymbol{\theta}} \right\|_2,$$

Assume $\left\|\widetilde{\boldsymbol{T}}_n - \boldsymbol{T}_n\right\|_2 \le \frac{C_2}{\sqrt{m}}\left\|\boldsymbol{\theta} - \widetilde{\boldsymbol{\theta}}\right\|_2$ , then, with probability:

$$
\begin{aligned}
&\left\|\widetilde{\boldsymbol{T}}_{n+1} - \boldsymbol{T}_{n+1}\right\|_2 \\
&\le \left\|\frac{2}{\sqrt{m}}\sigma\left(\widetilde{\boldsymbol{W}}_{n+1}\boldsymbol{x}\right) * \left(\widetilde{\boldsymbol{T}}_n - \boldsymbol{T}_n\right)\right\|_2 + \left\|\boldsymbol{T}_n * \frac{2}{\sqrt{m}}\left(\sigma\left(\widetilde{\boldsymbol{W}}_{n+1}\boldsymbol{x}\right) - \sigma\left(\boldsymbol{W}_{n+1}\boldsymbol{x}\right)\right)\right\|_2 \\
&\le \frac{2}{\sqrt{m}}\left\|\sigma\left(\widetilde{\boldsymbol{W}}_{n+1}\boldsymbol{x}\right)\right\|_2 \frac{C_2}{\sqrt{m}}\left\|\widetilde{\boldsymbol{\theta}} - \boldsymbol{\theta}\right\|_2 + \frac{2}{\sqrt{m}}\|\boldsymbol{T}_n\|_2\left\|\left(\sigma\left(\widetilde{\boldsymbol{W}}_{n+1}\boldsymbol{x}\right) - \sigma\left(\boldsymbol{W}_{n+1}\boldsymbol{x}\right)\right)\right\|_2 \\
&\le \frac{2}{\sqrt{m}}\left\|\widetilde{\boldsymbol{W}}_{n+1}\right\|\frac{C_2}{\sqrt{m}}\left\|\widetilde{\boldsymbol{\theta}} - \boldsymbol{\theta}\right\|_2 + \frac{2^{2n+1}}{\sqrt{m}}\|\Delta\boldsymbol{W}\| \\
&\le \frac{2}{\sqrt{m}}\left\|\widetilde{\boldsymbol{W}}_{n+1}\right\|\frac{C_2}{\sqrt{m}}\left\|\widetilde{\boldsymbol{\theta}} - \boldsymbol{\theta}\right\|_2 + \frac{2^{2n+1}}{\sqrt{m}}\left\|\widetilde{\boldsymbol{\theta}} - \boldsymbol{\theta}\right\|_2 \\
&\le \frac{2}{\sqrt{m}}2\sqrt{m}\frac{C_2}{\sqrt{m}}\left\|\widetilde{\boldsymbol{\theta}} - \boldsymbol{\theta}\right\|_2 + \frac{2^{2n+1}}{\sqrt{m}}\left\|\widetilde{\boldsymbol{\theta}} - \boldsymbol{\theta}\right\|_2 \\
&\le \left(\frac{4C_2}{\sqrt{m}} + \frac{2^{2n+1}}{\sqrt{m}}\right)\left\|\widetilde{\boldsymbol{\theta}} - \boldsymbol{\theta}\right\|_2 ,
\end{aligned}
$$

where the first inequality uses the triangle inequality. Thus, we can bound with probability:

$$
\left\|\widetilde{\boldsymbol{T}}_{N-1} - \boldsymbol{T}_{N-1}\right\|_2 \le \frac{2^{3N-5}}{\sqrt{m}}\left\|\boldsymbol{\theta} - \widetilde{\boldsymbol{\theta}}\right\|_2 .
$$

Then Eq. (31) becomes:

$$
\left\|\partial_{\widetilde{\boldsymbol{W}}_N}f(\boldsymbol{x}) - \partial_{\boldsymbol{W}_N}f(\boldsymbol{x})\right\|_{\mathrm{F}} \le \frac{8 + 2^{3N-5}}{\sqrt{m}}\left\|\boldsymbol{\theta} - \widetilde{\boldsymbol{\theta}}\right\|_2 .
$$

Now we consider all the weight matrices except $\boldsymbol{W}_{N+1}$ that is not trained. The following inequality holds with probability:

$$
\begin{aligned}
\|J(\boldsymbol{\theta}) - J(\widetilde{\boldsymbol{\theta}})\|_{\mathrm{F}} &= \sqrt{\sum_{\boldsymbol{x}\in\mathcal{X}}\left(\sum_{n=1}^{N}\left\|\frac{\partial f(\boldsymbol{x};\boldsymbol{\theta})}{\partial\boldsymbol{W}_n} - \frac{\partial f(\boldsymbol{x};\widetilde{\boldsymbol{\theta}})}{\partial\widetilde{\boldsymbol{W}}_n}\right\|_{\mathrm{F}}^2\right)} \\
&\le \frac{8 + 2^{3N-5}}{\sqrt{m}}\sqrt{N|\mathcal{X}|} \le \frac{8 + 2^{3N-5}}{r + \sqrt{d} + 2\log(2N/\delta)}\sqrt{N|\mathcal{X}|} = \gamma_2,
\end{aligned}
$$

where $\gamma_2$ does not depend on the width $m$. This completes the proof of Lemma 5. $\qquad\square$

Finally, note that Theorem 3 is an extension of Lee et al. [2019, Theorem G.1] from MLP to NNs-Hp, the proof of Lemma 5 is quite different for different networks while the idea of the remaining steps is based on the induction rule over time step $t$, which do not rely on the network. Applying Lemma 5 with $\gamma_1 = \gamma_2 = \frac{3QR_0}{\lambda_{min}(\boldsymbol{K})}$ completes the proof, which is similar to the extension from MLPs to ResNets in Tirer et al. [2020, Theorem 5], RNN in Alemohammad et al. [2021, Theorem 2].

## C   Proofs of extrapolation

### C.1   Proof of Theorem 4

*Proof of Theorem 4.* Recall that an infinite-width neural network trained through gradient descent is equivalent to kernel regression, the network output for any $\boldsymbol{x}\in\mathbb{R}^d$ is given by

$$
f(\boldsymbol{x}) = \left(K(\boldsymbol{x},\boldsymbol{x}_1),\cdots,K(\boldsymbol{x},\boldsymbol{x}_{|\mathcal{X}|})\right) \cdot \boldsymbol{K}^{-1}\boldsymbol{y},
$$

where $\boldsymbol{K}\in\mathbb{R}^{|\mathcal{X}|\times|\mathcal{X}|}$ is the NTK Gram matrix for training data, $K(\boldsymbol{x},\boldsymbol{x}_i)$ is the kernel value between test data $\boldsymbol{x}$ and training data $\boldsymbol{x}_i$, and $\boldsymbol{y}\in\mathbb{R}^{|\mathcal{X}|}$ are the training labels. Since the NTK $K(\boldsymbol{x},\boldsymbol{x}')$ is

1-homogeneous w.r.t $\boldsymbol{x}$, the network output $f(\boldsymbol{x})$ is also $N$-homogeneous w.r.t $\boldsymbol{x}$. Therefore, given inputs $\boldsymbol{x}_0 = t\boldsymbol{v}$ and $\boldsymbol{x} = \boldsymbol{x}_0 + h\boldsymbol{v}$, the outputs of the network are:

$$
\begin{aligned}
f(\boldsymbol{x}_0) &= f(t\boldsymbol{v}) = t^N \cdot f(\boldsymbol{v}) \\
f(\boldsymbol{x}) &= f\left(\left((t+h)^N\right)\boldsymbol{v}\right) = (t+h)^N \cdot f(\boldsymbol{v}),
\end{aligned}
$$

Thus:

$$
f(\boldsymbol{x}) - f(\boldsymbol{x}_0) = f(t\boldsymbol{v}) - f((t+h)\boldsymbol{v}) = \left((t+h)^N - t^N\right) \cdot f(\boldsymbol{v}),
$$

Thus, the network extrapolates to at most $N$-degree function with respect to $h$. $\qquad\square$

## C.2 Proof of Theorem 5

*Proof of Theorem 5.* The NTK of MLPs defined in Section 2.2, denoted by $K^{(N)}(\boldsymbol{x}, \boldsymbol{x}')$, can be rewritten in a more compact form [Nguyen et al., 2021]:

$$
K^{(N)}(\boldsymbol{x}, \boldsymbol{x}') = G^{(N)}(\boldsymbol{x}, \boldsymbol{x}') + \sum_{n=1}^{N-1} G^{(n)}(\boldsymbol{x}, \boldsymbol{x}') * \dot{G}^{(n+1)}(\boldsymbol{x}, \boldsymbol{x}') * \ldots * \dot{G}^{(N)}(\boldsymbol{x}, \boldsymbol{x}'), \qquad (32)
$$

where for $n \in [3, N]$:

$$
\begin{aligned}
K^{(1)}(\boldsymbol{x}, \boldsymbol{x}') &= G^{(1)}(\boldsymbol{x}, \boldsymbol{x}') = \boldsymbol{x}^\top \boldsymbol{x}' \\
G^{(2)}(\boldsymbol{x}, \boldsymbol{x}') &= 2\mathbb{E}_{\boldsymbol{w}\sim\mathcal{N}(0,I)}\left[\sigma(\boldsymbol{w}^\top \boldsymbol{x})\sigma(\boldsymbol{w}^\top \boldsymbol{x}')\right] \\
G^{(n)}(\boldsymbol{x}, \boldsymbol{x}') &= 2\mathbb{E}_{\boldsymbol{w}\sim\mathcal{N}(0,1)}\left[\sigma\left(\sqrt{G^{(n-1)}(\boldsymbol{x}, \boldsymbol{x}')}\boldsymbol{w}\right)\sigma\left(\sqrt{G^{(n-1)}(\boldsymbol{x}, \boldsymbol{x}')}\boldsymbol{w}\right)\right],
\end{aligned}
\qquad (33)
$$

for $n \in [2, N]$:

$$
\begin{aligned}
K^{(n)}(\boldsymbol{x}, \boldsymbol{x}') &= K^{(n-1)}(\boldsymbol{x}, \boldsymbol{x}') * \dot{G}^{(n)}(\boldsymbol{x}, \boldsymbol{x}') + G^{(n)}(\boldsymbol{x}, \boldsymbol{x}') \\
\dot{G}^{(n)}(\boldsymbol{x}, \boldsymbol{x}') &= 2\mathbb{E}_{w\sim\mathcal{N}(0,1)}\left[\sigma'\left(\sqrt{G^{(n-1)}(\boldsymbol{x}, \boldsymbol{x}')}w\right)\sigma'\left(\sqrt{G^{(n-1)}(\boldsymbol{x}, \boldsymbol{x}')}w\right)\right].
\end{aligned}
\qquad (34)
$$

Since the ReLU activation function $\sigma$ is 1-homogeneous and its derivative $\sigma'$ is 0-homogeneous, $G^{(n)}(\boldsymbol{x}, \boldsymbol{x}')$ is 1-homogeneous and $\dot{G}^{(n)}(\boldsymbol{x}, \boldsymbol{x}')$ 0-homogeneous w.r.t $\boldsymbol{x}$, for $n \in [1, N]$. Thus, the NTK $K^{(N)}(\boldsymbol{x}, \boldsymbol{x}')$ is 1-homogeneous w.r.t $\boldsymbol{x}$. Since the NTK $K^{(N)}(\boldsymbol{x}, \boldsymbol{x}')$ is 1-homogeneous w.r.t $\boldsymbol{x}$, the network output $f(\boldsymbol{x})$ is also 1-homogeneous w.r.t $\boldsymbol{x}$. Therefore, given inputs $\boldsymbol{x}_0 = t\boldsymbol{v}$ and $\boldsymbol{x} = \boldsymbol{x}_0 + h\boldsymbol{v}$, the outputs of the network are:

$$
\begin{aligned}
f(\boldsymbol{x}_0) &= f(t\boldsymbol{v}) = t \cdot f(\boldsymbol{v}) \\
f(\boldsymbol{x}) &= f\left((t+h)\boldsymbol{v}\right) = (t+h) \cdot f(\boldsymbol{v})
\end{aligned}
$$

Thus:

$$
f(\boldsymbol{x}) - f(\boldsymbol{x}_0) = h \cdot f(\boldsymbol{v}) \qquad (35)
$$

Since the term $f(\boldsymbol{v})$ in Eq. (35) is finite in our assumption, the network extrapolates to a linear function with respect to $h$. This completes the proof. $\qquad\square$

## C.3 Proof of Theorem 6

**Lemma 6.** *A specific feature map $\phi(\boldsymbol{x})$ induced by the NTK of a two-degree NNs-Hp with ReLU activation function is*

$$
\begin{aligned}
\phi(\boldsymbol{x}) &= \left(c'\boldsymbol{x} \cdot \dot{\sigma}\langle\boldsymbol{w}^{(k)}, \boldsymbol{x}\rangle \cdot \sigma(\langle\boldsymbol{w}^{(k)}, \boldsymbol{x}\rangle), c''\sigma(\langle\boldsymbol{w}^{(k)}, \boldsymbol{x}\rangle) \cdot \sigma(\langle\boldsymbol{w}^{(k)}, \boldsymbol{x}\rangle)\right) \\
&= \left(c'\boldsymbol{x} \cdot \langle\boldsymbol{w}^{(k)}, x\rangle \cdot \dot{\sigma}(\langle\boldsymbol{w}^{(k)}, \boldsymbol{x}\rangle), c''(\langle\boldsymbol{w}^{(k)}, x\rangle)^2 \cdot \dot{\sigma}(\langle\boldsymbol{w}^{(k)}, \boldsymbol{x}\rangle)\right),
\end{aligned}
\qquad (36)
$$

*where $\boldsymbol{w}^{(k)}$ is sampled from $\mathcal{N}(\boldsymbol{0}, \boldsymbol{I})$, $c'$ and $c''$ are constants, the last equality is due to the property of ReLU function: $\sigma(a) = a\dot{\sigma}(a)$ for any $a \in \mathbb{R}$.*

*Proof of Lemma 6.* The NTK for the second degree NNs-Hp with ReLU activation is given by

$$
\boldsymbol{K}(\boldsymbol{x}, \boldsymbol{x}') = \frac{8}{m} \cdot \mathop{\mathbb{E}}_{\boldsymbol{w} \sim \mathcal{N}(\boldsymbol{0}, \boldsymbol{I})} \left( \boldsymbol{x}^{\top} \boldsymbol{x}' \cdot \dot{\sigma}\left(\boldsymbol{w}^{\top} \boldsymbol{x}\right) \cdot \dot{\sigma}\left(\boldsymbol{w}^{\top} \boldsymbol{x}'\right) \right) \cdot \mathop{\mathbb{E}}_{\boldsymbol{w} \sim \mathcal{N}(\boldsymbol{0}, \boldsymbol{I})} \left( \sigma(\boldsymbol{w}^{\top} \boldsymbol{x}) \cdot \sigma(\boldsymbol{w}^{\top} \boldsymbol{x}') \right)
$$
$$
+ \frac{4}{m} \cdot \mathop{\mathbb{E}}_{\boldsymbol{w} \sim \mathcal{N}(\boldsymbol{0}, \boldsymbol{I})} \left( \sigma(\boldsymbol{w}^{\top} \boldsymbol{x}) \cdot \sigma(\boldsymbol{w}^{\top} \boldsymbol{x}') \right) \cdot \mathop{\mathbb{E}}_{\boldsymbol{w} \sim \mathcal{N}(\boldsymbol{0}, \boldsymbol{I})} \left( \sigma(\boldsymbol{w}^{\top} \boldsymbol{x}) \cdot \sigma(\boldsymbol{w}^{\top} \boldsymbol{x}') \right).
$$
(37)

Then, we utilize the kernel formula to construct the feature map that need to satisfy the following condition:

$$
K(\boldsymbol{x}, \boldsymbol{x}') = \langle \phi(\boldsymbol{x}), \phi(\boldsymbol{x}') \rangle. \tag{38}
$$

The following feature map would satisfy Eq. (38) because the inner product of $\phi(\boldsymbol{x})$ and $\phi(\boldsymbol{x}')$ for any $\boldsymbol{x}$, $\boldsymbol{x}'$ is equivalent to the expected value in Eq. (37), after integrating with respect to the density function of $\boldsymbol{w}$.

$$
\begin{aligned}
\phi(\boldsymbol{x}) &= \left( c' \boldsymbol{x} \cdot \dot{\sigma} \langle \boldsymbol{w}^{(k)}, \boldsymbol{x} \rangle \cdot \sigma(\langle \boldsymbol{w}^{(k)}, \boldsymbol{x} \rangle), c'' \sigma(\langle \boldsymbol{w}^{(k)}, \boldsymbol{x} \rangle) \cdot \sigma(\langle \boldsymbol{w}^{(k)}, \boldsymbol{x} \rangle) \right) \\
&= \left( c' \boldsymbol{x} \cdot \langle \boldsymbol{w}^{(k)}, x \rangle \cdot \dot{\sigma}(\langle \boldsymbol{w}^{(k)}, \boldsymbol{x} \rangle), c''(\langle \boldsymbol{w}^{(k)}, x \rangle)^2 \cdot \dot{\sigma}(\langle \boldsymbol{w}^{(k)}, \boldsymbol{x} \rangle) \right),
\end{aligned}
$$
(39)

where $\boldsymbol{w}^{(k)}$ is sampled from $\mathcal{N}(\boldsymbol{0}, \boldsymbol{I})$, $c'$ and $c''$ are constants, the last equality is due to the following property of ReLU function: $\sigma(a) = a\dot{\sigma}(a)$ for any $a \in \mathbb{R}$. $\qquad \square$

Sequentially, we are ready to prove Theorem 6.

*Proof of Theorem 6.* According to Xu et al. [2021, Lemma 2], the kernel regression solution is equivalent to the following form:

$$
f(\boldsymbol{x}) = \boldsymbol{\beta}^{\top} \phi(\boldsymbol{x}), \tag{40}
$$

where the representation coefficient $\boldsymbol{\beta}$ holds:

$$
\begin{aligned}
\min_{\boldsymbol{\beta}'} &\ \|\boldsymbol{\beta}'\|_2 \\
\text{s.t.} &\ \phi(\boldsymbol{x}_i)^{\top} \boldsymbol{\beta}' = y_i, \quad i = 1, \ldots, |\mathcal{X}|.
\end{aligned}
$$
(41)

The feature map $\phi(\boldsymbol{x})$ for a two-degree NNs-Hp with ReLU activation is given in Lemma 6

$$
\phi(\boldsymbol{x}) = \left( c' \boldsymbol{x} \cdot \langle \boldsymbol{w}^{(k)}, x \rangle \cdot \dot{\sigma}(\langle \boldsymbol{w}^{(k)}, \boldsymbol{x} \rangle), c''(\langle \boldsymbol{w}^{(k)}, x \rangle)^2 \cdot \dot{\sigma}(\langle \boldsymbol{w}^{(k)}, \boldsymbol{x} \rangle) \right),
$$

where $\boldsymbol{w}^{(k)}$ is sampled from $\mathcal{N}(\boldsymbol{0}, \boldsymbol{I})$, and $c', c''$ are constant. Below, for avoiding complicating the notation, we will discard the index and use $\boldsymbol{w}$ to represent a specific $\boldsymbol{w}^{(k)}$, We assume the constants $c'$ and $c''$ are 1. Note that $\boldsymbol{\beta}$ consists of weights for each $\boldsymbol{x}\boldsymbol{x}^{\top}\boldsymbol{w} \cdot \mathbb{I}\left(\boldsymbol{w}^{\top}\boldsymbol{x} \geq 0\right) \in \mathbb{R}^d$ and $\boldsymbol{x}^{\top}\boldsymbol{w}\boldsymbol{w}^{\top}\boldsymbol{x} \cdot \mathbb{I}\left(\boldsymbol{w}^{\top}\boldsymbol{x} \geq 0\right) \in \mathbb{R}$. For any $\boldsymbol{w} \in \mathbb{R}^d$, the weight vectors corresponding to $\boldsymbol{x}\boldsymbol{x}^{\top}\boldsymbol{w} \cdot \mathbb{I}\left(\boldsymbol{w}^{\top}\boldsymbol{x} \geq 0\right)$ are symbolized by $\hat{\boldsymbol{\beta}}_{\boldsymbol{w}} = (\hat{\boldsymbol{\beta}}_{\boldsymbol{w}}^{(1)}, \ldots, \hat{\boldsymbol{\beta}}_{\boldsymbol{w}}^{(k)}) \in \mathbb{R}^d$ and the weight vectors for $\boldsymbol{x}^{\top}\boldsymbol{w}\boldsymbol{w}^{\top}\boldsymbol{x} \cdot \mathbb{I}\left(\boldsymbol{w}^{\top}\boldsymbol{x} \geq 0\right)$ are symbolized by $\hat{\boldsymbol{\beta}}_{\boldsymbol{w}}' \in \mathbb{R}$. Given the fact that if $\boldsymbol{w}^{\top}\boldsymbol{x}_i \geq 0$ for any $\boldsymbol{w} \in \mathbb{R}^d$, then $c\boldsymbol{w}^{\top}\boldsymbol{x}_i \geq 0$ for any $c > 0$, we use the notation $\boldsymbol{\beta}_{\boldsymbol{w}}$ and $\boldsymbol{\beta}_{\boldsymbol{w}}'$ to represent the combined effect of all weights $(\hat{\boldsymbol{\beta}}_{c\boldsymbol{w}}^{(1)}, \ldots, \hat{\boldsymbol{\beta}}_{c\boldsymbol{w}}^{(k)}) \in \mathbb{R}^d$ and $\hat{\boldsymbol{\beta}}_{c\boldsymbol{w}}' \in \mathbb{R}$ for all $c\boldsymbol{w}$ with $c > 0$. This allows us to change the distribution of $\boldsymbol{w}$ from $\mathcal{N}(\boldsymbol{0}, \boldsymbol{I}_d)$ to $\mathrm{Unif}(\mathbb{S}^d)$. Specifically, for each $\boldsymbol{w} \sim \mathrm{Unif}(\mathbb{S}^d)$, $\boldsymbol{\beta}_{\boldsymbol{w}}^{(j)}$ is denoted as the total effect of the weights in the same direction of $\boldsymbol{w}$.

$$
\boldsymbol{\beta}_{\boldsymbol{w}}^{(j)} = \int \hat{\boldsymbol{\beta}}_{\boldsymbol{u}}^{(j)} \mathbb{I}\left( \frac{\boldsymbol{w}^{\top}\boldsymbol{u}}{\|\boldsymbol{w}\| \cdot \|\boldsymbol{u}\|} = 1 \right) d\mathbb{P}(\boldsymbol{u}), \quad j \in [d]
$$

where $\boldsymbol{u} \sim \mathcal{N}(\boldsymbol{0}, \boldsymbol{I})$. Similarly, $\boldsymbol{\beta}_{\boldsymbol{w}}'$ is defined as follows:

$$
\boldsymbol{\beta}_{\boldsymbol{w}}' = \int \hat{\boldsymbol{\beta}}_{\boldsymbol{u}} \mathbb{I}\left( \frac{\boldsymbol{w}^{\top}\boldsymbol{u}}{\|\boldsymbol{w}\| \cdot \|\boldsymbol{u}\|} = 1 \right) \cdot \|\boldsymbol{u}\| d\mathbb{P}(\boldsymbol{u}) \tag{42}
$$

Then, the min-norm solution in Eq. (41) is equivalent to:

$$\min_{\boldsymbol{\beta}} \int \left(\boldsymbol{\beta}_{\boldsymbol{w}}^{(1)}\right)^2 + \left(\boldsymbol{\beta}_{\boldsymbol{w}}^{(2)}\right)^2 + ... + \left(\boldsymbol{\beta}_{\boldsymbol{w}}^{(k)}\right)^2 + (\boldsymbol{\beta}'_{\boldsymbol{w}})^2 \; \mathrm{d}\mathbb{P}(\boldsymbol{w}) \tag{43}$$

$$\text{s.t.} \quad \int_{\boldsymbol{w}^\top \boldsymbol{x}_i \geq 0} \boldsymbol{x}_i^\top \boldsymbol{\beta}_{\boldsymbol{w}} \boldsymbol{w}^\top \boldsymbol{x}_i + \boldsymbol{x}_i^\top \boldsymbol{\beta}'_{\boldsymbol{w}} \boldsymbol{w} \boldsymbol{w}^\top \boldsymbol{x}_i \; \mathrm{d}\mathbb{P}(\boldsymbol{w}) = \boldsymbol{x}_i^\top \boldsymbol{\beta}_g \boldsymbol{x}_i \quad \forall i \in [|\mathcal{X}|], \tag{44}$$

where $\boldsymbol{w} \in \mathrm{Unif}(\mathbb{S}^d)$. Thus, $\mathbb{P}(\boldsymbol{w})$ is a constant, which indicates that only half of the $\boldsymbol{w}$ on the unit sphere activate each specific $\boldsymbol{x}_i$. Therefore, we can further simplify the constraint in Eq. (44) as

$$\int_{\boldsymbol{w}^\top \boldsymbol{x}_i \geq 0} \boldsymbol{x}_i^\top \left(\boldsymbol{\beta}_{\boldsymbol{w}} \boldsymbol{w}^\top + \boldsymbol{\beta}'_{\boldsymbol{w}} \boldsymbol{w} \boldsymbol{w}^\top - 2\boldsymbol{\beta}_g\right) \boldsymbol{x}_i \; \mathrm{d}\mathbb{P}(\boldsymbol{w}) = 0 \quad \forall i \in [|\mathcal{X}|], \tag{45}$$

where Eq. (45) follows from the following steps

$$\int_{\boldsymbol{w}^\top \boldsymbol{x}_i \geq 0} \boldsymbol{x}_i^\top \boldsymbol{\beta}_{\boldsymbol{w}} \boldsymbol{w}^\top \boldsymbol{x}_i + \boldsymbol{x}_i^\top \boldsymbol{\beta}'_{\boldsymbol{w}} \boldsymbol{w} \boldsymbol{w}^\top \boldsymbol{x}_i \mathrm{d}\mathbb{P}(\boldsymbol{w}) = \boldsymbol{x}_i^\top \boldsymbol{\beta}_g \boldsymbol{x}_i \; \forall i \in [|\mathcal{X}|],$$

$$\iff \int_{\boldsymbol{w}^\top \boldsymbol{x}_i \geq 0} \boldsymbol{x}_i^\top \boldsymbol{\beta}_{\boldsymbol{w}} \boldsymbol{w}^\top \boldsymbol{x}_i + \boldsymbol{x}_i^\top \boldsymbol{\beta}'_{\boldsymbol{w}} \boldsymbol{w} \boldsymbol{w}^\top \boldsymbol{x}_i \mathrm{d}\mathbb{P}(\boldsymbol{w})$$

$$= \frac{1}{\int_{\boldsymbol{w}^\top \boldsymbol{x}_i \geq 0} \mathrm{d}\mathbb{P}(\boldsymbol{w})} \cdot \int_{\boldsymbol{w}^\top \boldsymbol{x}_i \geq 0} \mathrm{d}\mathbb{P}(\boldsymbol{w}) \cdot \boldsymbol{x}_i^\top \boldsymbol{\beta}_g \boldsymbol{x}_i \quad \forall i \in [|\mathcal{X}|],$$

$$\iff \int_{\boldsymbol{w}^\top \boldsymbol{x}_i \geq 0} \boldsymbol{x}_i^\top \boldsymbol{\beta}_{\boldsymbol{w}} \boldsymbol{w}^\top \boldsymbol{x}_i + \boldsymbol{x}_i^\top \boldsymbol{\beta}'_{\boldsymbol{w}} \boldsymbol{w} \boldsymbol{w}^\top \boldsymbol{x}_i \mathrm{d}\mathbb{P}(\boldsymbol{w})$$

$$= 2 \cdot \int_{\boldsymbol{w}^\top \boldsymbol{x}_i \geq 0} \boldsymbol{x}_i^\top \boldsymbol{\beta}_g \boldsymbol{x}_i \mathrm{d}\mathbb{P}(\boldsymbol{w}) \quad \forall i \in [|\mathcal{X}|],$$

$$\iff \int_{\boldsymbol{w}^\top \boldsymbol{x}_i \geq 0} \boldsymbol{x}_i^\top \left(\boldsymbol{\beta}_{\boldsymbol{w}} \boldsymbol{w}^\top + \boldsymbol{\beta}'_{\boldsymbol{w}} \boldsymbol{w} \boldsymbol{w}^\top - 2\boldsymbol{\beta}_g\right) \boldsymbol{x}_i \; \mathrm{d}\mathbb{P}(\boldsymbol{w}) = 0 \quad \forall i \in [|\mathcal{X}|].$$

**Lemma 7.** *The global optimum of Eq. (43) subject to Eq. (45), i.e.,*

$$\min_{\boldsymbol{\beta}} \int \left(\boldsymbol{\beta}_{\boldsymbol{w}}^{(1)}\right)^2 + \left(\boldsymbol{\beta}_{\boldsymbol{w}}^{(2)}\right)^2 + ... + \left(\boldsymbol{\beta}_{\boldsymbol{w}}^{(k)}\right)^2 + (\boldsymbol{\beta}'_{\boldsymbol{w}})^2 \; \mathrm{d}\mathbb{P}(\boldsymbol{w}) \tag{46}$$

$$\text{s.t.} \quad \int_{\boldsymbol{w}^\top \boldsymbol{x}_i \geq 0} \boldsymbol{x}_i^\top \left(\boldsymbol{\beta}_{\boldsymbol{w}} \boldsymbol{w}^\top + \boldsymbol{\beta}'_{\boldsymbol{w}} \boldsymbol{w} \boldsymbol{w}^\top - 2\boldsymbol{\beta}_g\right) \boldsymbol{x}_i \; \mathrm{d}\mathbb{P}(\boldsymbol{w}) = 0 \quad \forall i \in [|\mathcal{X}|], \tag{47}$$

*satisfies $\boldsymbol{\beta}_{\boldsymbol{w}} \boldsymbol{w}^\top + \boldsymbol{\beta}'_{\boldsymbol{w}} \boldsymbol{w} \boldsymbol{w}^\top = 2\boldsymbol{\beta}_g$ for all $\boldsymbol{w}$.*

*Proof of Lemma 7.* Through Lemma 7, we can achieve the goal of our proof towards Theorem 6, i.e., $f(\boldsymbol{x}) = f_\rho(\boldsymbol{x})$. The reason is that if Lemma 7 holds, for any $\boldsymbol{x} \in \mathbb{R}^d$:

$$f(\boldsymbol{x}) = \int_{\boldsymbol{w}^\top \boldsymbol{x} \geq 0} \boldsymbol{x}^\top \left(\boldsymbol{\beta}_{\boldsymbol{w}} \boldsymbol{w}^\top + \boldsymbol{\beta}'_{\boldsymbol{w}} \boldsymbol{w} \boldsymbol{w}^\top\right) \boldsymbol{x} \; \mathrm{d}\mathbb{P}(\boldsymbol{w})$$

$$= \int_{\boldsymbol{w}^\top \boldsymbol{x} \geq 0} 2\boldsymbol{x}^\top \boldsymbol{\beta}_g \boldsymbol{x} \; \mathrm{d}\mathbb{P}(\boldsymbol{w})$$

$$= \int_{\boldsymbol{w}^\top \boldsymbol{x} \geq 0} \mathrm{d}\mathbb{P}(\boldsymbol{w}) 2\boldsymbol{x}^\top \boldsymbol{\beta}_g \boldsymbol{x}$$

$$= \frac{1}{2} 2\boldsymbol{x}^\top \boldsymbol{\beta}_g \boldsymbol{x} = f_\rho(\boldsymbol{x}).$$

Therefore, the remaining step is to prove Lemma 7. Since Eq. (46) is a convex optimization problem with affine constraint Eq. (47), we can introduce the Lagrange multipliers and use the Karush–Kuhn–Tucker (KKT) condition. The Lagrange multiplier has the following form:

$$\mathcal{L}(\boldsymbol{\beta}, \lambda) = \int \left(\boldsymbol{\beta}_{\boldsymbol{w}}^{(1)}\right)^2 + \left(\boldsymbol{\beta}_{\boldsymbol{w}}^{(2)}\right)^2 + ... + \left(\boldsymbol{\beta}_{\boldsymbol{w}}^{(k)}\right)^2 + (\boldsymbol{\beta}'_{\boldsymbol{w}})^2 \; \mathrm{d}\mathbb{P}(\boldsymbol{w}) \tag{48}$$

$$+ \sum_{i=1}^{|\mathcal{X}|} \lambda_i \cdot \left(\int_{\boldsymbol{w}^\top \boldsymbol{x}_i \geq 0} \boldsymbol{x}_i^\top \left(\boldsymbol{\beta}_{\boldsymbol{w}} \boldsymbol{w}^\top + \boldsymbol{\beta}'_{\boldsymbol{w}} \boldsymbol{w} \boldsymbol{w}^\top - 2\boldsymbol{\beta}_g\right) \boldsymbol{x}_i \; \mathrm{d}\mathbb{P}(\boldsymbol{w}) = 0\right). \tag{49}$$

By setting the partial derivative to zero, we obtain:

$$\frac{\partial \mathcal{L}}{\partial \boldsymbol{\beta}_{\boldsymbol{w}}^{(k)}} = 2\boldsymbol{\beta}_{\boldsymbol{w}}^{(k)}\mathbb{P}(\boldsymbol{w}) + \sum_{i=1}^{|\mathcal{X}|} \lambda_i \cdot (\boldsymbol{x}_i \boldsymbol{x}_i^\top \boldsymbol{w})^d \cdot \mathbb{I}\left(\boldsymbol{w}^\top \boldsymbol{x}_i \geq 0\right) = 0 \tag{50}$$

$$\frac{\partial \mathcal{L}}{\partial \boldsymbol{\beta}_{\boldsymbol{w}}'} = 2\boldsymbol{\beta}_{\boldsymbol{w}}'\mathbb{P}(\boldsymbol{w}) + \sum_{i=1}^{|\mathcal{X}|} \lambda_i \cdot \boldsymbol{x}_i \boldsymbol{w}\boldsymbol{w}^\top \boldsymbol{x}_i \cdot \mathbb{I}\left(\boldsymbol{w}^\top \boldsymbol{x}_i \geq 0\right) = 0 \tag{51}$$

$$\frac{\partial \mathcal{L}}{\partial \lambda_i} = \int_{\boldsymbol{w}^\top \boldsymbol{x}_i \geq 0} \boldsymbol{x}_i^\top \left(\boldsymbol{\beta}_{\boldsymbol{w}}\boldsymbol{w}^\top + \boldsymbol{\beta}_{\boldsymbol{w}}'\boldsymbol{w}\boldsymbol{w}^\top - 2\boldsymbol{\beta}_g\right)\boldsymbol{x}_i \ \mathrm{d}\mathbb{P}(\boldsymbol{w}) = 0\,. \tag{52}$$

It is obvious that the solution in Lemma 7 satisfies Eq. (52). Thus, the remaining step is to show that there exist a set of $\lambda_i$ where $i \in [|\mathcal{X}|]$ that satisfies Eq. (50) and Eq. (51). We simplify Eq. (50) and Eq. (51) as follows:

$$\boldsymbol{\beta}_{\boldsymbol{w}}^{(k)} = c \cdot \sum_{i=1}^{|\mathcal{X}|} \lambda_i \cdot (\boldsymbol{x}_i \boldsymbol{x}_i^\top \boldsymbol{w})^d \cdot \mathbb{I}\left(\boldsymbol{w}^\top \boldsymbol{x}_i \geq 0\right), \tag{53}$$

$$\boldsymbol{\beta}_{\boldsymbol{w}}' = c \cdot \sum_{i=1}^{|\mathcal{X}|} \lambda_i \cdot \boldsymbol{x}_i \boldsymbol{w}\boldsymbol{w}^\top \boldsymbol{x}_i \cdot \mathbb{I}\left(\boldsymbol{w}^\top \boldsymbol{x}_i \geq 0\right), \tag{54}$$

where $c$ is a constant. Combining Eq. (53) and Eq. (54), we can simplify the constraint Eq. (54) as follows:

$$\boldsymbol{\beta}_{\boldsymbol{w}}' = \boldsymbol{\beta}_{\boldsymbol{w}}\boldsymbol{w}^\top\,. \tag{55}$$

The remaining step is to show that based on the condition on training data, there exists a set of $\lambda_i$ that satisfy Eq. (53) and Eq. (55). For each $\boldsymbol{w}$, there must exist a set of $\lambda_i$ so that the following equations satisfy:

$$\boldsymbol{\beta}_{\boldsymbol{w}}^{(k)} = c \cdot \sum_{i=1}^{|\mathcal{X}|} \lambda_i \cdot (\boldsymbol{x}_i \boldsymbol{x}_i^\top \boldsymbol{w})^d \cdot \mathbb{I}\left(\boldsymbol{w}^\top \boldsymbol{x}_i \geq 0\right) \tag{56}$$

$$\boldsymbol{\beta}_{\boldsymbol{w}}' = \boldsymbol{\beta}_{\boldsymbol{w}}^\top \boldsymbol{w} \tag{57}$$

$$\boldsymbol{\beta}_{\boldsymbol{w}}\boldsymbol{w}^\top + \boldsymbol{\beta}_{\boldsymbol{w}}'\boldsymbol{w}\boldsymbol{w}^\top = 2\boldsymbol{\beta}_g\,, \tag{58}$$

where $\boldsymbol{\beta}_g$ and $\boldsymbol{w}$ are fixed. From Eq. (57) and Eq. (58), we can see that $\boldsymbol{\beta}_{\boldsymbol{w}}$ is determined by $\boldsymbol{\beta}_g$ and $\boldsymbol{w}$, and there exists a solution for this consistent linear system. Next, we are left with the following linear system that contains $d$ linear equations

$$\boldsymbol{\beta}_{\boldsymbol{w}}^{(k)} = c \cdot \sum_{i=1}^{|\mathcal{X}|} \lambda_i \cdot (\boldsymbol{x}_i \boldsymbol{x}_i^\top \boldsymbol{w})^d \cdot \mathbb{I}\left(\boldsymbol{w}^\top \boldsymbol{x}_i \geq 0\right), \quad \forall k \in [d].$$

Recall the assumption for the training data, there exist at least $d$ linearly independent $\boldsymbol{x}_i$ that activates a specific $\boldsymbol{w}$. This implies that for any $\boldsymbol{w}$ there exists at least $d$ free variables. Thus, the solutions for this linear system exist. $\qquad\square$

Thus, the proof of Theorem 6 is finished. $\qquad\square$

## D  Proof of spectral bias

Let us recall the core notation. Denote by $\{Y_{k,j}\}_{j=1}^{F(d,k)}$ the $k$-degree spherical harmonics in $d+1$ variables. $G_k^{(\gamma)}$ represents the Gegenbauer polynomials with respect to the weight function $x \mapsto (1-x^2)^{\gamma-\frac{1}{2}}$ and degree $k$. Finally, denote by $F(d,k) := \frac{2k+d-1}{k}\binom{k+d-2}{d-1}$.

Given the fact that $\kappa_1$ and $\kappa_2$ are also dot-product Mercer kernels, their corresponding decompositions in terms of Gegenbauer polynomials can be provided based on Eq. (5):

$$
\begin{aligned}
\langle \boldsymbol{x}, \boldsymbol{x}' \rangle \kappa_1(\boldsymbol{x}, \boldsymbol{x}') &= \sum_{k=0}^{\infty} \mu_{1,k} F(d,k) G_k^{(\frac{d-1}{2})}(\langle \boldsymbol{x}, \boldsymbol{x}' \rangle) \\
\kappa_2(\boldsymbol{x}, \boldsymbol{x}') &= \sum_{k=0}^{\infty} \mu_{2,k} F(d,k) G_k^{(\frac{d-1}{2})}(\langle \boldsymbol{x}, \boldsymbol{x}' \rangle).
\end{aligned}
\tag{59}
$$

Note that the decay in $\mu_{1,k} = \mu_{2,k}$ is $\Omega(k^{-d-1})$ [Bach, 2017, Cao et al., 2019, Bietti and Mairal, 2019]. Sequentially, we are ready to prove Theorem 7.

*Proof of Theorem 7.* For $N$-degree NNs-Hp, in order to study the decay rate of the eigenvalues, we express the NTK obtained in Eq. (2) as the product of multiple kernels:

$$
\begin{aligned}
K(\boldsymbol{x}, \boldsymbol{x}') = 2 &\left( \sqrt{\frac{2}{m}} \sum_{k=0}^{\infty} (N\mu_{1,k} + \mu_{2,k}) F(d,k) G_k^{(\frac{d-1}{2})}(\langle \boldsymbol{x}, \boldsymbol{x}' \rangle) \right) \\
&\cdot \left( \sqrt{\frac{2}{m}} \left( \sum_{k=0}^{\infty} \mu_{2,k} F(d,k) G_k^{(\frac{d-1}{2})}(\langle \boldsymbol{x}, \boldsymbol{x}' \rangle) \right) \right)^{N-1}
\end{aligned}
\tag{60}
$$

Comparing the above equation with Eq. (5), it turns out that we need to simplify Eq. (60) and equate the polynomial coefficients on both equations. It is obvious that we get the form of the product of multiple polynomials in Eq. (60). Fortunately, the following Lemma allows us to express the product of two Gegenbauer polynomials as a linear combination of other Gegenbauer polynomials.

**Lemma 8.** *[Carlitz, 1961, Eq (8)] For $b \in \mathbb{R}$ and any $p, q \in \mathbb{N}$, there exists a set of positive coefficients $\{\lambda_s^{(p,q)}\}_{s=0}^{\min(p,q)}$ such that*

$$
G_p^{(b)}(x) G_q^{(b)}(x) = \sum_{s=0}^{\min(p,q)} \lambda_s^{(p,q)} G_{p+q-2s}^{(b)}(x),
\tag{61}
$$

*where*

$$
\lambda_s^{(p,q)} = \frac{p+q+v-2s}{p+q+v-s} \cdot \frac{(v)_s (v)_{p-s} (v)_{q-s}}{s!(p-s)!(q-s)!} \cdot \frac{(2v)_{p+q-s}}{(v)_{p+q-s}} \cdot \frac{(p+q-2s)!}{(2v)_{p+q-2s}},
$$

*and*

$$
(v)_k := v(v+1)(v+2)....(v+k-1), \quad (v)_0 := 1.
$$

For convenience, we assume $v$ is an integer and $k$ even, then we set $p = q = k$, $s = 0$, and apply Lemma 8 recursively by $N$ times, we can obtain the lower bound regarding the coefficient of the term $C_{Nk}^{(\frac{d-1}{2})}$, which is the $(Nk)^{\text{th}}$ harmonic:

$$
\mu_{Nk} F(d, Nk) \geq \left( \sqrt{\frac{2}{m}} F(d,k) \right)^N (N\mu_{1,k} + \mu_{2,k}) \mu_{2,k}^{N-1} \prod_{\alpha=1}^{N} \lambda_0^{(k,\alpha k)}.
\tag{62}
$$

It suffices to obtain the form of $\lambda_0^{(k,\alpha k)}$. The coefficient $\lambda_0^{(k,\alpha k)}$ defined in Lemma 8

$$
\begin{aligned}
\lambda_0^{(k,\alpha k)} &= \frac{(\alpha k + k) + v}{(\alpha k + k) + v} \cdot \frac{(v)_0 (v)_k (v)_{\alpha k}}{0!(k)!(\alpha k)!} \cdot \frac{(2v)_{(\alpha k+k)}}{(v)_{(\alpha k+k)}} \cdot \frac{(\alpha k + k)!}{(2v)_{(\alpha k+k)}} \\
&= \frac{(v)_k (v)_{\alpha k}}{(k)!(\alpha k)!} \cdot \frac{(\alpha k + k)!}{(v)_{(\alpha k+k)}} \\
&= \frac{(v+k-1)!(v+\alpha k-1)!}{((v-1)!)^2 (k)!(\alpha k)!} \cdot \frac{(\alpha k + k)!(v-1)!}{(v+\alpha k+k-1)!} \\
&= \frac{(v+k-1)!(v+\alpha k-1)!}{(v-1)!(k)!(\alpha k)!} \cdot \frac{(\alpha k + k)!}{(v+\alpha k+k-1)!} \\
&\sim \frac{(v+k-1)^{(v+k-0.5)}(v+\alpha k-1)^{(v+\alpha k-0.5)}}{(v-1)^{(v-0.5)} k^{(k+0.5)}(\alpha k)^{(\alpha k+0.5)}} \cdot \frac{(\alpha k + k)^{(\alpha k+k+0.5)}}{(v+\alpha k+k-1)^{(v+\alpha k+k-0.5)}},
\end{aligned}
\tag{63}
$$

where we apply the Stirling's approximation ($n! \sim \sqrt{2\pi n}(\frac{n}{e})^n$) at the final step. Next, we consider the case when $k \gg v$. In order to match the term $C_{Nk}^{(\frac{d-1}{2})}$ in Eq. (60), we set $v = (d-1)/2$ and obtain:

$$
\begin{aligned}
\lambda_0^{(k,\alpha k)} &\sim \frac{(k)^{(v+k-0.5)}(\alpha k)^{(v+\alpha k-0.5)}}{k^{(k+0.5)}(\alpha k)^{(\alpha k+0.5)}} \cdot \frac{(\alpha k + k)^{(\alpha k+k+0.5)}}{(\alpha k + k)^{(v+\alpha k+k-0.5)}} \\
&\sim (k)^{(v-1)}(\alpha k)^{(v-1)}(\alpha k + k)^{(-v-1)} \sim \left(\frac{\alpha k}{1+\alpha}\right)^{v-1} = \left(\frac{\alpha k}{1+\alpha}\right)^{\frac{d-3}{2}}.
\end{aligned}
\tag{64}
$$

Plugging Eq. (64) into Eq. (62), we obtain:

$$
\begin{aligned}
\mu_{Nk} &\geq \frac{\left(\sqrt{\frac{2}{m}}F(d,k)\right)^N}{F(d,Nk)}(N\mu_{1,k}+\mu_{2,k})\mu_{2,k}^{N-1}\prod_{\alpha=1}^{N}\lambda_0^{(k,\alpha k)} \\
&\sim \frac{F(d,k)^N}{F(d,Nk)}(N\mu_{1,k}+\mu_{2,k})\mu_{2,k}^{N-1}\left(\frac{k}{N}\right)^{\frac{d-3}{2}} \\
&\sim \frac{k^{Nd}}{(Nk)^d}(N\mu_{1,k}+\mu_{2,k})\mu_{2,k}^{N-1}\left(\frac{k}{N}\right)^{\frac{d-3}{2}} \quad \text{(by Stirling)} \\
&\sim \frac{k^{Nd}}{(Nk)^d}\Omega(k^{-Nd-N})\left(\frac{k}{N}\right)^{\frac{d-3}{2}} \\
&\sim \Omega((kN^3)^{-\frac{d}{2}})
\end{aligned}
\tag{65}
$$

Setting $k = k'/N$ allows us to conclude the proof. $\qquad\square$

# E    Details on the numerical experiments

In the following content, we will describe the setup of several experiments including learning analytically-known function (Appendix E.1), variation of darkness (Appendix E.2), arithmetic extrapolation (Appendix E.3), and learning harmonics (Appendix E.4). The experiment of visual analogy task is included in Appendix E.5. The experiment on the spectral bias in image classification is contained in Appendix E.6

## E.1    Experimental setup in learning analytically-known function

We describe the experimental setup corresponding to Section 4.1. $N$-layer fully-connected NNs are compared against NNs-Hp with $N-1$ degree multiplicative interactions. The reason is that one-degree PNNs are equivalent to two-layer fully-connected NNs, according to the formula of PNNs provided in Eq. (1). In the experiment, the training set consists of 20000 data points in total. The networks are trained for 50 epochs with batch size 256. The squared loss is minimized through ADAM optimizer [Kingma and Ba, 2015] with $\beta_1 = 0.9$, $\beta_2 = 0.999$, learning rate $= 10^{-4}$.

As a complement, we show additional results of fitting two-variable functions in Figure 6 to further examine the power of Hadamard product.

In our work, the extrapolation relies on the support of the training data [van Schuppen, 2021], as suggested by the previous work of Xu et al. [2021]. We note that for certain applications and input data types, the convex hull might be required, however, we leave this as future work.

## E.2    Experimental setup in variation of darkness

This section describes the experimental setup of the variation of darkness experiment in Section 4.2. The following two datasets are used: (a) MNIST dataset [LeCun et al., 1998], which contains handwritten digits images from zero to nine. There are $60,000$ examples in the training set and

10, 000 examples in the testing set. Each image has the resolution $28 \times 28$. (b) Fashion-MNIST dataset [Xiao et al., 2017], which contains images of clothing with 10 classes. There are 60, 000 examples in the training set and 10, 000 examples in the testing set. Each image has the resolution $28 \times 28$. The networks are trained for 20 epochs with batch size 128 with the criterion of cross entropy loss. The learning rate is chosen as 0.01 . The width of the networks is 256. Each network is trained for 3 runs.

### E.3   Experimental setup in arithmetic extrapolation

This section describes the experimental setup of the arithmetic extrapolation experiment in Section 4.2. We construct a new dataset based on the MNIST dataset [LeCun et al., 1998] to demonstrate the addition of two (visual) numbers. We randomly pick 90 combinations of two digits for training (out of the 100 total combinations), and then we use the rest 10 for extrapolation set. Specifically, in our three-fold cross-validation, we randomly pick up 90 combinations of two digits and we sample 2000

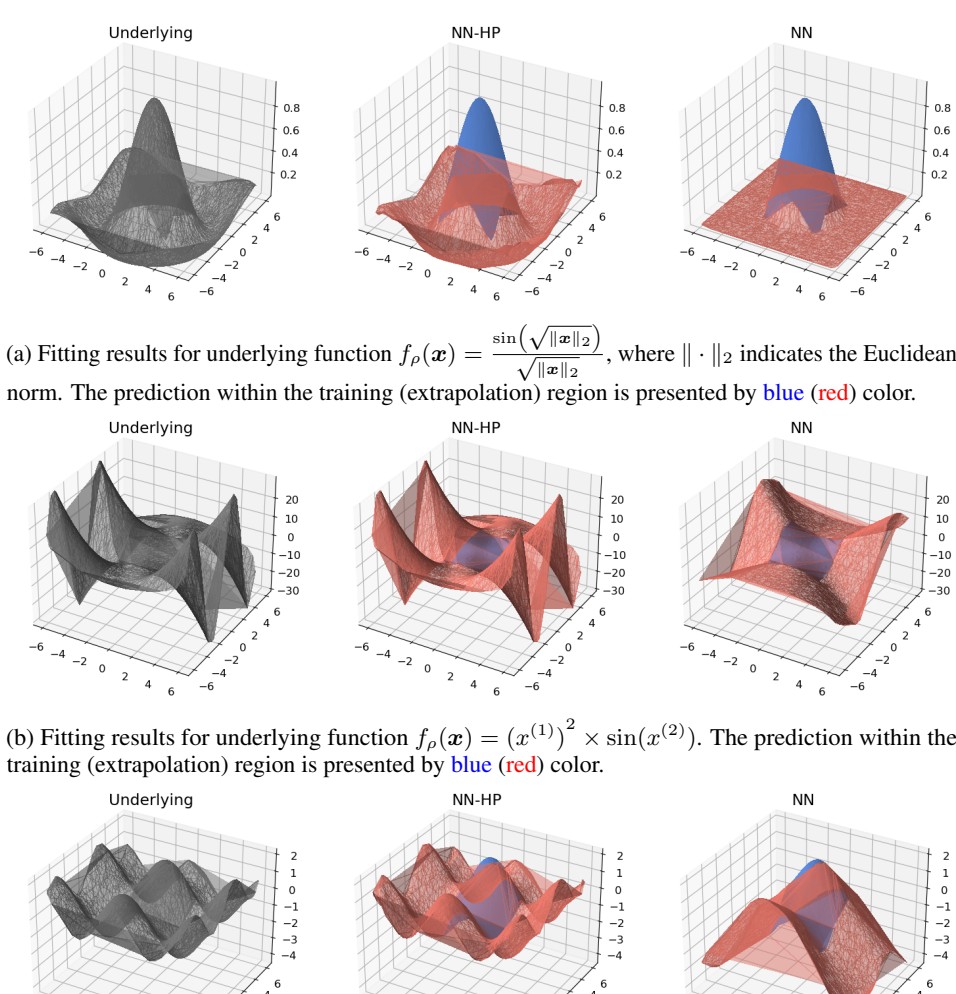

(a) Fitting results for underlying function $f_\rho(\boldsymbol{x}) = \dfrac{\sin\left(\sqrt{\|\boldsymbol{x}\|_2}\right)}{\sqrt{\|\boldsymbol{x}\|_2}}$, where $\|\cdot\|_2$ indicates the Euclidean norm. The prediction within the training (extrapolation) region is presented by blue (red) color.

(b) Fitting results for underlying function $f_\rho(\boldsymbol{x}) = \left(x^{(1)}\right)^2 \times \sin(x^{(2)})$. The prediction within the training (extrapolation) region is presented by blue (red) color.

(c) Fitting results for the underlying function $f_\rho(\boldsymbol{x}) = \cos(x^{(1)}) + \sin(x^{(2)})$. The prediction within the training (extrapolation) region is presented by blue (red) color.

Figure 6: This figure shows the results of fitting several analytically-known two-variable functions. We can see even though both NNs-Hp and NNs can learn well in the training region, NNs-Hp is much more flexible than standard NNs during extrapolation.

pairs for each combination to construct the training set. There are $90 \times 2000$ pairs in the training set and $10 \times 2000$ pairs in the testing set. Each network is trained for 100 epochs with batch size 128. The width of the networks is 256. Each network is trained with squared-loss for 3 runs.

## E.4 Experimental setup in learning harmonics

This section describes the experimental setup corresponding to Section 4.3. We follow the setup in Cao et al. [2019], Choraria et al. [2022]. The number of sample points is 1000. The width of the network is 32768. The network is trained for 30000 iterations and optimized via stochastic gradient descent with learning rate 0.0016.

## E.5 Visual analogy task

In this section, we scrutinize the extrapolation capability of NNs-Hp on the visual analogy task on VAEC dataset Webb et al. [2020]. For each pair of four images $A, B, C, D$, the proportional analogy problem is in the form $A : B :: C : D$ based on the brightness, size, and 2-D location. The model is required to select the correct $D$ among several candidates when given $A, B, C$. We conduct the scale extrapolation experiment introduced in the paper as it's similar to our experiment on the variation of brightness, which treats the scale factor $\alpha$ as the extent of extrapolation. $\alpha = 1$ indicates the training set. $\alpha \in \{2, ..., 6\}$ indicates the extrapolation set, where the values of the dataset are multiplied by a scale factor $\alpha$ ranging from 2 to 6. We use the original best model in the paper as baseline (NNs) and insert Hadamard product as NNs-Hp to compare. Apart from the network architecture, the training details are the same as in Webb et al. [2020]. We run each method 8 times and report the mean of accuracy in Table 3. Results show that both models achieve similar performance in the training regime while NNs-HP extrapolates better than standard NNs in most regimes.

Table 3: Experimental results in the task of visual analogy on VAEC dataset. 'Ext' abbreviates 'extrapolation'. We can see that NNs-HP has better extrapolation performance in most extrapolation regimes.

|  | Training ($\alpha = 1$) | Ext ($\alpha = 2$) | Ext ($\alpha = 3$) | Ext ($\alpha = 4$) | Ext ($\alpha = 5$) | Ext ($\alpha = 6$) |
|---|---|---|---|---|---|---|
| NNs | 99.5% | **76.2%** | 55.5% | 46.3% | 42.6% | 40.4% |
| NNs-Hp | 99.7% | 73.7% | **57.5%** | **49.0%** | **45.3%** | **42.9%** |

## E.6 Spectral bias in image classification

This experiment studies how the frequency of the noise affects the validation performance in image classification, which further validates our theoretical result on spectral bias in Section 3.3. Specifically, we follow the standard set up in Rahaman et al. [2019]. we consider a binary classification task with labels 3 and 8 on the MNIST dataset. We add noises with different frequencies to the label. We test NNs-Hp with three, six, and nine-degree multiplicative interaction and compare it with the corresponding standard fully-connected neural networks. Both networks are optimized through Adam. We select mean squared loss as the criterion and choose learning rate 0.0001. We train each network for 1000 iterations. The width of the network is 256. The results in Figure 7 present the 'dip' of validation mean squared error (MSE) during the process of training. In the comparison of each order, for instance, in Figure 7a we can see that in the case of higher frequencies, e.g., 0.3 and 0.5, the validation dips of NNs-Hp are apparently smaller than that of NNs in the early stage of during training. The reason is that NNs-Hp can speed up the learning of high-frequency information based on Section 3.3.

## F Additional result on multiplicative filter networks

Multiplicative filter network (MFN) is another instance of NNs-Hp which inserts the Hadamard between the sinusoidal or Gabor wavelet functions among each layer [Fathony et al., 2021]. MFN has demonstrated stronger performance over standard neural networks in several representation tasks.

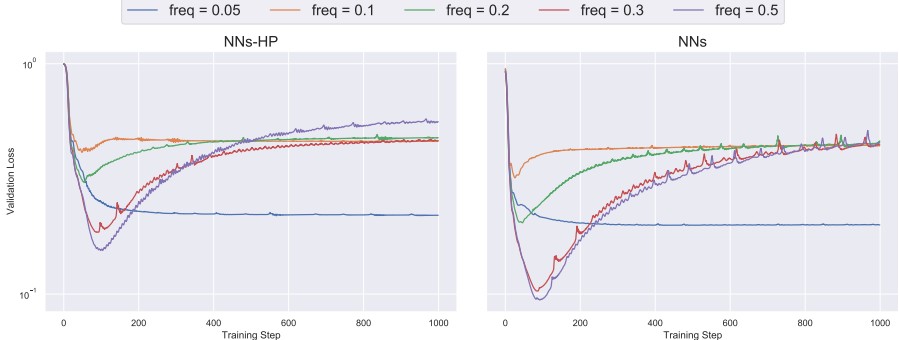

(a) Results of NNs-Hp with three-degree multiplicative interaction and the corresponding NNs.

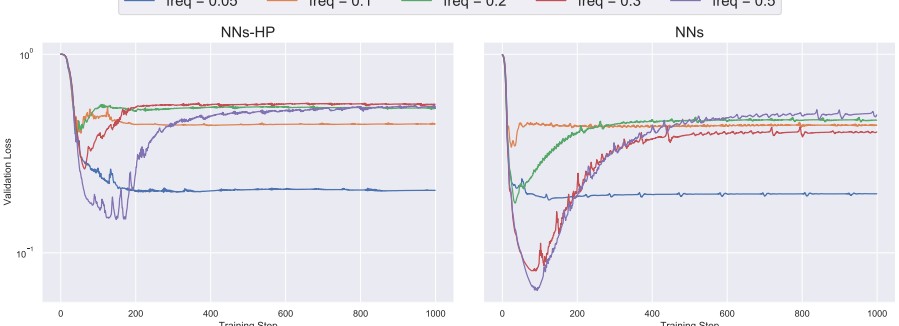

(b) Results of NNs-Hp with six-degree multiplicative interaction and the corresponding NNs.

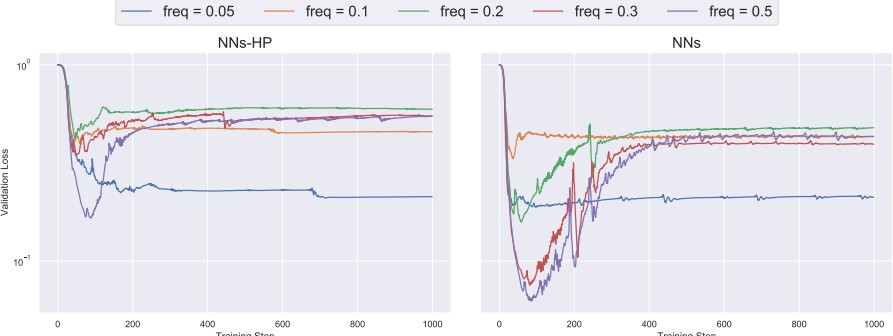

(c) Results of NNs-Hp with nine-degree multiplicative interaction and the corresponding NNs.

Figure 7: The above figures show the dip of validation loss of NNs-Hp and NNs during training. Since NNs-Hp is able to speed up learning high-frequency information, we can see that for high-frequency noise, such dip for NNs-Hp is smaller than that of NNs at the early stage of training.

### F.1 Theoretical analysis

In this section, we will derive the neural tangent kernel of MFN and then analyze the extrapolation behavior of MFN. We consider the following MFN:

$$\boldsymbol{y}_1 = \sqrt{\frac{2}{m}}\sin(\boldsymbol{W}_1\boldsymbol{x}), \ \ f(\boldsymbol{x}) = \sqrt{\frac{2}{m}}(\boldsymbol{W}_{N+1}\boldsymbol{y}_N), \ \ \boldsymbol{y}_n = \sqrt{\frac{2}{m}}\sin(\boldsymbol{W}_n\boldsymbol{x}) * \boldsymbol{y}_n, \ n = 2, \ldots, N \,,$$

where each element in $\boldsymbol{W}_{N+1} \in \mathbb{R}^{1 \times m}$ and $\boldsymbol{W}_n \in \mathbb{R}^{m \times d}$, for $n = 1, \ldots, N$ is independently sampled from $\mathcal{N}(0, 1)$. Note that we multiply by the scaling factor $\sqrt{\frac{2}{m}}$ after each degree to ensure that the norm of the network output is preserved at initialization with infinite-width setting. In Lemma 9 we develop the NTK $K(\boldsymbol{x}, \boldsymbol{x}')$ of MFN.

**Lemma 9.** *The neural tangent kernel of the MFN has the following form:*

$$K(\boldsymbol{x}, \boldsymbol{x}') = 2N \cdot \langle \boldsymbol{x}, \boldsymbol{x}' \rangle \kappa_3(\boldsymbol{x}, \boldsymbol{x}')(\kappa_4(\boldsymbol{x}, \boldsymbol{x}'))^{N-1} + 2(\kappa_4(\boldsymbol{x}, \boldsymbol{x}'))^N, \tag{66}$$

*where $\kappa_3$ and $\kappa_4$ are defined by taking the random Gaussian vector $\boldsymbol{w} \in \mathbb{R}^d$,*

$$\kappa_3 = \mathbb{E}_{\boldsymbol{w} \sim \mathcal{N}(\mathbf{0}, \sqrt{\frac{2}{m}} \cdot \boldsymbol{I})} \left( \cos(\boldsymbol{w}^\top \boldsymbol{x}) \cdot \cos(\boldsymbol{w}^\top \boldsymbol{x}') \right), \kappa_4 = \mathbb{E}_{\boldsymbol{w} \sim \mathcal{N}(\mathbf{0}, \sqrt{\frac{2}{m}} \cdot \boldsymbol{I})} \left( \sin(\boldsymbol{w}^\top \boldsymbol{x}) \cdot \sin(\boldsymbol{w}^\top \boldsymbol{x}') \right).$$

*Proof of Lemma 9.* We will compute the gradient with respect to each weight and then sum up the inner products to obtain the NTK. Below, we denote by $\tilde{\boldsymbol{\alpha}}_n = \boldsymbol{W}_n \boldsymbol{x}, n \in [N]$. Firstly, we compute the contribution to the NTK w.r.t $\boldsymbol{W}_1$, its corresponding derivative is as follows:

$$\partial_{\boldsymbol{W}_1} f(\boldsymbol{x}) = \sqrt{\frac{2}{m}} \left[ \boldsymbol{W}_{N+1}^\top \left( \prod_{n=2}^N \mathrm{Diag} \left( \sin \left( \sqrt{\frac{2}{m}} \tilde{\boldsymbol{\alpha}}_n(\boldsymbol{x}) \right) \right) \right) \cos \left( \sqrt{\frac{2}{m}} \tilde{\boldsymbol{\alpha}}_1(\boldsymbol{x}) \right) \right]^\top (\partial_{\boldsymbol{W}_1} \tilde{\boldsymbol{\alpha}}_1(\boldsymbol{x}))^\top$$

$$= \sqrt{\frac{2}{m}} \left[ \boldsymbol{W}_{N+1}^\top \left( \prod_{n=2}^N \mathrm{Diag} \left( \sin \left( \sqrt{\frac{2}{m}} \tilde{\boldsymbol{\alpha}}_n(\boldsymbol{x}) \right) \right) \right) \cos \left( \sqrt{\frac{2}{m}} \tilde{\boldsymbol{\alpha}}_1(\boldsymbol{x}) \right) \right]^\top \boldsymbol{x}^\top$$

where $\mathrm{Diag}(\cdot)$ converts a vector to a diagonal matrix. The inner product follows that:

$$\langle \partial_{\boldsymbol{W}_1} f(\boldsymbol{x}), \partial_{\boldsymbol{W}_1} f(\boldsymbol{x}') \rangle$$

$$= \frac{2}{m} \sum_{j=1}^m W_{N+1}^{(j)} W_{N+1}^{(j)} \left( \prod_{n=2}^N \left( \frac{2}{m} \sin \left( \tilde{\alpha}_n^{(j)}(\boldsymbol{x}) \right) \sin \left( \tilde{\alpha}_n^{(j)}(\boldsymbol{x}') \right) \right) \right) \left( \frac{2}{m} \cos \left( \tilde{\alpha}_1^{(j)}(\boldsymbol{x}) \right) \cos \left( \tilde{\alpha}_1^{(j)}(\boldsymbol{x}') \right) \right) \boldsymbol{x}^\top \boldsymbol{x}'. \tag{67}$$

By the law of large numbers, we obtain:

$$\lim_{m \to \infty} \langle \partial_{\boldsymbol{W}_1} f(\boldsymbol{x}), \partial_{\boldsymbol{W}_1} f(\boldsymbol{x}') \rangle = 2 \langle \boldsymbol{x}, \boldsymbol{x}' \rangle \kappa_3(\boldsymbol{x}, \boldsymbol{x}')(\kappa_4(\boldsymbol{x}, \boldsymbol{x}'))^{N-1}. \tag{68}$$

Since the formula of the network is symmetric w.r.t $\{\boldsymbol{W}_i\}_{i=1}^N$, the contributions of $\{\boldsymbol{W}_i\}_{i=1}^N$ to the NTK are the same, we can trivially multiply Eq. (68) by $N$.

Next, we will compute the contribution to the NTK w.r.t $\boldsymbol{W}_{N+1}$, its corresponding derivative is as follows:

$$\partial_{\boldsymbol{W}_{N+1}} f(\boldsymbol{x}) = \sqrt{\frac{2}{m}} \left( \sqrt{\frac{2}{m}} \sin(\tilde{\boldsymbol{\alpha}}_N) * \ldots * \sqrt{\frac{2}{m}} \sin(\tilde{\boldsymbol{\alpha}}_1) \right).$$

The inner product follows that:

$$\langle \partial_{\boldsymbol{W}_{N+1}} f(\boldsymbol{x}), \partial_{\boldsymbol{W}_{N+1}} f(\boldsymbol{x}') \rangle = \frac{2}{m} \sum_{j=1}^m \left( \prod_{n=1}^N \left( \frac{2}{m} \sin \left( \tilde{\alpha}_n^{(j)}(\boldsymbol{x}) \right) \sin \left( \tilde{\alpha}_n^{(j)}(\boldsymbol{x}') \right) \right) \right).$$

By the law of large numbers:

$$\lim_{m \to \infty} \langle \partial_{\boldsymbol{W}_{N+1}} f(\boldsymbol{x}), \partial_{\boldsymbol{W}_{N+1}} f(\boldsymbol{x}') \rangle$$
$$= 2 \cdot (\kappa_4(\boldsymbol{x}, \boldsymbol{x}'))^N. \tag{69}$$

The proof is completed by multiplying Eq. (68) by $N$ and adding by Eq. (69). $\square$

The derived kernel enables us to study how MFN trained by gradient descent extrapolates.

**Theorem 8.** *Suppose we train MFN with $N$-degree multiplicative interaction with infinite-width on $\{(\boldsymbol{x}_i, y_i)\}_{i=1}^{|\mathcal{X}|}$, and the network is optimized with squared loss in the NTK regime. For any direction $\boldsymbol{v} \in \mathbb{R}^d$ that satisfies $\|\boldsymbol{v}\|_2 = \max\{\|\boldsymbol{x}_i\|^2\}$, let $\boldsymbol{x}_0 = t\boldsymbol{v}$ and $\boldsymbol{x} = \boldsymbol{x}_0 + h\boldsymbol{v}$ with $t > 1$ and $h > 0$ be the extrapolation data points, the output $f(\boldsymbol{x}_0 + h\boldsymbol{v})$ can extrapolate to $\mathrm{poly} = (\sin(\alpha t), \cos(\alpha t), t)$, where $\alpha$ is constant and the order of $\sin(\alpha t)$ is up to $N$, the order of $\cos(\alpha t)$ and $t$ is one.*

*Proof of Theorem 8.* A specific feature map $\phi(\boldsymbol{x})$ induced by the NTK of MFN is

$$\phi(\boldsymbol{x}) = \left( c'\boldsymbol{x} \cdot \cos\langle \boldsymbol{w}, \boldsymbol{x}\rangle \cdot \sin(\langle\boldsymbol{w},\boldsymbol{x}\rangle)^{N-1}, c''\sin(\langle\boldsymbol{w},\boldsymbol{x}\rangle)^N \right), \tag{70}$$

where $\boldsymbol{w}$ is sampled from $\mathcal{N}(\boldsymbol{0}, \boldsymbol{I})$, $c'$ and $c''$ are constants. Note that kernel regression solution is equivalent to the following form:

$$f(\boldsymbol{x}) = \boldsymbol{\beta}^\top \phi(\boldsymbol{x}), \tag{71}$$

where the representation coefficient $\boldsymbol{\beta}$ holds:

$$\min_{\boldsymbol{\beta}'} \|\boldsymbol{\beta}'\|_2$$
$$\text{s.t.} \quad \phi(\boldsymbol{x}_i)^\top \boldsymbol{\beta}' = y_i, \quad i = 1, \dots, |\mathcal{X}|. \tag{72}$$

Therefore, given inputs $\boldsymbol{x}_0 = t\boldsymbol{v}$ and $\boldsymbol{x} = \boldsymbol{x}_0 + h\boldsymbol{v}$, we have:

$$\begin{aligned}
f(\boldsymbol{x}) - f(\boldsymbol{x}_0) &= \boldsymbol{\beta}^\top \left( \phi((t+h)\boldsymbol{v}) - \phi(t\boldsymbol{v}) \right) \\
&= \boldsymbol{\beta}_1^\top \left( c'(t+h)\boldsymbol{v} \cdot \cos\langle\boldsymbol{w}, (t+h)\boldsymbol{v}\rangle \cdot \sin(\langle\boldsymbol{w}, (t+h)\boldsymbol{v}\rangle)^{N-1} \right) \\
&\quad - \boldsymbol{\beta}_1^\top \left( c't\boldsymbol{v} \cdot \cos\langle\boldsymbol{w}, t\boldsymbol{v}\rangle \cdot \sin(\langle\boldsymbol{w}, t\boldsymbol{v}\rangle)^{N-1} \right) \\
&\quad + \beta_2 \left( c''\sin(\langle\boldsymbol{w}, (t+h)\boldsymbol{v}\rangle)^N \right) - \beta_2 \left( c''\sin(\langle\boldsymbol{w}, t\boldsymbol{v}\rangle)^N \right),
\end{aligned}$$

Therefore, the network can extrapolate to poly $= (\sin(\alpha t), \cos(\alpha t), t)$, where $\alpha$ is constant and the order of $\sin(\alpha t)$ is up to $N$. This completes the proof. □

### F.2 Numerical result

In this section, we provide additional experimental results on extrapolation with MFN. Firstly, we follow the setup in Appendix E.5 and present the result on VAEC dataset as follows, where we can see that We can see that MFN has better extrapolation performance than standard NN in most extrapolation regimes.

Table 4: Experimental results in the task of visual analogy on VAEC dataset. 'Ext' abbreviates 'extrapolation'.

|  | Training ($\alpha=1$) | Ext ($\alpha=2$) | Ext ($\alpha=3$) | Ext ($\alpha=4$) | Ext ($\alpha=5$) | Ext ($\alpha=6$) |
|---|---|---|---|---|---|---|
| NN | 99.5% | **76.2%** | 55.5% | 46.3% | 42.6% | 40.4% |
| MFN | 99.1% | 74.5% | **56.2**% | **47.1**% | **43.0**% | **40.6**% |

Next, we apply MFN in the task of arithmetic extrapolation, as introduced in Section 4.2, and follow the same experimental setup. The following result further showcases the improvement of MFN over standard NN.

Table 5: Results with MFN and standard NN in the task of arithmetic extrapolation.

| Method |  | Rounding | Floor/ceiling | ±1 |
|---|---|---|---|---|
| NN(Dense) | Interpolation | 0.980 | 0.999 | 0.999 |
|  | Extrapolation | 0.436 | 0.805 | 0.887 |
| MFN (Dense) | Interpolation | 0.996 | 0.997 | 0.999 |
|  | Extrapolation | **0.720** | **0.874** | **0.916** |
| NN(Conv) | Interpolation | 0.945 | 0.983 | 0.994 |
|  | Extrapolation | 0.617 | 0.918 | 0.953 |
| MFN (Conv) | Interpolation | 0.947 | 0.996 | 0.999 |
|  | Extrapolation | **0.824** | **0.925** | **0.954** |

## G   Additional result on non-local networks with Hadamard product

Non-local networks have demonstrated stellar performance in capturing long-range dependencies of the input signals [Wang et al., 2018]. Particularly, Poly-NL is one of the non-local networks that

utilize three-degree polynomial to reduce the complexity of traditional non-local networks from quadratic to linear. Note that the formula in [Babiloni et al., 2021] is based on standard polynomial expansion, which we have analyzed in the main body. To make our analysis more general, we consider the following single Poly-NL block that uses Softmax as activation function:

$$\boldsymbol{y}_1 = \boldsymbol{x}, \quad \boldsymbol{y}_2 = \sigma(\boldsymbol{W}_1 \boldsymbol{y}_1),$$
$$\boldsymbol{y}_3 = \text{Softmax}\{(\boldsymbol{w}_Q \boldsymbol{y}_2^\top) * (\boldsymbol{w}_K \boldsymbol{y}_2^\top)\}(\boldsymbol{y}_2 w_V),$$
$$f(\boldsymbol{x}) = \sqrt{\frac{2}{m}}(\boldsymbol{w}_2^\top \boldsymbol{y}_3),$$

where $\boldsymbol{x} \in \mathbb{R}^d$, $\boldsymbol{W}_1 \in \mathbb{R}^{m \times d}$, $\boldsymbol{w}_Q \in \mathbb{R}^m$, $\boldsymbol{w}_K \in \mathbb{R}^m$, $w_V \in \mathbb{R}$, $\boldsymbol{w}_2 \in \mathbb{R}^m$, each element in the weight is independently sampled from $\mathcal{N}(0, 1)$, the Softmax is row-wise. Firstly, we give the neural tangent kernel of the Poly-NL, wherein we only train the weight $w_Q$ and $w_K$.

**Lemma 10.** *The neural tangent kernel of the Poly-NL has the following form:*

$$K(\boldsymbol{x}, \boldsymbol{x}') = 4 \cdot \mathbb{E}_{w_3, w_4 \sim \mathcal{N}(0,1)} \boldsymbol{y}_2^\top \left(Diag(\boldsymbol{\tau}) - \boldsymbol{\tau}\boldsymbol{\tau}^\top\right)(\boldsymbol{y}_2 * \boldsymbol{y}_2) \boldsymbol{y}_2'^\top \left(Diag(\boldsymbol{\tau}') - \boldsymbol{\tau}'\boldsymbol{\tau}'^\top\right)(\boldsymbol{y}_2' * \boldsymbol{y}_2'),$$

*where we denote by $\boldsymbol{\tau} = Softmax(w_3 w_4 (\boldsymbol{y}_2 * \boldsymbol{y}_2))$, $\boldsymbol{\tau}' = Softmax(w_3 w_4 (\boldsymbol{y}_2' * \boldsymbol{y}_2'))$, $\boldsymbol{y}_2 = \sigma(\boldsymbol{W}_1 \boldsymbol{x})$, $\boldsymbol{y}_2' = \sigma(\boldsymbol{W}_1 \boldsymbol{x}')$, where $w_3$ and $w_4$ is independently sampled from $\mathcal{N}(0, 1)$.*

*Proof of Lemma 10.* Firstly, we compute the Jacobian with respect to $\boldsymbol{w}_Q$:

$$\partial_{\boldsymbol{w}_Q} f(\boldsymbol{x}) = \sqrt{\frac{2}{m}} \frac{\partial \left(\sum_{i=1}^m w_2^{(i)} y_3^{(i)}\right)}{\partial \boldsymbol{w}_Q}$$
$$= \sqrt{\frac{2}{m}} \sum_{i=1}^m w_2^{(i)} w_V \boldsymbol{y}_2^\top \frac{\partial \text{Softmax}\left(w_Q^{(i)} w_K^{(i)} (\boldsymbol{y}_2 * \boldsymbol{y}_2)\right)}{\partial \boldsymbol{w}_Q}$$
$$= \sqrt{\frac{2}{m}} \sum_{i=1}^m w_2^{(i)} w_K^{(i)} w_V \boldsymbol{y}_2^\top \left(\text{Diag}(\boldsymbol{\varphi}_i) - \boldsymbol{\varphi}_i \boldsymbol{\varphi}_i^\top\right)(\boldsymbol{y}_2 * \boldsymbol{y}_2) \boldsymbol{e}_i^\top,$$

where we denote by $\boldsymbol{\varphi}_i = \text{Softmax}\left(w_Q^{(i)} w_K^{(i)} (\boldsymbol{y}_2 * \boldsymbol{y}_2)\right) \in \mathbb{R}^m$. Next, in order to obtain the NTK, we calculate the inner product of the Jacobian:

$$\langle \partial_{\boldsymbol{w}_Q} f(\boldsymbol{x}), \partial_{\boldsymbol{w}_Q} f(\boldsymbol{x}') \rangle$$
$$= \frac{2}{m} \sum_{i=1}^m (w_2^{(i)} w_K^{(i)} w_V)^2 \boldsymbol{y}_2^\top \left(\text{Diag}(\boldsymbol{\varphi}_i) - \boldsymbol{\varphi}_i \boldsymbol{\varphi}_i^\top\right)(\boldsymbol{y}_2 * \boldsymbol{y}_2) \boldsymbol{y}_2'^\top \left(\text{Diag}(\boldsymbol{\varphi}'^i) - \boldsymbol{\varphi}'^i \boldsymbol{\varphi}'^{i^\top}\right)(\boldsymbol{y}_2' * \boldsymbol{y}_2')$$

By the law of large numbers, as $m \to \infty$, we obtain:

$$\lim_{m \to \infty} \langle \partial_{\boldsymbol{w}_Q} f(\boldsymbol{x}), \partial_{\boldsymbol{w}_Q} f(\boldsymbol{x}') \rangle$$
$$= 2 \cdot \mathbb{E}_{w_3, w_4 \sim \mathcal{N}(0,1)} \boldsymbol{y}_2^\top \left(\text{Diag}(\boldsymbol{\tau}) - \boldsymbol{\tau}\boldsymbol{\tau}^\top\right)(\boldsymbol{y}_2 * \boldsymbol{y}_2) \boldsymbol{y}_2'^\top \left(\text{Diag}(\boldsymbol{\tau}') - \boldsymbol{\tau}'\boldsymbol{\tau}'^\top\right)(\boldsymbol{y}_2' * \boldsymbol{y}_2'), \quad (73)$$

where we denote by $\boldsymbol{\tau} = \text{Softmax}(w_3 w_4 (\boldsymbol{y}_2 * \boldsymbol{y}_2))$. Since the weight $\boldsymbol{w}_Q$ and $\boldsymbol{w}_K$ are symmetric in the formula of Poly-NL, the proof is completed by multiplying Eq. (73) by two. $\square$

Now we are ready to analyze the extrapolation behaviour of Poly-NL.

**Theorem 9.** *Suppose we train Poly-NL with infinite-width on $\{(\boldsymbol{x}_i, y_i)\}_{i=1}^{|\mathcal{X}|}$, and the network is optimized with squared loss in the NTK regime. For any direction $\boldsymbol{v} \in \mathbb{R}^d$ that satisfies $\|\boldsymbol{v}\|_2 = \max\{\|\boldsymbol{x}_i\|^2\}$, let $\boldsymbol{x}' = t\boldsymbol{v}$ and $\boldsymbol{x} = \boldsymbol{x}' + h\boldsymbol{v}$ with $t > 1$ and $h > 0$ be the extrapolation data points, then for $\delta \in (0, 1)$ and some constant $C$, when $m \geq 2 \ln(2/\delta) + d + \sqrt{8d \ln(2/\delta)}$, with probability at least $1 - \delta$, we have:*
$$|f(\boldsymbol{x}) - f(\boldsymbol{x}')| \leq C t^3 h^3 m^{\frac{3}{2}} \|\boldsymbol{v}\|^3.$$

*Proof of Theorem 9.* We first bound the spectral norm of the weight matrix $\boldsymbol{W}_1$.

**Lemma 11.** *Based on the randomness of the weight $\boldsymbol{W}_1$, for $\delta \in (0,1)$, when $m \geq 2\ln(2/\delta) + d + \sqrt{8d\ln(2/\delta)}$, with probability at least $1 - \delta$, we have $\|\boldsymbol{W}_1\| \leq 2\sqrt{m}$.*

We can choose a certain feature map $\phi(\boldsymbol{x})$ induced by the NTK in Lemma 10 is

$$\phi(\boldsymbol{x}) = \left(\boldsymbol{y}_2^\top \left(\mathrm{Diag}(\boldsymbol{\tau}) - \boldsymbol{\tau}\boldsymbol{\tau}^\top\right)(\boldsymbol{y}_2 * \boldsymbol{y}_2)\right), \tag{74}$$

where $\widetilde{\boldsymbol{\tau}} = \mathrm{Softmax}(w_5 w_6 \,(\boldsymbol{y}_2 * \boldsymbol{y}_2))$, $w_5, w_6$ are iid sampled from $\mathcal{N}(0,1)$, $c'$ and $c''$ are constants. Similarly, using the solution of kernel regression, we can calculate the output of the network as follows. Given $\boldsymbol{x}' = t\boldsymbol{v}$ and $\boldsymbol{x} = \boldsymbol{x}' + h\boldsymbol{v}$, we add the prime symbol to the variable in the network associated to the input $\boldsymbol{x}'$, we have:

$$\begin{aligned}
|f(\boldsymbol{x}) - f(\boldsymbol{x}')| &= |\beta\left(\phi((t+h)\boldsymbol{v}) - \phi(t\boldsymbol{v})\right)| \\
&\leq |\beta|\|\boldsymbol{y}_2\|\|\left(\mathrm{Diag}(\boldsymbol{\tau}) - \boldsymbol{\tau}\boldsymbol{\tau}^\top\right)\|\|\boldsymbol{y}_2 * \boldsymbol{y}_2 - \boldsymbol{y}_2' * \boldsymbol{y}_2'\| \\
&\quad + |\beta|\|\boldsymbol{y}_2' * \boldsymbol{y}_2'\|\|\boldsymbol{y}_2\|\left\|\left(\mathrm{Diag}(\boldsymbol{\tau}) - \boldsymbol{\tau}\boldsymbol{\tau}^\top\right) - \left(\mathrm{Diag}(\boldsymbol{\tau}') - \boldsymbol{\tau}'\boldsymbol{\tau}'^\top\right)\right\| \\
&\quad + |\beta|\|\boldsymbol{y}_2' * \boldsymbol{y}_2'\|\left\|\left(\mathrm{Diag}(\boldsymbol{\tau}') - \boldsymbol{\tau}'\boldsymbol{\tau}'^\top\right)\right\|\|\boldsymbol{y}_2 - \boldsymbol{y}_2'\|.
\end{aligned} \tag{75}$$

We start by bounding the first term in Eq. (75). For $\delta \in (0,1)$, when $m \geq 2\ln(2/\delta) + d + \sqrt{8d\ln(2/\delta)}$, with probability at least $1 - \delta$, we have:

$$\begin{aligned}
&|\beta|\|\boldsymbol{y}_2\|\|\left(\mathrm{Diag}(\boldsymbol{\tau}) - \boldsymbol{\tau}\boldsymbol{\tau}^\top\right)\|\|\boldsymbol{y}_2 * \boldsymbol{y}_2 - \boldsymbol{y}_2' * \boldsymbol{y}_2'\| \\
&\leq |\beta|\|\boldsymbol{y}_2\|\left(\|\mathrm{Diag}(\boldsymbol{\tau})\| + \|\boldsymbol{\tau}\boldsymbol{\tau}^\top\|\right)\left(\|\boldsymbol{y}_2\| + \|\boldsymbol{y}_2'\|\right)\left(\|\boldsymbol{y}_2 - \boldsymbol{y}_2'\|\right) \\
&\leq 2|\beta|\|\boldsymbol{W}_1(t+h)\boldsymbol{v}\|\left(\|(t+h)\boldsymbol{W}_1\boldsymbol{v}\| + \|t\boldsymbol{W}_1\boldsymbol{v}\|\right)\left(\|h\boldsymbol{W}_1\boldsymbol{v}\|\right) \\
&\leq 4|\beta|(t+h)\sqrt{m}\|\boldsymbol{v}\|\left(2\sqrt{m}(t+h)\|\boldsymbol{v}\| + 2\sqrt{m}t\|\boldsymbol{v}\|\right)\left(2\sqrt{m}h\|\boldsymbol{v}\|\right) \\
&= 16|\beta|m^{\frac{3}{2}}\|\boldsymbol{v}\|^3\left(2t^2h + 3th^2 + h^3\right).
\end{aligned} \tag{76}$$

where the first inequality comes from triangle inequality, the second inequality is due to the fact that the output of softmax ranges from zero to one, and the 1-Lipschitz of ReLU. Next, we bound the second term in Eq. (75). Similarly, by Lemma 11, over the same randomness of the weight $\boldsymbol{W}_1$, with probability at least $1 - \delta$, we have:

$$\begin{aligned}
&|\beta|\|\boldsymbol{y}_2' * \boldsymbol{y}_2'\|\|\boldsymbol{y}_2\|\left\|\left(\mathrm{Diag}(\boldsymbol{\tau}) - \boldsymbol{\tau}\boldsymbol{\tau}^\top\right) - \left(\mathrm{Diag}(\boldsymbol{\tau}') - \boldsymbol{\tau}'\boldsymbol{\tau}'^\top\right)\right\| \\
&\leq |\beta|\|\boldsymbol{y}_2'\|^2\|\boldsymbol{y}_2\|\left\|(\mathrm{Diag}(\boldsymbol{\tau} - \boldsymbol{\tau}')\| + \|\boldsymbol{\tau}\boldsymbol{\tau}^\top - \boldsymbol{\tau}'\boldsymbol{\tau}'^\top\|\right) \\
&\leq 4|\beta|mt^2\|\boldsymbol{v}\|^2 \times 2\sqrt{m}(t+h)\|\boldsymbol{v}\| \times 4 = 32|\beta|m^{\frac{3}{2}}\|\boldsymbol{v}\|^3(t^3 + t^2h).
\end{aligned} \tag{77}$$

Next, we bound the third term in Eq. (75). Similarly, by Lemma 11, over the same randomness of the weight $\boldsymbol{W}_1$, with probability at least $1 - \delta$, we have:

$$\begin{aligned}
&|\beta|\|\boldsymbol{y}_2' * \boldsymbol{y}_2'\|\left\|\left(\mathrm{Diag}(\boldsymbol{\tau}') - \boldsymbol{\tau}'\boldsymbol{\tau}'^\top\right)\right\|\|\boldsymbol{y}_2 - \boldsymbol{y}_2'\| \\
&\leq |\beta|\|\boldsymbol{y}_2'\|^2\left(\|\mathrm{Diag}(\boldsymbol{\tau}')\| + \|\boldsymbol{\tau}'\boldsymbol{\tau}'^\top\|\right)\|\boldsymbol{y}_2 - \boldsymbol{y}_2'\| \\
&\leq 4mh^2|\beta|\|\boldsymbol{v}\|^2 \times 2 \times 2\sqrt{m}h\|\boldsymbol{v}\| = 16|\beta|m^{\frac{3}{2}}\|\boldsymbol{v}\|^3h^3.
\end{aligned} \tag{78}$$

Therefore, the proof is completed by summing up Eq. (76) to (78). $\qquad\square$

# H   Societal impact

This work studies a cutting-edge network architecture, i.e., neural network with Hadamard product (NN-Hp), from a theoretical perspective. The analysis of the corresponding NTK lays a theoretical foundation for the interested practitioner to further study other priorities of NN-Hp such as convergence and generalization. Furthermore, our current analysis mainly focuses on the theoretical side of extrapolation. We believe our insight and empirical evidence in extrapolation will allow the investigation of other more complicated OOD problems among the ML community, such as domain adaption and invariant learning. Therefore, we do not expect any negative societal bias from this work.

# I    Limitations

In this work, we illustrate how our theory can be applicable in a variety of experimental settings, especially on extrapolation. Nevertheless, we do not focus explicitly on obtaining state-of-the-art numerical results in real-world applications, which could be one limitation of this work.

Our proof framework is based on NTK for understanding theoretical properties of neural networks. However, NTK still works in "linear" regime [Lee et al., 2019, Woodworth et al., 2020], which appears difficult to fully demonstrate the success of practical neural networks. Nevertheless, this is a common limitation of NTK-based analysis in the community.