# OpenReview forum: "Extrapolation and Spectral Bias of Neural Nets with Hadamard Product: a Polynomial Net Study"
_NeurIPS.cc/2022/Conference — NeurIPS 2022 Accept_

### Official Review · Reviewer_cA3P · 2022-07-09

**Rating:** 7
**Confidence:** 4
**Soundness:** 3 good
**Presentation:** 4 excellent
**Contribution:** 3 good

**Summary:**

This paper derives the neural tangent kernel (NTK) of neural networks "with hadamard products" (NN-Hp), and proves several related results:
1) convergence at initialization and during training on least squared regression;
2) extrapolation outside the domain of training data (OOD);
3) decay of the eigenspectrum of the kernel.

NN-Hp are basically deep fully connected ReLU neural networks in which the output of each layer is multiplied (element-wise) by the output of another layer processing the input.
This kind of architecture is used e.g. in Style-GAN, and is compared with standard deep fully connected ReLU neural networks (NN).
The paper proves that NN-Hp, in contrast to NN, extrapolates non-linearly for OOD data, and that the NTK spectrum of NN-Hp decays more slowly that that of NN, implying a prominence of higher frequencies.
Experiments are shown in support of the theoretical results.


**Questions:**

Is it possible to test standard NN on the experiment of Figure 5 (in addition to NN-Hp)?
Comparing NN and NN-Hp on the problem of fitting harmonics at different frequencies would provide support for Theorem 6, which is currently untested.
As far as I understood the eigenfunctions are identical for the kernels of NN and NN-Hp, only the eigenvalues differ, therefore the comparison would be fair.

**Limitations:**

I could not find any description of the limitations of this work, even though the checklist says "yes"

**Strengths And Weaknesses:**

Strengths:
- The paper is very well written and clear.
- This problem is interesting, it fills a hole in the literature.
- The coverage of the topic is substantial, the density of results in the paper is high.

Weaknesses:
- I was a bit disappointed that little discussion was given about the result on the eigenspectrum. What does that imply?

---

> ### Author Response · Authors · 2022-08-02
> **Response to reviewer cA3P**
>
> We are thankful to the reviewer cA3P  for their insightful feedback and appreciation of our work. The concerns are addressed below.
>
> ___
>
> Q1: [Implication of the result on the eigenspectrum and additional result for NNs]
>
> A1: The results on the eigenspectrum present a slower decay rate when inserting Hadamard product into standard NNs, which further implies that NNs-HP can speed up the learning of high-frequency components while NNs learn the low-frequency components faster. We have conducted the experiment on the spectral bias of noisy labels in a classification task following [1], see Sec. E.6 in the updated version for more details.
>
> ___
>
> Q2: Where are the limitations?
>
> A2: We have already discussed the limitations in Sec. G in our submission, e.g., more results on OOD setting and the linearity in NTK regime. We are thankful to the reviewer for spotting this issue in the checklist and we have fixed it (See line 431).
>
> ___
>
> We hope that our responses have clarified the questions of the reviewer cA3P. If there are any remaining questions, we are happy to discuss further.
>
> ### References
>
> [1] Rahaman, N., Baratin, A., Arpit, D., Draxler, F., Lin, M., Hamprecht, F., Bengio, Y. and Courville, A.. On the spectral bias of neural networks. ICML, 2019.

---

> > ### Comment · Reviewer_cA3P · 2022-08-07
> > **Thanks**
> >
> > Thanks for the response and the additional experiments of section E.6.
> > I'm not sure that I understand Fig.7 correctly. The standard NN (right panels) seems to reach lower values of the validation loss with respect to NNs-Hp (left panels). Shouldn't NNs-Hp be better at learning higher frequencies?
> > In fact, from Fig.7 it seems that standard NN has better loss at all frequencies.

---

> > > ### Author Response · Authors · 2022-08-08
> > > **Clarification on the figures**
> > >
> > > Dear reviewer cA3P,
> > >
> > > We are thankful for your response. Our results are consistent with the results on spectral bias that have been reported in the literature. Concretely, in Fig. 4 of [1], the spectral bias of ReLU nets learns low frequency functions at the **early stage** of training, which results in a dip in the validation loss **early in the training**.
> > >
> > > If this is unclear or the reviewer requires additional information on this interesting phenomenon, we are happy to discuss this further.
> > >
> > > ### References
> > >
> > > [1] Rahaman, N., Baratin, A., Arpit, D., Draxler, F., Lin, M., Hamprecht, F., Bengio, Y. and Courville, A..  On the spectral bias of neural networks. ICML, 2019.

---

### Official Review · Reviewer_ckdZ · 2022-07-09

**Rating:** 5
**Confidence:** 3
**Soundness:** 4 excellent
**Presentation:** 2 fair
**Contribution:** 3 good

**Summary:**

The paper presents a complete study of Neural Tangent Kerel (NTK) for neural networks with Hadamard products (NN-Hps). The equivalence with kernel regression is first established. Based on this result, the authors establish two advantages of NN-Hps over fully-connected NNs: better extrapolation performance and slower-decaying sprectrum.

**Questions:**

1. I am personally not familiar with Gegenbauer polynomials and spherical harmonics. Could the authors explain the implications of the experiment in subsection 4.3? Specifically, how is the degree of the Gegenbauer polynomials related to the frequency of the information?

Moreover, according to Theorem 6, the decay rate of the spectrum is independent with $$N$$ as long as $$N\ge2$$. The experiment, however, shows that increasing $$N$$ makes the decay rate slower. Could the authors further explain this contradiction?

**Limitations:**

The limitations of this paper are well evaluated by the authors.

**Strengths And Weaknesses:**

Strengths:
1. The authors establish two advantages of NN-Hps over fully-connected NNs. Both are important and convincing.

Weaknesses:
1. The remark for Theorem 2 lacks some rigure. If $$\hat K_t$$ and $$\hat K_0$$ are both functions of $$n$$, then one cannot write $$\hat K_t\to\hat K_0$$ as $$n\to\infty$$. Moreover, should the condition be $$m\to\infty$$ rather than $$n\to\infty$$?

2. Theorem 5 requires more interpretation. The requirement that $$\{x_i\}$$ includes the orthogonal basis is quite weird, which may not hold in practical cases. Moreover, this theorem only studies the case of quardratic functions.

3. The figures of the experiment for learning are a bit confusing, and a better explanation is required for different colors. The figure itself is blurry as well.

---

> ### Author Response · Authors · 2022-08-02
> **Response to reviewer ckdZ**
>
> We are thankful to the reviewer ckdZ for their insightful feedback. We address their core concerns below.
>
> ___
> Q1: [Typos in Theorem 2]
>
> A1: We thank the reviewer for pointing out this typo. It should be $m \rightarrow \infty$ (the width) instead of $n$ (the number of training data) in the remark. We have fixed the error in the updated manuscript.
>
> ___
> Q2: Theorem 5 requires more interpretation. The requirement that $x_i$ includes the orthogonal basis is quite weird, which may not hold in practical cases. Moreover, this theorem only studies the case of quadratic functions.
>
> A2: The condition of the orthogonal basis is required in our derivation (see the end of Sec. C.3) since we require there exist at least $d$ linearly independent $x_i$. Since our proof relies on the construction of the feature map of NTK (Lemma 7), which is harder for high-degree cases. For now, we can only consider the case of NNs-HP with one Hadamard product, which intuitively can learn the quadratic functions.
>
> We have added the following remark for this theorem in the updated manuscript. “The above result only considers quadratic functions as our proof heavily relies on the construction of the feature map of NTK, which is harder for high-degree case.”
>
> ___
> Q3: [Blurry figure and its explanation]
>
> A3:  Thanks for the suggestion. In the manuscript, we have exported new, vectorized versions of Fig. 1 and Fig. 2 and revised their corresponding captions.
> ___
>
> Q4: [Implications of the experiment in Sec. 4.3]
>
> A4: The training process and convergence rate of neural networks with infinite width can be characterized by the eigenfunctions of NTK [1]. As a case study, when the input data are uniform on the unit sphere, one can explicitly calculate the eigendecomposition of NTK and thus give explicit convergence rates [2].
>
> Since the normalized Gegenbauer polynomial is a spherical harmonic, the target function in the experiment is a linear combination of several spherical harmonics. When we apply the following Mercer’s decomposition in the form of spherical harmonics, we can see the role of the degree of Gegenbauer polynomials (see Eq. 60-62).
>
> Furthermore, we have added a complementary experiment in Sec. E.6 to show the effect of spectral bias of noisy labels in the classification task, which validates that NNs-HP can speed up the learning of high-frequency components.
> ___
>
> Q5: According to Theorem 6, the decay rate of the spectrum is independent with $N$ as long as $N\geq2$. The experiment, however, shows that increasing $N$ makes the decay rate slower. Could the authors further explain this contradiction?
>
> A5:  This is not a contradiction as $N$ is included in Eq. 67 in our derivation, as presented at the end of Sec. D of the appendix.  We do not explicitly show it just for better comparison with standard neural networks.
>
> To avoid unclear descriptions, we explicitly show it in Theorem 6 in the updated manuscript, i.e.,  $\mu_{k} = \Omega((N^2k)^{-d/2})$ It demonstrates that increasing $N$ makes the spectral decay slower and leads to better approximation properties, which coincides with our experiments.
>
> ___
> We hope that our responses have clarified the questions and improved the opinion of the reviewer ckdZ about our work. If there are any remaining questions, we are happy to discuss further.
>
> ### References
>
> [1] Du, S., Lee, J., Li, H., Wang, L. and Zhai, X.. Gradient descent finds global minima of deep neural networks. ICML, 2019.
>
> [2] Cao, Y., Fang, Z., Wu, Y., Zhou, D.X. and Gu, Q.. Towards understanding the spectral bias of deep learning. IJCAI, 2021.

---

> > ### Author Response · Authors · 2022-08-06
> > **Are there any remaining questions from the reviewer?**
> >
> > Dear reviewer ckdZ,
> >
> > We are thankful for your constructive feedback, e.g. thoroughly checking our claims. As you can notice from [the discussion](https://openreview.net/forum?id=_cXUMAnWJJj&noteId=z-9ldjKEsR2) with the reviewer SBXv, we have also added a new comparison and added several clarifications. We would like to check if you have been covered by our explanations, and improvements in the manuscript or if there are any additional questions we could address.

---

### Official Review · Reviewer_rMyZ · 2022-07-10

**Rating:** 7
**Confidence:** 3
**Soundness:** 2 fair
**Presentation:** 3 good
**Contribution:** 3 good

**Summary:**

This paper derives the NTK of Hadamard-product neural networks (NNs-Hp). Through an analysis of the NTKs of NNs-Hp, the paper shows that NNs-Hp and standard MLPs have different generalization abilities in the extrapolation regime. These understandings are then illustrated using numerical experiments.

**Questions:**

My main question is if the authors can clarify the relationship between NNs-Hp and polynomial neural networks (PNNs). It seems to me that NNs-Hp should be used to refer a generic class of neural networks with Hadamard product. This study, however, is not generally applicable to this wide class of networks. Indeed, as the authors pointed out in line 127-128, their study focuses on polynomial neural networks (PNNs). In that case, I would suggest the authors to position their contribution in the context of PNNs and avoid overloading NNs-Hp with PNNs.

**Limitations:**

The current paper focuses on a specific type of neural network with Hadamard product -- the polynomial neural network. To me, it seems difficult to distinguish whether the phenomenon observed in the numerical experiments is a result of polynomial parametrization or a result of the usage of the Hadamard product in neural networks. It would be great if the authors can clarify this.

In addition, the evaluation of the extrapolation ability of NNs-Hp are confined to toy-like experiments. In my opinions, some of these extrapolation experiments, such as fitting sine curves or spherical harmonics, may be biased toward giving NNs-Hp an advantage by design: The underlying functions are close to polynomials, and PNNs provide a good inductive bias in these tasks. As a result, while it is good to see that NNs-Hp perform better in these cases, their advantage in out-of-distribution generalization in such scenarios are not too surprising.

**Strengths And Weaknesses:**

**Strength**: The main contribution of this paper -- the derivation of NTK of Hadamard-product neural networks -- appears novel and meaningful to me.

**Weakness**:
It is unfortunate that the paper contains a lot of typos and grammatical glitches. Examples include:

Line 128: while we still use the term NN-Hp for consistence -> consistency

Line 242: we access the extrapolation performance beyond synthetic datasets. -> we assess

Line 280: The analysis of the NTK paves the way for knowing interesting proporties -> properties

In the caption of Table 1: (a) Rounding, the output is rounded to the the nearest integer. -> extra "the"

Line 244: This experiment is preformed on ... -> performed

---

> ### Author Response · Authors · 2022-08-02
> **Response to reviewer rMyZ**
>
> We are thankful to the reviewer rMyZ for the insightful feedback. We address the core concerns below.
>
> ___
>
> Q1: [NNs-HP vs PNNs]
>
> A1:
> Our motivation is to study neural networks with Hadamard products. Indeed, polynomial neural networks (PNNs) are a specific type of neural network with Hadamard product (NNs-HP). NNs-Hp also contain other networks, e.g., gated mechanisms, multiplicative RNNs [1]. Nevertheless, it is indeed easy to confuse these two terms. For instance, the Eq. 3 in the paper [2] resembles the formula of PNNs but is not a PNN when the function $g(x,\theta)$ is not a linear (or a piecewise linear) transform.
> The reason we choose PNNs in theoretical analysis is that
> *  PNNs are powerful models that have achieved competitive performance recently in several benchmarks, e.g., [face verification](https://paperswithcode.com/sota/face-verification-on-megaface) of [5],  [face detection](https://paperswithcode.com/sota/face-detection-on-wider-face-hard) of [3].
> * This can simplify our analysis as PNNs have a simple recursive formula with an arbitrary number of Hadamard products.
>
> Furthermore, in our numerical experiment, we plug the Hadamard product into neural networks and compare it with the original network, which is fair. Therefore, we believe there is no overclaim in our paper.
>
> We do agree that a further step, that is both intriguing and challenging, is to extend the results into a general form of NN-Hp. However, at the moment, there is not even a general equation of this class of functions. If the reviewer has any objections, we would be happy to discuss this topic further.
>
>
> ___
> Q2: [Toy-like experiments and Inductive bias]
>
> A2 We respectfully disagree with the reviewer.
>
> The experiment of learning spherical harmonics is a standard experiment from [4]. We also included digit addition experiment (from [6]) and variation of brightness experiment (created by us) on MNIST datasets (Sec. 4.2). In the revised version, we extend our experimental validation further by including an experiment (Sec. E.5) on the VAEC dataset as requested by the reviewer SBXv, and an experiment (Sec. E.6) on the spectral bias of noisy labels in classification tasks following [4]. These experiments are non-toy and present fair comparisons. We are thankful to the reviewer for pointing out that NNs-HP has good inductive bias, recognizing that NNs-HP is valuable.
>
> ___
> Q3: [Grammar mistake]
>
> A3: Thanks for spotting the grammar mistakes. We have carefully checked and fixed them in the updated manuscript.
>
> ___
>
> We believe we have addressed these three main concerns the reviewer has raised. If there are any remaining questions, we are happy to discuss further. Otherwise, we would be grateful if the reviewer can reconsider their score.
>
> ### References
>
> [1] Jayakumar, S.M., Czarnecki, W.M., Menick, J., Schwarz, J., Rae, J., Osindero, S., Teh, Y.W., Harley, T. and Pascanu, R.. Multiplicative interactions and where to find them. ICLR, 2020.
>
> [2] Fathony, R., Sahu, A.K., Willmott, D. and Kolter, J.Z.. Multiplicative filter networks. ICLR, 2020.
>
> [3] Babiloni, F., Marras, I., Kokkinos, F., Deng, J., Chrysos, G. and Zafeiriou, S.. Poly-NL: Linear Complexity Non-local Layers With 3rd Order Polynomials. ICCV, 2021.
>
> [4] Rahaman, N., Baratin, A., Arpit, D., Draxler, F., Lin, M., Hamprecht, F., Bengio, Y. and Courville, A.. On the spectral bias of neural networks. ICML, 2019.
>
> [5] Chrysos, G.G., Moschoglou, S., Bouritsas, G., Panagakis, Y., Deng, J. and Zafeiriou, S.. $Π$-nets: Deep polynomial neural networks. CVPR, 2020.
>
> [6] Bloice, M.D., Roth, P.M. and Holzinger, A.. Performing arithmetic using a neural network trained on digit permutation pairs. International Symposium on Methodologies for Intelligent Systems, 2020.

---

> > ### Comment · Reviewer_rMyZ · 2022-08-06
> > **Response to the rebuttal**
> >
> > I'd like to thank the authors for their response. I find the extended experiment section very helpful.
> >
> > Regarding the NNs-HP vs PNNs issue, however, I am not entirely convinced. To my understanding, all theoretical analysis and numerical experiments of the paper focus on and only on polynomial networks, which is a small proper subset of NNs-HP. Indeed, as the authors themselves pointed out, there are plenty of NNs-HPs that are not PNNs, including StyleGAN, multiplicative filter networks, multiplicative RNNs, etc. To these non-PNN architectures, the analysis of the paper is not directly relevant. In that case, wouldn't it be more suitable to position the contributions in the scope of PNNs, instead of NNs-HPs? The authors may argue that their *motivation* is to study NNs-HP, but because this is a class too complex to analyze, they only study PNNs instead. But this is exactly the problem - if the results do not apply to NNs-HPs in general, it may be advisable to not present them in a way as if they are, in the title and the main text.

---

> > > ### Author Response · Authors · 2022-08-08
> > > **Updates to the paper to better reflect the theoretical contributions; theoretical and empirical extensions beyond PNNs**
> > >
> > > Dear reviewer rMyZ,
> > >
> > > We are thankful for your response. Additionally, we appreciate the acknowledgment of our extended experiments.  We agree with the reviewer that our *theoretical framework* mainly centers around PNNs instead of general NNs-HP. To position our work better in the literature, we have made a number of extensions and updates in the manuscript, which we are describing below.
> > >
> > > According to the reviewers’ suggestions, we have made the following updates in the paper:
> > >
> > > * We updated the title to “**Extrapolation and Spectral Bias of Neural Nets with Hadamard Product: a Polynomial Net Study**”. We polished our introduction/conclusion, and clarified our contribution to reflect this new perspective.
> > >
> > > * We illustrate more clearly how the results we have derived can be extended beyond PNNs. This is a core issue you had identified, and we indeed agree that extending it to other classes of functions is an interesting avenue. To that end, we have added theoretical and experimental results of **multiplicative filter network** (MFN) [1] in Appendix (Sec. F). The experiment in arithmetic extrapolation and visual analogy extrapolation demonstrate the benefits over standard NNs.
> > >
> > > * Furthermore, we analyze one additional type of NNs-HP, i.e., **non-local networks with Hadamard product** [2] by deriving its neural tangent kernel and its extrapolation behavior (See detail in Sec. G). Note that to our knowledge, there is no general equation for NNs-HP, which we could analyze and include as special cases all of the above. This is consistent with NNs that vary significantly (e.g. feedforward nets from recurrent nets, NNs-HP, etc).
> > >
> > > * We emphasize that the experiment on the VAEC dataset that we added in the previous revision was already a non-polynomial net. That is, we simply converted the feedforward NN into an NN-HP by inserting Hadamard products. So, we already had experimental evidence beyond PNNs. We have significantly expanded on the NNs-HP as we exhibit above.
> > >
> > > Overall, we are thankful for encouraging us to extend our work further and make *both theoretical and empirical contributions beyond the PNNs*, such that we expand the scope of our work.
> > >
> > > If the reviewer has any further questions or suggestions, we are happy to elaborate further.
> > >
> > > ____
> > > ### Further details on the newly added experiments
> > >
> > > 1. Additional experiment on arithmetic extrapolation with MFN:
> > >
> > >  _|Rounding|Floor/ceiling| $\pm$1|
> > > :--|:--|:--|:--|
> > > NN (Dense)|43.6%|80.5%|88.7%
> > > MFN (Dense)|**72.0**%|**87.4**%|**91.6**%
> > >
> > > _|Rounding|Floor/ceiling| $\pm$1|
> > > :--|:--|:--|:--|
> > > NN (Conv)|61.7%|91.8%|95.3%
> > > MFN (Conv)|**82.4**%|**92.5**%|**95.4**%
> > >
> > > 2. Additional experiment on Visual analogy extrapolation with MFN:
> > >
> > > _|Training ($\alpha=1$)|Ext ($\alpha=2$)| Ext ($\alpha=3$)| Ext ($\alpha=4$)|Ext ($\alpha=5$)| Ext ($\alpha=6$)
> > > :--|:--|:--|:--|:--|:--|:--
> > > NN|99.5%|**76.2**%|55.5%|46.3%|42.6%|40.4%
> > > MFN|99.1%|74.5%|**56.2**%|**47.1**%|**43.0**%|**40.6**%
> > >
> > > ### References
> > >
> > > [1]  Fathony, R., Sahu, A.K., Willmott, D. and Kolter, J.Z.. Multiplicative filter networks. ICLR, 2020.
> > >
> > > [2] Babiloni, F., Marras, I., Kokkinos, F., Deng, J., Chrysos, G. and Zafeiriou, S.. Poly-NL: Linear Complexity Non-local Layers With 3rd Order Polynomials. ICCV, 2021.

---

> > > > ### Comment · Reviewer_rMyZ · 2022-08-09
> > > > **Thanks for the clarification!**
> > > >
> > > > I would like to thank the authors for their clarification and their additional analysis & experiments on non-PNN architectures such as the multiplicative filter networks in Appendix F. To my knowledge, the derivation of the neural tangent kernel of MFN is a novel contribution. Additionally, separating the theoretical and empirical contributions indeed addresses my concerns! I have lifted my score accordingly.

---

### Official Review · Reviewer_SBXv · 2022-07-11

**Rating:** 7
**Confidence:** 5
**Soundness:** 3 good
**Presentation:** 3 good
**Contribution:** 3 good

**Summary:**

This paper studies NNs with Hadamard products in relation to NTKs. Then it compares the NNs-Hp with standard neural networks with respect to spectral bias and extrapolation. Contributions of the paper include both theory and experiments.

**Questions:**

I would like to see more experiments on extrapolation datasets used in the literature. Specifically, VAEC dataset seems like a good choice as authors have already cited the paper that introduced it.

There should be discussion and clarification making it clear what authors mean by extrapolation vs OOD.

All image classification tasks require extrapolation. Authors can experiment with standard image classification datasets to demonstrate effectiveness of their analysis regarding extrapolation. See this paper:

Balestriero, R., Pesenti, J. and LeCun, Y., 2021. Learning in high dimension always amounts to extrapolation. arXiv preprint arXiv:2110.09485.


If authors would like to extrapolate with feedforward networks, they can consider the models trained by Neyshabur on MNIST and CIFAR datasets.

Neyshabur, B., 2020. Towards learning convolutions from scratch. Advances in Neural Information Processing Systems, 33, pp.8078-8088.

In the introduction subsection on extrapolation, I am not sure why there is emphasis on limitation of other models in non-linear extrapolation. For feedforward networks, one can easily use non-linear activation functions, and that could lead to non-linear extrapolation and interpolation.

I'm also not sure why authors don't consider interpolation properties of their model when they provide such analysis for extrapolation. These seem to belong to a single paper, and not two separate papers?

Have authors considered the fact that for image classification, all testing samples fall outside the convex hull of training set and all classifications require some considerable amount of extrapolation?

When authors use the term extrapolation, do they mean any function evaluation outside the convex hull of training set? Sometime authors seem to interchangeably refer to this as out-of-distribution. These two terms are not necessarily equivalent.

**Limitations:**

I think authors have not discussed the limitations of their work. Their answer to the questionnaire claims that limitations are explained in Section 5, but I don't see such discussion in the single paragraph of Section 5.

**Strengths And Weaknesses:**

Strengths:
This is an interesting work. The approach and the theory appear to be sound.

Weaknesses:
Empirical results can be improved in my opinion. Specifically, for extrapolation there are benchmark datasets that authors have not considered. It would be insightful to see their results on such datasets and the comparison with prior work. For example, authors can experiment with the VAEC dataset provided by Webb et al which they have cited.

Webb, T., Dulberg, Z., Frankland, S., Petrov, A., O’Reilly, R. and Cohen, J., 2020. Learning representations that support extrapolation. In International conference on machine learning (pp. 10136-10146). PMLR.

There seems to be inaccuracies in the literature review of extrapolation. For example, the statement "Xu et al. [2021] theoretically and empirically point out that 2-layer fully-connected NNs can only extrapolate to linear function." seems inaccurate, because Xu et al make that claim only for NNs with ReLU activation. If someone wants to extrapolate non-linearly, they can use NNs with non-linear activation functions.

There are more inaccuracies in that literature review. For example, the statement "On the contrary, NNs usually lack such [extrapolation] ability" again seems inaccurate. Note that all image classification tasks amount to extrapolation and NNs have the ability to perform such extrapolations. For example, see the reference below from NeurIPS 2002 which uses SVMs to extrapolate and build a classification model for the MNIST dataset.

Haffner, P., 2001. Escaping the convex hull with extrapolated vector machines. Advances in Neural Information Processing Systems, 14.

Therefore, to study extrapolation capability of NNs, one dataset to experiment with is the actual MNIST dataset.

The remark on lines 145-148 seems unclear and hand wavy. What is the contribution of this paper with regards to Theorem 1 and the value of \rho?

---

> ### Author Response · Authors · 2022-08-02
> **Response (2 out of 2) to reviewer SBXv**
>
> ___
>
> Q6: [Interpolation vs extrapolation]
>
> A6: Interpolation and extrapolation are two complementary topics. In the interpolation regime, both theoretical and empirical evidence demonstrate that neural networks are able to interpolate to (noisy) data well, achieve (nearly) zero training loss, and still generalize well, a.k.a, benign overfitting [5,6]. Extrapolation is a harder task compared with interpolation both theoretically [3] and experimentally (e.g., visual analogy [7] and mathematical reasoning [8][9]).
> Accordingly, extrapolation is often regarded as an independent topic, as suggested by previous work. We follow this setting, center around extrapolation, and demonstrate the separation on the extrapolation ability of different models.
> ___
>
> Q7: [Inaccuracies in the literature review]
>
> A7: We revised as follows in the updated version: “Xu et al. [2021] theoretically and empirically point out that 2-layer fully-connected NNs with ReLU can only extrapolate to linear function.” See line 47 for details.
>
> We revised as follows in the updated version: “Neural networks have demonstrated weaknesses in extrapolating simple arithmetic problems [Saxton et al., 2019] or learning simple functions [Haley and Soloway, 1992, Sahoo et al., 2018].” See line 42-44 for details.
>
> ___
>
>
> Q8: The remark on lines 145-148 seems unclear and hand wavy. What is the contribution of this paper with regards to Theorem 1 and the value of $\rho$?
>
> A8: Theorem 1 shows the inner product of the Jacobian converges to the NTK at initialization, which has not been studied before for NNs-Hp. By doing so, this theorem allows us to further analyze the extrapolation of networks from the perspective of NTK. The remark we add here is to show $\rho \approx 2^N$ is still a constant as the degree $N$ in practice is not large. According to your suggestions,  we polish this remark in the updated manuscript (See lines 144-146).
>
> ___
> Q9: I think authors have not discussed the limitations of their work.
>
> A9: We have already discussed the limitations in Sec. G in our submission, e.g., more results on the OOD setting and the linearity in NTK regime. We are thankful to the reviewer for spotting this issue in the checklist and we have fixed it (See line 431).
>
> ___
> We hope that our responses have clarified the questions of the reviewer SBXv. If there are any remaining questions, we are happy to discuss further.
>
> ### References
>
> [1] Ye, H., Xie, C., Cai, T., Li, R., Li, Z. and Wang, L.. Towards a theoretical framework of out-of-distribution generalization. NeurIPS, 2021.
>
> [2] Balestriero, R., Pesenti, J. and LeCun, Y.. Learning in high dimension always amounts to extrapolation. arXiv preprint arXiv:2110.09485, 2021.
>
> [3] Xu, K., Zhang, M., Li, J., Du, S.S., Kawarabayashi, K.I. and Jegelka, S.. How neural networks extrapolate: From feedforward to graph neural networks. ICLR, 2021.
>
> [4] Chrysos, G., Georgopoulos, M. and Panagakis, Y.. Conditional generation using polynomial expansions. NeurIPS, 2021.
>
> [5] Zhang, C., Bengio, S., Hardt, M., Recht, B. and Vinyals, O.. Understanding deep learning (still) requires rethinking generalization. Communications of the ACM, 2021.
>
> [6] Frei, S., Chatterji, N.S. and Bartlett, P.. Benign overfitting without linearity: Neural network classifiers trained by gradient descent for noisy linear data. Conference on Learning Theory, 2022.
>
> [7] Webb, T., Dulberg, Z., Frankland, S., Petrov, A., O’Reilly, R. and Cohen, J.. Learning representations that support extrapolation. ICML, 2020.
>
> [8] Saxton, D., Grefenstette, E., Hill, F. and Kohli, P.. Analysing mathematical reasoning abilities of neural models. ICLR, 2019.
>
> [9] Bloice, M.D., Roth, P.M. and Holzinger, A.. Performing arithmetic using a neural network trained on digit permutation pairs. International Symposium on Methodologies for Intelligent Systems, 2020.
>
> [10] Chrysos, G.G., Moschoglou, S., Bouritsas, G., Panagakis, Y., Deng, J. and Zafeiriou, S.. $Π$-nets: Deep polynomial neural networks. CVPR, 2020.
>
> [11] Jayakumar, S.M., Czarnecki, W.M., Menick, J., Schwarz, J., Rae, J., Osindero, S., Teh, Y.W., Harley, T. and Pascanu, R.. Multiplicative interactions and where to find them. ICLR, 2020.
>
> [12] Babiloni, F., Marras, I., Kokkinos, F., Deng, J., Chrysos, G. and Zafeiriou, S.. Poly-NL: Linear Complexity Non-local Layers With 3rd Order Polynomials. ICCV, 2021.

---

> ### Author Response · Authors · 2022-08-02
> **Response (1 out of 2) to reviewer SBXv**
>
> We thank the reviewer SBXv for the insightful feedback. We address the concerns below.
> ___
> Q1: [Experiment on VAEC dataset]
>
> A1: We are thankful to the reviewer for the suggestion on this challenging dataset. According to your suggestions, we have included the experimental result on the scale extrapolation experiment on the VAEC dataset, refer to Sec. E.5 in the updated manuscript for details.
> We use the best model in the original paper [7] as baseline (NN) and insert Hadamard products (to convert it into an NN-HP) to compare. We run each method eight times and report the mean of accuracy below. ‘Ext’ abbreviates ‘extrapolation’.
>
> _|Training ($\alpha=1$)|Ext ($\alpha=2$)| Ext ($\alpha=3$)| Ext ($\alpha=4$)|Ext ($\alpha=5$)| Ext ($\alpha=6$)
> :--|:--|:--|:--|:--|:--|:--
> NNs|99.5%|**76.2**%|55.5%|46.3%|42.6%|40.4%
> NNs-HP|99.7%|73.7%|**57.5**%|**49.0**%|**45.3**%|**42.9**%
>
> These results comprehensively evaluate the extrapolation ability of NNs-HP and show that NNs-HP extrapolate better than standard NNs.
> ___
>
> Q2: [Extrapolation vs OOD]
>
> A2: We are thankful to the reviewer for pointing out this issue. Supervised learning algorithms often assume that the training and testing data are independent and identically distributed. If this does not hold, it falls in the out-of-distribution (OOD) regime [1]. The definition of OOD is straightforward and clear but extrapolation admits different definitions, e.g., [2] [3].
> * [2]: Extrapolation occurs for a sample when it falls outside of that convex hull of the training set.
> * [3]: Extrapolation addresses predictions on a domain that is larger than the support of the training distribution.
>
> In this work, we follow the definition in [3], and accordingly, we do some experiments on learning simple functions to verify the extrapolation regime that is larger than the support when choosing a probability measure over a compact set.
> In the updated manuscript, precisely, we used the extrapolation terminology instead of OOD.
> ___
> Q3:  [All image classification tasks require extrapolation]
>
> A3: We agree with the reviewer that all image tasks belong to extrapolation due to the high-dimensional setting according to [2]. Though NNs perform well on image classification tasks, this does not mean that NNs could extrapolate well on complicated tasks or any nonlinear smooth function. In fact, different models have different extrapolation abilities, e.g., SVM (as the reviewer mentioned) vs. NNs (with different architectures). Such separation on subtle extrapolation ability of different architectures, e.g., NNs vs. NNs-Hp, is the main target of this work.
>
> We show this separation on various experiments. For instance, in our experiment on digit addition, calculating the sum of unseen pairs is a very clear and not easy extrapolation. In our experiment on variation of brightness, the data in the extrapolation regime has a completely different distribution compared with the training data.
> ___
> Q4: [Experiments on standard image classification datasets]
>
> A4: Our goal is not to devise new architectures for image classification here, but rather to use existing architectures to **extrapolate on more complicated settings** as we emphasized in the previous question. Having said that, NNs-Hp have demonstrated strong performance in a range of papers, including [face verification](https://paperswithcode.com/sota/face-verification-on-megaface) of [10], multi-task learning [11], [face detection](https://paperswithcode.com/sota/face-detection-on-wider-face-hard) of [12], image generation [4], etc. Nevertheless, in the experiments of digit addition and variation of brightness with MNIST data, we also report the performance on the regime similar to the training set  (Sec. 4.2).
> ___
>
> Q5: [Extrapolation with NNs with non-linear activation function]
>
> A5: In theory, different activation functions would affect the test performance. However, using activation functions beyond ReLU still lack enough approximation ability in the extrapolation regime. For example, [3, Sec. 3.3] supports that even with a tanh activation function, NNs cannot approximate well the target quadratic function, and vice versa. In this sense, we believe that models in nonlinear extrapolation are important and deserve to be studied. Our work follows the problem setting of [3] using the ReLU NNs for analyses since ReLUs are the most popular activation functions used in practice.

---

> > ### Comment · Reviewer_SBXv · 2022-08-04
> > **[Convex hull vs Support] of training set**
> >
> > Thank you for your clear and thoughtful response. I am glad to see that your method shows an improvement on the VAEC extrapolation dataset.
> >
> > The only remaining question for me is about the distinction you make under Q2, in the rebuttal, citing your references [2] and [3]. It is not clear to me what is the definition for "support of training set" and how it can be different from the convex hull of training set. Do you have a formal definition for the "support of training set"? If you don't have a formal definition, how do you distinguish between the "support of training distribution" and the "convex hull of training set" in practice?

---

> > > ### Author Response · Authors · 2022-08-05
> > > **Response to the reviewer SBXv**
> > >
> > > We are thankful to the reviewer SBXv for their response.
> > >
> > > The formal definition of “support of training set” depends on “support of a random variable”,
> > >  refer to page 31 in [1]:
> > > > Consider a probability space $(\Omega, F, P)$, a random variable $x: \Omega \rightarrow X$  for which there exists a probability density function $p_{x}$. Define the support set of the random variable $x$ as the set $X_{support}=$ Closure(\{ ${ v \in X \mid p_{x}(v)>0  }$  \}).
> > >
> > > Here we give a 2-D example to show the difference between the definition of “support of training set” and “convex hull”. The convex hull excludes the setting of data on the unit sphere (i.e., circle in the 2-D case), which can be described by support of the training set. Nevertheless, a convex hull of training data sometimes can cover different domains beyond the support of the training set. For example, considering training data are in a three-quarter pie (see link https://imgur.com/a/VOysKas). The support of training data is the closure of this pie; the convex hull of training data can include some region of the remaining one-quarter pie.
> > >
> > > Importantly, we do not believe one definition is better than the other. Even though we follow the definition regarding the support from [2], **both of the definitions are suitable for our experiments**. For instance, for an illuminating example in our experiment of learning simple functions, see the table below. In this case, the domain of our extrapolation data is out of the support of training data and the convex hull of the training data.
> > >
> > > Data| Dom(X)
> > > :--|:--
> > > Training distribution (uniform)|$[-2.5, 2.5]$
> > > Extrapolation distribution  (uniform)|$[-5, -2.5] \cup [2.5, 5]$
> > > Support of training data|$[-2.5, 2.5]$
> > > Convex hull of training data|$[a, b]$ where $-2.5<=a<=b<2.5$
> > >
> > > If there are any remaining questions, we are happy to discuss further.
> > >
> > > ### References
> > >
> > > [1] van Schuppen, J.H.. Control and System Theory of Discrete-Time Stochastic Systems. 2021.
> > >
> > > [2] Xu, K., Zhang, M., Li, J., Du, S.S., Kawarabayashi, K.I. and Jegelka, S.. How neural networks extrapolate: From feedforward to graph neural networks. ICLR, 2021.

---

> > > > ### Comment · Reviewer_SBXv · 2022-08-05
> > > > **Clear**
> > > >
> > > > Thank you for the response. This clarification is convincing, especially in the context of the experiments in section 4.1 in the paper.
> > > >
> > > > For other applications, such as image classification and the ones you consider in section 4.2, you may need specialized methods, e.g., OOD detection methods, in order to distinguish between the above two definitions. For certain input types, e.g., images, drawing a line that defines what is inside or outside a distribution is often intractable. But the convex hull of training set can be clearly identified in most applications.
> > > >
> > > > Please note that Figure 3 does not appear correctly in the revised version of the paper.
> > > >
> > > > This discussion addressed my concerns, and I'll submit a revised score according to NeurIPS guidelines.

---

> > > > > ### Author Response · Authors · 2022-08-06
> > > > > **Thank you for your multiple rounds of responses**
> > > > >
> > > > > Dear reviewer SBXv,
> > > > >
> > > > > Thank you for your prompt response to our remarks, and for letting us know about the issue with Fig. 3. The figure has been updated. In addition, we added a remark for the applications using the convex hull in the supplementary (sec. E.1; line 757-759):
> > > > > “In our work, the extrapolation relies on the support of the training data~\citep{van2021control}, as suggested by the previous work of \citet{xu2021how}. We note that for certain applications and input data types, the convex hull might be required, however, we leave this as future work.”.

---

### Author Response · Authors · 2022-08-02
**General response**

Dear reviewers and AC,

We appreciate your insightful comments. In our responses, we would like to provide a number of extensions to our work. Particularly, we bear the following extensions in this revision:

* We include additional extrapolation experiments on the VAEC dataset [1] as requested by the reviewer SBXv.
* We improve the readability of the paper: we add clarifications (and remarks) as requested by the reviewers.
* We include experiments of learning noisy label data in classification on MNIST dataset to confirm that high-degree NNs-HP pick up the high-frequency information faster than NNs. This was requested by the reviewers ckdZ and cA3P.

The changes in the manuscript are denoted in red for convenience.

We are grateful for the time and the effort of the reviewers. We are confident that the updates have made our submission stronger; we hope that the reviewers and the AC will appreciate the updated work even more.

### References

[1] Webb, T., Dulberg, Z., Frankland, S., Petrov, A., O’Reilly, R. and Cohen, J.. Learning representations that support extrapolation. ICML, 2020.

---

### Author Response · Authors · 2022-08-09
**Updates during the rebuttal; we appreciate the constructive feedback**

Dear reviewers and AC,

**Tl-dr**: We are truly thankful for all your insightful comments. Those constructive comments have enabled us to make a number of updates in the paper that we believe have further clarified our contributions, strengthened our theoretical contributions (e.g. new proofs in NNs-HP), and provided new empirical evidence (e.g. experiments on VAEC dataset).

## Detailed update:

In details, we have made the following updates in the manuscript during the rebuttal:

* We include additional experiments on a standard dataset for extrapolation. The VAEC dataset [1] was selected (as suggested by the reviewer SBXv) and it validates the improvement of NNs-Hp (Sec. E5).

* We refine the perspective on extrapolation as appearing in our work and recent papers. This will illuminate the context of our proofs better as the reviewer SBXv identified.

* We revise the theoretical contributions with respect to PNNs and NNs-HP as suggested by the reviewer rMyZ.

* We extend our theory to other types of NNs-HP. In particular, we consider two recent networks proposed for this end: a) the multiplicative filter network (MFN) [2] and the non-local networks with Hadamard product [3]. Our newly added proofs can be found in sec. F, G. These results extend our proofs well beyond the PNNs as suggested by the reviewer rMyZ.

* To validate our newly added results on other types of networks, we conduct a new set of experiments on MFN (arithmetic extrapolation and visual analogy extrapolation), which can be found in sec. F.

* We conduct experiments on learning noisy label data in classification on MNIST dataset to confirm the spectral bias (Sec. E6). This was motivated by the suggestion from reviewers ckdZ and cA3P.

* We improve the readability of the paper: we add clarifications (and remarks) as requested by the reviewers.

* Once again, we are grateful for the time and the effort of the reviewers. We are confident that the updates have made our submission stronger. We are happy to further improve the paper in the final version if the reviewers have additional suggestions.

### References

[1] Webb, T., Dulberg, Z., Frankland, S., Petrov, A., O’Reilly, R. and Cohen, J.. Learning representations that support extrapolation. ICML, 2020.

[2]  Fathony, R., Sahu, A.K., Willmott, D. and Kolter, J.Z.. Multiplicative filter networks. ICLR, 2020.

[3] Babiloni, F., Marras, I., Kokkinos, F., Deng, J., Chrysos, G. and Zafeiriou, S.. Poly-NL: Linear Complexity Non-local Layers With 3rd Order Polynomials. ICCV, 2021.

---

### Meta-Review · Area_Chair_vPgP · 2022-08-27

**Recommendation:** Accept
**Confidence:** Certain

**Metareview:**

This paper derives the NTK of Hadamard-product neural networks and shows the different behavior from the standard neural networks in terms of spectral bias in the extrapolation regime. After the author response and author-reviewer discussion, all the reviewers are in support of accepting this paper. Therefore, I recommend acceptance. Regarding the NTK-based optimization analysis, it seems that the following paper is not mentioned, which is a concurrent work with [Allen-Zhu et al.,2019, Chizat et al., 2019, Du et al., 2019a, 2018].
[*] Zou et al. Gradient Descent Optimizes Over-parameterized Deep ReLU Networks, Machine Learning, 2020. Please address the missing reference and prepare the camera ready by incorporating the author response.

**Award:**

No

---

### Decision · Program_Chairs · 2022-09-14

Accept